# Mesoscale dynamics of an intrathermocline eddy in the Canary Eddy Corridor

Luis P. Valencia[1], Ángel Rodríguez-Santana[1], Borja Aguiar-González[1], Javier Arístegui[2], Xosé A. Álvarez-Salgado[3], Josep Coca[4], María Dolores Gelado-Caballero[5], and Antonio Martínez-Marrero[2]

[1]University Institute for Research in Sustainable Aquaculture and Marine Ecosystems (IU-ECOAQUA), University of Las Palmas de Gran Canaria (ULPGC), Las Palmas, Spain
[2]Institute of Oceanography and Global Change (IOCAG), University of Las Palmas de Gran Canaria (ULPGC), Las Palmas, Spain
[3]Institute of Marine Research, Spanish National Research Council (CSIC), Vigo, Spain
[4]Cartográfica de Canarias S.A. (GRAFCAN), Las Palmas, Spain
[5]Department of Chemistry, University of Las Palmas de Gran Canaria (ULPGC), Las Palmas, Spain

**Correspondence:** Luis P. Valencia (luis.valencia102@alu.ulpgc.es)

**Abstract.** High-resolution observations of an intrathermocline eddy were conducted in November 2022 within the Canary Eddy Corridor. Formed in early summer 2022, this mature mesoscale eddy exhibited a vertical extent of 550 m, with its core centered at 110 m depth, and a segmented horizontal structure comprising a 23 km-radius solid-body core surrounded by a 47 km-wide outer ring. Propagating southwestward at 4.5 km d$^{-1}$, its motion was consistent with the phase speed of a first-mode baroclinic Rossby wave. The eddy's rotational dynamics featured a 3.9-day inner-core rotation period shaped by stratification, leading to the formation of distinct rotational layers. Rossby number estimates (maximum of -0.7 at the center, and -0.5 on average) and low core potential vorticity ($\sim 10^{-11}$ m$^{-1}$ s$^{-1}$, 90% lower than surrounding values) revealed a regime dominated by planetary rotation, yet with a dynamically significant centripetal contribution—suggestive of a cyclogeostrophic momentum balance—and strong water mass isolation. Burger numbers, ranging from 1.27 to 0.14 (length-scale based) and from 0.21 to 0.69 (energy-based), underscored the role of stratification and buoyancy forces in shaping the eddy's vertical structure. The eddy carried available heat and salt anomalies of $6.550 \times 10^{18}$ J and $0.015 \times 10^{12}$ kg, driving heat and salt (freshwater equivalent) fluxes of $4.60 \times 10^{12}$ W and $0.42 \times 10^{9}$ kg s$^{-1}$ (-0.012 Sv, where 1 Sv = $10^6$ m$^3$ s$^{-1}$), highlighting its role in transporting coastal upwelling waters into the ocean interior. The intrathermocline nature of the Bentayga eddy appears to have developed during its growth phase, likely driven by surface convergence linked to the interaction with upwelling filaments and subsequent isopycnal deepening as it propagated offshore. Low dissolved oxygen concentrations (100–110 $\mu$mol kg$^{-1}$) and low apparent oxygen utilization (20–30 $\mu$mol kg$^{-1}$) within the eddy core support the hypothesis of recent trapping of surface-derived upwelled waters. Over the course of its year-long lifespan, the eddy experienced intrinsic instabilities and eddy-to-eddy interactions, ultimately decaying by early summer 2023. The distinct properties of this eddy, together with the apparent variability among similar features in the Canary Eddy Corridor, underscore the need for expanded quasi-synoptic high-resolution studies. Comprehensive observational programs and advanced numerical simulations are essential to better understand the role of ITEs as zonal pathways for heat, salt, and biogeochemical properties within regional ocean circulation.

# 1 Introduction

Mesoscale activity in the central portion of the Canary Upwelling System (cCANUS, 25–30°N) is characterized by mesoscale eddies, upwelling fronts, and filaments, which are defining features of this Eastern Boundary Upwelling System (EBUS) (Arístegui et al., 1997; Sangrà et al., 2009; Pacheco et al., 2001; Marchesiello and Estrade, 2009; Hernández-Hernández et al., 2020). Similar to other EBUS, mesoscale variability in this region is driven by instabilities that develop along the transition zone between the coastal upwelling jet and the open ocean flow within the northwest African upwelling system (e.g., Barton et al., 1998; Pelegrí et al., 2005; Benítez-Barrios et al., 2011; Ruiz et al., 2014). Additionally, the presence of islands and archipelagos in cCANUS provides a supplementary mechanism for generating and transforming mesoscale phenomena (e.g., Barton, 2001; Caldeira et al., 2014; Ruiz et al., 2014; Cardoso et al., 2020).

The Canary Archipelago (∼28°N) is a key topographic feature within cCANUS, significantly contributing to the generation of mesoscale structures that form offshore (e.g., Arístegui et al., 1994, 1997; Sangrà et al., 2009). The islands act as topographic barriers, obstructing the southwestward flow of the Canary Current (CC) and the Trade Winds (La Violette, 1974; Van Camp et al., 1991; Hernández-Guerra et al., 1993; Arístegui et al., 1994, 1997; Basterretxea et al., 2002). As a result, mesoscale eddies are generated and detached in a Von Kármán street pattern, propagating southwest of the archipelago (e.g., Chopra and Hubert, 1965; Chopra, 1973; Barton, 2001).

In the wake of the islands, wind shear imparts additional momentum and vorticity to the CC outflow, enabling the detachment of eddies even under relatively weak CC flow (Sangrà et al., 2005, 2007; Jiménez et al., 2008; Piedeleu et al., 2009). These detached eddies typically have lifespans of at least three months, propagate southwestward, and travel thousands of kilometers from their origin (Pingree, 1996; Pingree and Garcia-Soto, 2004; Sangrà et al., 2005, 2007, 2009; Barceló-Llull et al., 2017). This dynamic process defines the Canary Eddy Corridor (CEC) (Sangrà et al., 2009), a permanent feature of cCANUS that provides a pathway for transporting the physical and biogeochemical properties of water masses from this EBUS into the oligotrophic ocean interior.

The CEC extends from 22°N to 29°N and is predominantly populated by long-lived anticyclonic eddies (AEs) generated south of Tenerife and Gran Canaria islands (Sangrà et al., 2005, 2007, 2009). The dominance of these AEs has been extensively documented through a combination of in situ and satellite observations (Pingree, 1996; Pingree and Garcia-Soto, 2004; Sangrà et al., 2009), as well as various eddy detection and tracking methods (Chaigneau et al., 2009; Sangrà et al., 2009; Mason et al., 2014). Notably, Pegliasco et al. (2015) reported that 40% of the long-lived AEs within the CEC exhibit subsurface intensification, with warm and salty anomalies extending to depths of up to 800 m.

Subsurface-intensified mesoscale AEs, also referred to as intrathermocline eddies (ITEs), are characterized by a lens-shaped core located at subsurface depths within the thermocline (Dugan et al., 1982; Kostianoy and Belkin, 1989; Armi and Zenk, 1984). Their typical structure includes dome-shaped isopleths at the upper boundary and bowl-shaped isopleths at the lower boundary, enclosing a relatively homogeneous core (pycnostad) with properties similar to the water masses at their formation site (e.g., Dugan et al., 1982; Armi and Zenk, 1984; Shapiro and Meschanov, 1991; Gordon et al., 2002; Johnson and Mctaggart, 2010; Hormazabal et al., 2013; Thomsen et al., 2016).

As these eddies propagate into the ocean interior, they induce significant hydrographic and biogeochemical anomalies relative to surrounding waters (e.g., Chaigneau et al., 2011; Pegliasco et al., 2015; Cornejo et al., 2016; Machín and Pelegrí, 2016; Barceló-Llull et al., 2017; Karstensen et al., 2017). The intensity of these anomalies depends on the integrity of the core during propagation. A useful proxy indicator of this is the non-linearity parameter ($\vartheta$), defined as: $\vartheta = \frac{U_{\max}}{c}$, where $U_{\max}$ is the maximum circum-averaged speed of the eddy and $c$ its translational speed. Non-linearity is characterized by $\vartheta > 1$, indicating trapped waters within the eddy, while highly non-linear eddies exhibit $\vartheta > 5$, with extreme cases exceeding 10 (Chelton et al., 2011).

The ability to trap water is intrinsically linked to the core structure, where the presence of a pycnostad reduces static stability and leads to large reductions in potential vorticity (PV) (e.g., Nauw et al., 2006; Johnson and Mctaggart, 2010; Pidcock et al., 2013; Barceló-Llull et al., 2017; Bosse et al., 2019). The low-PV core of ITEs, resulting from diminished stratification, not only enhances their capacity to isolate and transport water masses but also establishes a strong PV gradient that acts as a dynamic barrier to lateral exchanges, particularly below the mixed layer (Bosse et al., 2019).

Observational evidence confirms the presence of ITEs in many regions of the world's oceans. However, comprehensive studies on their formation and early evolution are still limited (Caldeira et al., 2014), and most of the current knowledge comes from numerical simulations under both realistic and idealized oceanic conditions. Thus, understanding the origin of ITEs remains an active area of research.

Two foundational studies have proposed important theoretical pathways for the formation of subsurface eddies. McWilliams (1988) demonstrated that localized diapycnal mixing in a stratified ocean, followed by nonlinear geostrophic adjustment, can give rise to anticyclonic lenses with low stratification and coherent rotation. The resulting submesoscale coherent vortices (SCVs) closely resemble observed ITEs and are particularly relevant in settings where vertical mixing is induced by filaments or frontal structures in upwelling systems. Complementarily, D'Asaro (1988) described a bottom boundary layer mechanism in which friction over the inner slope of a submarine canyon generates anticyclonic vorticity within dense coastal currents of reduced PV. As these low-PV parcels detach from the slope, they can evolve into SCVs that undergo geostrophic adjustment, forming coherent subsurface vortices. This mechanism may also contribute to the early stages of ITE formation in topographically complex coastal environments, including island wakes and continental slopes.

Hogan and Hurlburt (2006) identified three non-exclusive mechanisms of ITE formation in a realistically simulated Japan/East Sea:

1. advection of stratified water capping a pre-existing anticyclone,

2. restratification of the upper layers in a pre-existing anticyclone due to seasonal heating or cooling, and

3. frontal convergence and subduction of winter surface mixed layer water.

Frontal convergence and subduction have also been proposed as the formation mechanism for ITEs in the Southern Indian Ocean by Nauw et al. (2006). Further insights on frontal subduction as a source for ITE core waters were provided by Thomas (2008), who concluded that winds blowing in the direction of a frontal jet reduce the PV of waters through friction. Vertical circulation driven by frontal meanders subsequently subducts these low-PV waters into the pycnocline, forming ITEs.

In addition to frontal convergence, recent studies have demonstrated that the interaction between the winter mixed layer and pre-existing subsurface eddy cores can significantly modulate the vertical structure of anticyclonic eddies. Yu et al. (2017) observed that repeated winter convection inside the Lofoten Basin Eddy progressively formed stacked, weakly stratified layers, leading to a long-lived, stepped core structure. Similarly, Barboni et al. (2023) reported that "connecting events", where winter mixed layers reach subsurface cores, lead to exceptional mixed-layer deepening and the formation of double-core eddies. These studies highlight the importance of internal stratification and restratification delay in shaping the longevity and thermohaline structure of ITEs, particularly in regions subject to strong seasonal forcing.

In the California and Peru-Chile upwelling systems, recent studies have explored the role of frictional forces exerted by continental slope topography on poleward undercurrents, leading to the formation of Slope Water Oceanic Eddies (Molemaker et al., 2015; Contreras et al., 2019). These regional observations support earlier theoretical work on subsurface eddy generation via bottom boundary-layer friction (D'Asaro, 1988), and further illustrate how topographically induced vorticity can contribute to ITE formation in diverse coastal environments. Locally induced mechanisms, such as the transformation of a surface AE into an ITE through eddy-wind interactions, have also been proposed. For example, Mcgillicuddy (2015) suggested that local Ekman suction within the AE can drive upwelling strong enough to dome the seasonal thermocline, converting the surface AE into an ITE.

Recent observational and modeling studies have also revealed additional mechanisms that may lead to the formation of double-core or vertically stacked intrathermocline eddies. Garreau et al. (2018) documented an anticyclonic eddy in the western Mediterranean combining a surface warm, salty core with a subsurface cooler, fresher lens. This structure was attributed to vertical alignment or stacking of distinct water masses, possibly inherited from separate formation events. Belkin et al. (2020) described a warm-core eddy formed by the vertical merger of two anticyclonic rings from the Gulf Stream. Their work synthesized multiple formation pathways for double-thermostad eddies, including vertical alignment, successive winter mixing, and isopycnal injections. These studies suggest that vertically complex ITEs may arise from the fusion of distinct vortices or from the successive layering of core waters formed during repeated winter convection. Such processes may operate in isolation or in combination, as observed in long-lived eddies like the Lofoten Basin Eddy (Bosse et al., 2019; Yu et al., 2017), further illustrating the diversity of pathways through which ITEs can emerge.

In the cCANUS, observational evidence of ITEs primarily pertains to Mediterranean Water lenses (meddies) (e.g., Armi and Zenk, 1984; Richardson et al., 2000; Machín and Pelegrí, 2016). Instabilities in the outflow of the Mediterranean Undercurrent generate these structures, and they are characterized by strong positive salinity and temperature anomalies with thicknesses of 500–1000 m, typically centered at intermediate depths (∼1000 m). Their lifespans range from one to three years and they can travel thousands of kilometers southwestward before their cores erode through lateral mixing or salt-fingering double diffusion processes (Ruddick and Hebert, 1988; Hebert et al., 1990; Schultz Tokos and Rossby, 1991), or abruptly collapse due to interactions with rough topography (Richardson et al., 1989).

Shallower ITEs, centered within the upper 500 m of the water column, have also been observed in cCANUS. Pingree (1996) provided the first detailed description of a long-lived AE centered at 190 m depth. This AE, with a diameter of ∼100 km and

a vertical extent of ∼150 m, was named *Flatty* due to its low aspect ratio. Nearly a year after its formation, Flatty had traveled approximately 1600 km from its origin in the CEC.

Caldeira et al. (2014) documented the formation of a wind-induced ITE in the lee of Madeira Islands in mid-summer 2011, based on remote sensing and in situ observations. The ITE was centered at ∼100 m depth and characterized by a 25 km radius, 100 m width, and a translational speed of $5 \, \text{km} \cdot \text{day}^{-1}$. More recently, Barceló-Llull et al. (2017) studied the anatomy of the ITE *PUMP*, which formed in the lee of Tenerife Island and exhibited a 46 km radius and a 500 m vertical extent, with two low-PV cores located at 85 m and 225 m depth, when surveyed four months after its formation and 550 km from its origin.

In November 2022, an AE originating in the lee of Gran Canaria Island in June 2022 (five months prior) was thoroughly sampled south of the Canary Archipelago. This eddy, named *Bentayga*, displayed the subsurface intensification typical of an ITE and shared broad similarities in formation, location, and timing with other documented eddies. Despite the similarities, it displayed remarkable hydrographic and dynamic differences, highlighting the complexity and variability inherent to these systems. Moreover, these differences emphasize the importance of cautious interpretation of findings to prevent overgeneralizing eddy dynamics to the entire population of ITEs within the CEC.

The current study integrates satellite observations with in situ measurements to investigate the formation and evolution of the Bentayga eddy in the CEC. The primary objective is twofold: (1) to explore the mechanisms responsible for the eddy's formation; and (2) to evaluate its influence on the physical properties of the regional ocean circulation. Additionally, the study compares the ITE with similar structures both within the CEC and in other regions of the world, offering new insights into the complexity and variability of mesoscale eddy activity.

The structure of this manuscript is as follows: Sect. 2 describes the survey, datasets, and methodology; Sect. 3 presents the results, including a Lagrangian analysis of the eddy's geometric properties and two- and three-dimensional analyses of circulation dynamics; Sect. 4 discusses the findings in comparison to other intrathermocline eddies; and concluding remarks are presented in Sect. 5.

## 2 Data and methods

### 2.1 Survey description

An interdisciplinary oceanographic survey was conducted aboard the R/V *Sarmiento de Gamboa* from 9 November to 4 December 2022, as part of the *Biogeochemical Impact of Mesoscale and Sub-mesoscale Processes Along the Life History of Cyclonic and Anticyclonic Eddies* (eIMPACT) project. This survey, called the eIMPACT2 survey, targeted a mature AE which originated southwest of Gran Canaria Island (GCI). The eddy has been named Bentayga, inspired by the Saharan-Berber heritage of the Canarian aborigines, reflecting its connection to Saharan upwelling waters off the northwestern African coast. At the time of the survey, it was approximately 4.5 months old and had traveled 560 km from its formation site (Fig. 1).

CTD-O (*Conductivity, Temperature, Depth – Oxygen*) data, collected using a towed undulating vehicle (SeaSoar Mk II; *Chelsea Technologies Group*) equipped with a *Sea-Bird* SBE911plus system and an adjoined SBE43 dissolved oxygen sensor, together with VMADCP (*Vessel-Mounted Acoustic Doppler Current Profiler*; *RDI* Ocean Surveyor, 75 kHz) records, were

gathered during the eIMPACT2 SeaSoar phase (9–14 November 2022) to construct three-dimensional hydrographic and dynamical fields of the eddy and its surroundings. This phase employed a southwest–northeast sampling grid comprising seven transects (278 km each), separated by ∼22 km, and sampled vertical levels between 30 and 340 m. A total of 955 CTD-O casts were recorded, with an average horizontal spacing of ∼2 km; continuous VMADCP measurements, providing horizontal velocity profiles from approximately 30 to 800 m depth with 16 m vertical resolution, were acquired concurrently along the same grid.

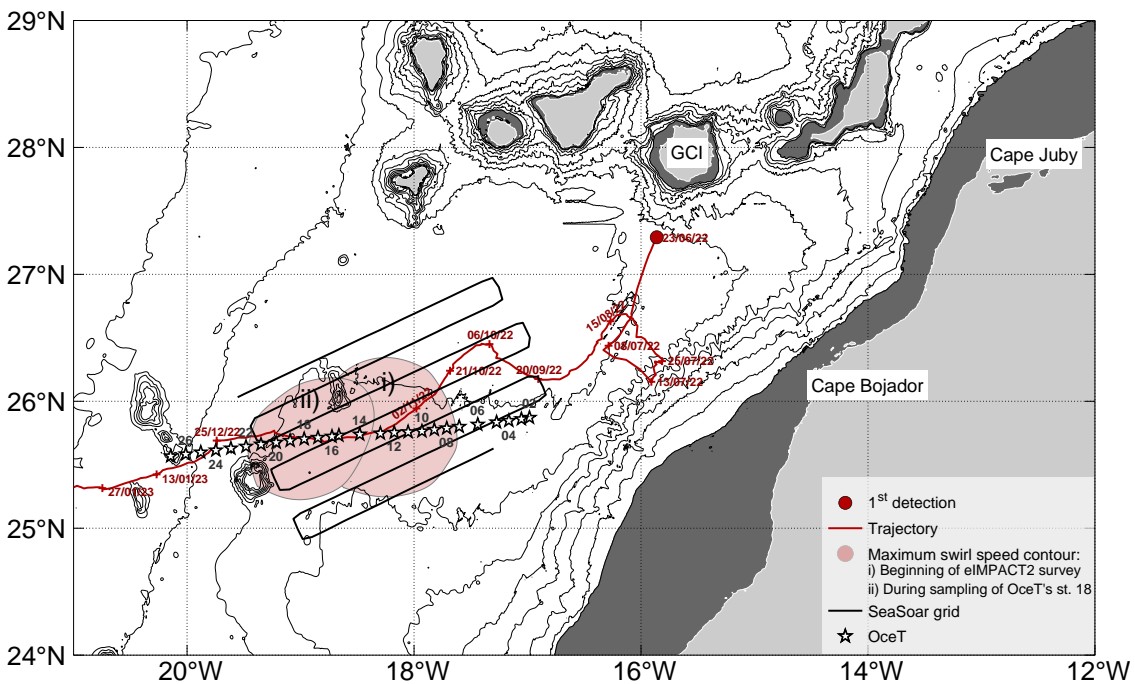

**Figure 1.** Map of the study area in the Canary Eddy Corridor (CEC) during the eIMPACT2 survey (9 November–4 December 2022). Land and the continental shelf (0 and 200 m depth) are shown in light and dark grey for the Canary Islands and the northwestern African coast, respectively. Bathymetric contours are drawn every 500 m from 500 to 5000 m depth (thin black lines). The thick black lines represents the ship's grid-like trajectory during the SeaSoar phase (9–14 November). White stars mark the locations of stations from the oceanographic transect (OceT, 19–28 November). The eddy is depicted by its first detection (dark red circle), its trajectory (thick dark red line with dates), and the areas enclosed by the contour of maximum swirl speed at two key dates: i) the start of the eIMPACT2 survey (9 November); and ii) during sampling at OceT station 18 (25 November), shown as light red patches.

In addition to the SeaSoar phase, the eIMPACT2 cruise included the OceT phase (19–28 November 2022), during which vertical CTD-O profiles were acquired using a *Sea-Bird* SBE911plus system equipped with a SBE43 dissolved oxygen sensor mounted on a rosette system, along a nearly zonal transect (OceT) crossing the eddy from east to west. This phase was selected

for the present study to investigate the mature stage of the Bentayga eddy. The OceT comprised 26 stations (stations 02–27), spaced $\sim$12.7 $\pm$ 2.8 km apart, and extended from the surface to $\sim$1500 m depth (Fig. 1). Alongside these profiles, continuous VMADCP velocity measurements spanning $\sim$30–800 m depth were recorded throughout the transect.

Given the duration of the two selected sampling phases—five days for the SeaSoar phase and ten days for the OceT—it was necessary to assess how this temporal spread might have affected or distorted our representation of the Bentayga eddy, particularly in analyses relying on its spatial delineation. To this end, we conducted a synopticity analysis, the results of which are presented in Supplementary Note 1. Intuitively, one might expect a quasi-synoptic sampling during the SeaSoar phase and a more asynchronous (non-synoptic) representation during OceT. Remarkably, the SeaSoar transects exhibited high synopticity, with an estimated eddy deformation of only $\sim$1%. By contrast, OceT introduced a distortion of approximately 12.6 $\pm$ 6.2%. However, sensitivity analyses of our eddy-core diagnostics and derived estimates (Supplementary Notes 4 and 5) indicate that the observed core deformation has a limited impact on the key results. These findings provide a certain degree of confidence in the analyses presented in the following sections.

Raw CTD-O data from both phases were processed at 1-meter intervals using the manufacturer's software, and TEOS-10 algorithms were applied to compute derived variables such as absolute salinity ($S_A$), conservative temperature ($\Theta$), potential density anomaly ($\sigma_\theta$), and brunt–väisälä frequency ($N$) (Feistel, 2003, 2008). VMADCP data were processed using the CODAS software (https://currents.soest.hawaii.edu/docs/adcp_doc/), resulting in 16-meter vertical bins for both phases (Martínez-Marrero et al., 2019). Dissolved oxygen ($O_2$) data collected during the OceT phase were calibrated against discrete water samples analyzed using the potentiometric Winkler method, following the protocol of Langdon (2010). The calibration yielded a linear relationship with a slope of 1.0957, an offset of $-1.0462$ $\mu$mol kg$^{-1}$, and a coefficient of determination $R^2 = 0.9908$. The precision of the Winkler titrations, expressed as the coefficient of variation, was 0.54% ($\pm$1.01 $\mu$mol kg$^{-1}$). Although no discrete $O_2$ samples were obtained during the SeaSoar phase due to its continuous sampling design, $O_2$ records from both phases were highly consistent, indicating stable and reliable sensor performance throughout the cruise.

To reduce noise in the OceT data, a two-dimensional Gaussian filter with horizontal and vertical scales of 12 km and 8 m, respectively, was applied to minimize the influence of unresolved short-scale fluctuations. These scales were estimated from the spatial autocorrelation of the noise field, reconstructed from high-order EOF modes (modes $\geq$ 4), using an integral length scale analysis. For the SeaSoar phase, CTD-O and VMADCP datasets were objectively interpolated onto a regular 11 $\times$ 11 km grid at 16-meter VMADCP levels. This process assumed planar mean CTD variables and constant mean VMADCP velocities (Rudnick and Luyten, 1996), employing a circular Gaussian covariance model with a 44 km scale (L$_x$=L$_y$), approximating the theoretical Rossby radius (Chelton et al., 1998), and incorporating 3% uncorrelated noise in the final fields (Bretherton et al., 1976, 1984; Le Traon, 1990).

## 2.2 Satellite information and derived products

The Atlas of Mesoscale Eddy Trajectories (META3.2exp), developed by SSALTO/DUACS and distributed by AVISO+ with the support of CNES, in collaboration with IMEDEA (Pegliasco et al., 2022) (https://www.aviso.altimetry.fr/), was used to track the geometric and dynamic properties of the Bentayga eddy throughout its life cycle, from formation to dissipation. The

use of the Near Real Time (NRT) product was necessary, as the Delayed Time (DT) version does not fully cover the period associated with the evolution of Bentayga. However, the META3.2 DT *allsat* product was used to provide a climatological reference for the eddy's trajectory, particularly to contextualize its interaction with filaments from the northwestern African upwelling system. The atlas is based on the global detection and tracking of mesoscale eddies using altimetry data from multiple platforms, applying the eddy detection method of Mason et al. (2014) to provide daily information on eddy location, speed, radius, and other characteristics, and classifying them by gyre type (CE and AE). This information was complemented with sea level anomaly and geostrophic current fields obtained from the European Seas Gridded L4 Sea Surface Height and Derived Variables NRT altimetry product (SEALEVEL_EUR_PHY_L4_NRT_008_060), processed by the DUACS multi-platform altimetry system and distributed by the Copernicus Marine Service (https://marine.copernicus.eu).

To characterize the interaction between the circulation driven by this eddy and the upwelling fronts and filaments off the northwest African coast, we used daily fields of Sea Surface Temperature (SST) from the GHRSST Level 4 MUR Global Foundation Sea Surface Temperature Analysis product. This dataset, with a resolution of 1 km $\times$ 1 km, is distributed by PO.DAAC (https://doi.org/10.5067/GHGMR-4FJ04). In addition, satellite chlorophyll-a data, with a resolution of 4 km $\times$ 4 km, were obtained from the Global Ocean Colour (Copernicus-GlobColour) Bio-Geo-Chemical Level 3 product distributed by the E.U. Copernicus Marine Service Information (https://doi.org/10.48670/moi-00280). Both of these products are derived from a combination of satellite information from all available platforms, utilizing specific processing, analysis, and merging schemes (Chin et al., 2017; Garnesson et al., 2019).

## 2.3 Dynamical properties

### 2.3.1 Potential vorticity

As mentioned in Sect. 1, a key signature of ITEs is their core of low PV, resulting from the nearly homogeneous waters they contain. Due to the Bentayga eddy's small aspect ratio, defined by a large horizontal extent relative to its vertical thickness, the following form of Ertel's PV (Müller, 1995) was employed:

$$PV = -\frac{(-\partial_z v, \partial_z u) \cdot \nabla_h \sigma_\theta}{\rho_0} - \frac{(\zeta + f)\partial_z \sigma_\theta}{\rho_0} \tag{1}$$

This expression includes two sources of PV: the first term on the right-hand side represents PV associated with isopycnal tilting, while the second term corresponds to PV due to water column stretching. In the first term, $(-\partial_z v, \partial_z u)$ is the rotated vertical shear vector of the horizontal velocity field, given by $\mathbf{u}_h(x,y,z) = u(x,y,z)\hat{\mathbf{i}} + v(x,y,z)\hat{\mathbf{j}}$, where $\mathbf{u}_h$ denotes the horizontal velocity vector, and $\nabla_h \sigma_\theta$ represents the horizontal gradient of the potential density anomaly ($\sigma_\theta$). In the second term, $(\zeta + f)$ is the absolute vorticity, defined as the sum of relative vorticity ($\zeta$) and planetary vorticity ($f$), while $\partial_z \sigma_\theta$ represents the vertical gradient of $\sigma_\theta$. For both terms, a mean background density $\rho_0 = 1026$ kg m$^{-3}$ was used. The calculations were based on the objectively interpolated fields.

### 2.3.2 Energy metrics

The total Available Potential Energy (APE) and Kinetic Energy (KE) within the eddy were estimated using volume integrals based on the approach of Schultz Tokos and Rossby (1991), applied to the hydrographic and current records collected along the OceT transect. For both integral calculations, the eddy's volume was assumed to be cylindrical and treated as an isolated structure within an infinitely wide ocean basin (Hebert, 1988), where unperturbed hydrographic fields explicitly define the reference ocean state. The estimation of these reference fields is described further in Sect. 3.2. The integrals were performed over the water column, from the surface ($z = 0$) to the trapping depth ($z = $H) (e.g., Dilmahamod et al., 2018; Morris et al., 2019), and radially, from its center ($r = 0$) to its core radius ($r = $R$= $R$_c$; see Sects. 3.2 and 3.5). The total APE was calculated as follows:

$$APE = 0.5\rho_r \int\limits_{H}^{0} \int\limits_{0}^{R} N_r(r,z)^2 \delta(r,z)^2 (2\pi r) dr dz \tag{2}$$

where $\rho_r$ is the density of the reference state, $N_r(r,z)$ is the Brunt-Väisälä frequency of the reference state, $\delta(z,r)$ represents the vertical departures of the isopycnal surfaces from the reference state, and $2\pi r dr$ is the infinitesimal area element. The total KE within the eddy was calculated as:

$$KE = 0.5\rho_r \int\limits_{H}^{0} \int\limits_{0}^{R} |\mathbf{u}_h(r,z)|^2 (2\pi r) dr dz \tag{3}$$

where $|\mathbf{u}_h(r,z)|^2 = u(r,z)^2 + v(r,z)^2$ is the squared horizontal velocity magnitude, with $u(r,z)$ and $v(r,z)$ representing the zonal and meridional velocity components obtained from the VMADCP.

### 2.3.3 Thermohaline anomalies

To evaluate the thermohaline perturbations induced by the eddy, the Available Heat Anomaly (AHA) and Available Salt Anomaly (ASA) were calculated following the methodology of Chaigneau et al. (2011). These volume integrals were computed under the same assumptions described earlier. The AHA and ASA were calculated as:

$$AHA = \int\limits_{H}^{0} \int\limits_{0}^{R_C} \rho(r,z) C_p \Theta'(r,z) (2\pi r) dr dz \tag{4}$$

$$ASA = 0.001 \int\limits_{H}^{0} \int\limits_{0}^{R_C} \rho(r,z) S_A{}'(r,z) (2\pi r) dr dz \tag{5}$$

In both equations, $\rho$ represents the potential density. In Eq. 4, $C_p$ is the specific heat capacity, set to 4000 J kg$^{-1}$ K$^{-1}$, while in Eq. 5, the factor 0.001 converts salinity into a fraction of kilograms of salt per kilograms of seawater. The conservative temperature ($\Theta'$) and absolute salinity ($S_A{}'$) anomalies were defined as departures from the same unperturbed reference ocean state used in the calculations of APE and KE (Eq. 2 and Eq. 3).

### 2.3.4 Eddy-driven fluxes

The volume, heat, and salt transported by the eddy were assessed by estimating the corresponding horizontal eddy-driven fluxes (Dilmahamod et al., 2018). Following the approach of Morris et al. (2019), the eddy was treated as a bulk entity, assuming a vertically uniform translational speed ($c$), calculated as the average propagation speed during the eIMPACT2 survey. In this formulation, the horizontal advection of water properties induced by the eddy occurs through its frontal cross-section as it propagates. The effective width of this vertical face is approximated by $2r$, where $r$ is the radial distance from the eddy center to the outer limit of its core. The volume ($V_e$), heat ($Q_{eh}$), salt ($Q_{es}$), and freshwater ($Q_{fw}$) fluxes were calculated as follows:

$$V_e = c \int_{H}^{0} (2r)dz \tag{6}$$

$$Q_{eh} = c \int_{H}^{0} \rho_0 C_p \Theta'(z)(2r)dz \tag{7}$$

$$Q_{es} = 0.001c \int_{H}^{0} \rho_0 S_A{'}(z)(2r)dz \tag{8}$$

$$Q_{fw} = \frac{-Q_{es}}{\rho_0 S_0} \tag{9}$$

In these equations: $V_e$ represents the eddy-driven volume transport in Sverdrups (1 Sv = $10^6$ m$^3$ s$^{-1}$), $Q_{eh}$ is the eddy-driven heat flux in Watts (W), $Q_{es}$ is the eddy-driven salt flux in kilograms per second (kg s$^{-1}$), and $Q_{fw}$ is the equivalent freshwater flux in Sv. Here, $\rho_0 = 1026$ kg m$^{-3}$ is the mean background density as in Eq. 1, $C_p = 4000$ J kg$^{-1}$ K$^{-1}$ is the specific heat capacity as in Eq. 4, and $S_0 = 35$ (dimensionless, representing salt mass fraction in psu) is the mean upper-ocean salinity. Conservative temperature and absolute salinity anomalies ($\Theta'(z)$ and $S_A{'}(z)$, respectively) were calculated as departures from the unperturbed reference state.

## 3 Results

### 3.1 Lagrangian evolution as seen from satellite altimetry data

The Bentayga eddy formed south of GCI in June 2022, with its first detection recorded on 23 June (Figs. 1–3). Shortly after formation, during its growth phase, it drifted away from GCI almost immediately (see trajectory and respective dates in Figs. 1 and 2). The initial 20 days were characterized by a rapid southwestward movement, which later shifted to an almost eastward direction, with speeds $\geq 10$ km d$^{-1}$, covering approximately 170 km from the origin to the continental margin of NW Africa,

near Cape Bojador ($\sim$26.1°N). During this phase, significant growth was observed only in its radius defined by the contour
of maximum circum-averaged speed (Fig. 3c). It remained in this region for over two weeks, during which it interacted with
upwelling filaments extending from Cape Juby to the north and Cape Bojador to the south (Fig. 2b). Throughout this period,
the sea level anomaly (SLA) amplitude and swirl speed increased considerably, rising by approximately 4 cm and 15 cm
s$^{-1}$, respectively (Figs. 3a, b). This behavior gave a strong non-linear character to the Bentayga eddy, as indicated by $\vartheta > 5$
(Fig. 3e), a feature that persisted throughout the analyzed period.

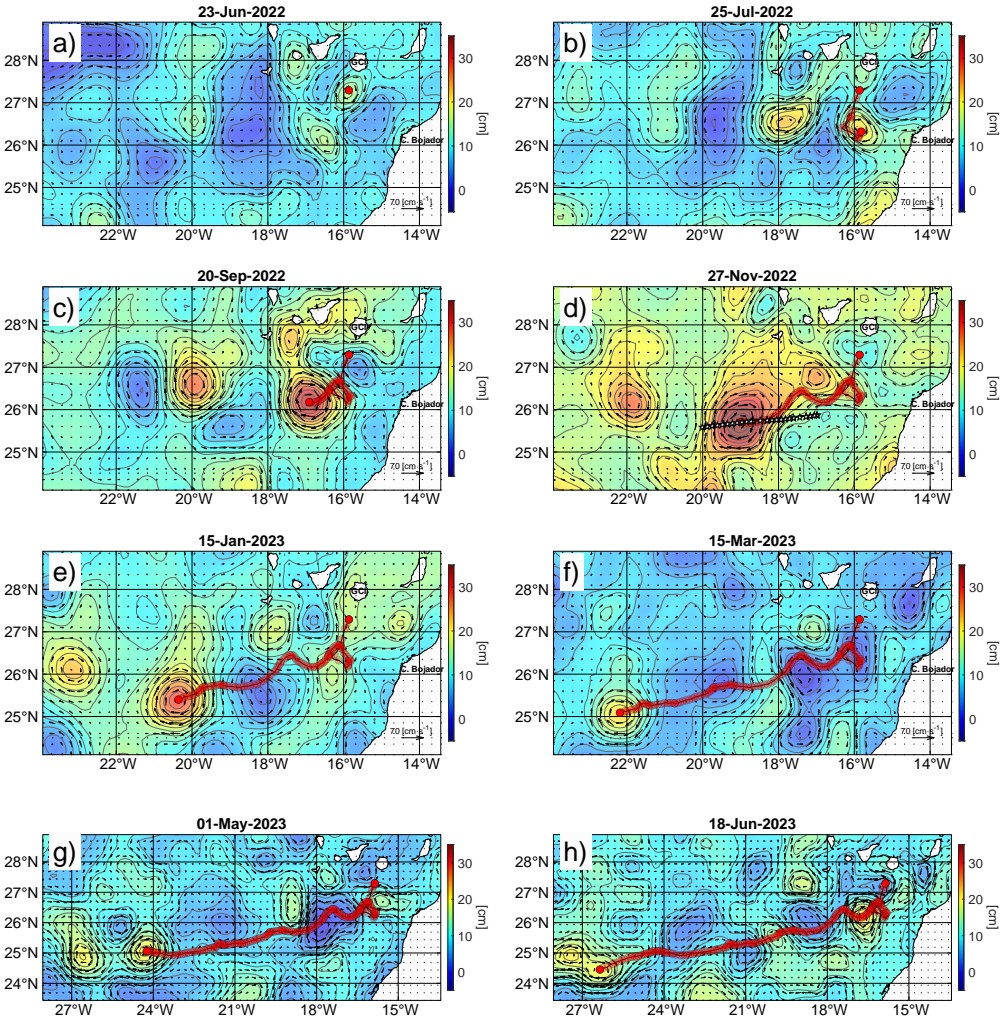

**Figure 2.** Sea level anomaly snapshots (color map and thin grey contours) and geostrophic currents (black arrows) derived from satellite
altimetry. The panels show the eddy's evolution from its first detection on 23 June 2022, through its various phases, including the eIMPACT2
survey, to its dissipation on 18 June 2023 (trajectory indicated by red circles). White stars denote oceanographic station locations during the
eIMPACT2 OceT phase.

By the end of July, it began to move northwestward into the open ocean, maintaining this trajectory while continuing to exhibit gradual increases in the aforementioned features until 15 August (Figs. 1 and 3). At this stage, peak values were reached in amplitude ($\sim$20 cm), swirl speed ($\sim$55 cm s$^{-1}$), and radius ($\sim$55 km), with Rossby number (Ro) values of $\sim$0.16 (Figs. 3a–d). These relatively higher values were sustained until the early days of November (Figs. 3a–c), when the temporal evolution of these features displayed a plateau behavior, particularly evident in the SLA amplitude and swirl speed, even after

accounting for the 15–30-d fluctuations observed throughout its life cycle (Figs. 3a, b). This plateau indicates that the Bentayga eddy had entered its mature phase by this time, with mean values of approximately $\sim$18$\pm$2 cm for SLA amplitude, $\sim$45$\pm$4 cm s$^{-1}$ for swirl speed, and $\sim$50$\pm$5 km for radius. The temporal evolution of its Ro values and non-linear character also evolved into a plateau during this phase (Figs. 3d, e); however, for Ro, it persisted only until mid-October, with a mean magnitude of approximately 0.15. Throughout this phase, it primarily followed a southwestward trajectory (Figs. 1 and 2c, d), although a

northwestward deflection was observed in its path between mid-September and early October (Fig. 1).

    Starting in the first weeks of November, a gradual decline in all the previously described properties became evident (Fig. 3) indicating that the eIMPACT2 survey occurred at the onset of this slow decline in the dynamic properties during the eddy's mature phase. Such a smooth and gentle decrease in the magnitude of the eddy's dynamic properties during this phase has been previously observed in the CANUS and other EBUS (Pegliasco et al., 2015). During the eIMPACT2 survey, the Bentayga

eddy moved with a translational speed ($c$) of $4.5 \pm 1.2$ km d$^{-1}$ and displayed a mean SLA amplitude of $15 \pm 1$ cm, swirl speed of $38 \pm 2$ cm s$^{-1}$, radius of $55 \pm 3$ km, mean Ro values of 0.11, and a strong non-linear character ($\vartheta > 5$). A few days after the campaign concluded, the nearly linear decline in these properties was interrupted by shorter-period fluctuations (with a temporal scale of 10–20 days), particularly noticeable in the SLA amplitude and radius. These fluctuations persisted until the end of its mature phase in mid-May 2023 (Figs. 3a, c). The onset of these shorter-scale fluctuations coincided with a slight

northwestward deflection observed in the eddy's trajectory shortly after the end of the eIMPACT survey (6 December, Fig. 1). Finally, from late May 2023 onward, the dynamical properties of the Bentayga eddy exhibited an abrupt decrease, marking its transition into the decay phase, which culminated in its dissipation on 18 June 2023.

    The temporal evolution of the eddy properties was dominated by two main modes of variability (Fig. 3). The first mode was a long-term component, spanning 7–11 months, associated with its overall life cycle: growth (mid-June 2022 to mid-August

2022), maturity (including the slow decline stage; mid-August 2022 to mid-May 2023), and rapid decay (starting in late May 2023) (Fig. 3). The second mode of variability involved shorter-scale fluctuations, ranging from weeks to a few months, as identified through wavelet analysis (see Fig. A1). Generally, these fluctuations displayed a degree of synchronicity among the analyzed properties, reflecting the dominance of geostrophic characteristics even at these timescales, as increases in SLA amplitude were systematically accompanied by increases in swirl speed. This relationship indicates that the eddy's velocity

field remained primarily controlled by the pressure gradient force balanced by the Coriolis force. However, the relative energy contributions of these fluctuations differed: the SLA amplitude and eddy radius exhibited larger variations than those observed in the eddy's swirl speed variability, suggesting that shorter-scale fluctuations more strongly modulated the eddy's geometry and surface expression than its velocity field. This behavior could be interpreted as a mode of eddy pulsation, similar to that described by Sangrà et al. (2005, 2009), suggesting a connection with changes in Bentayga's elliptical eccentricity and the

presence of sharp meanders at its edges. These phenomena are likely due to eddy-to-eddy interactions (Sangrà et al., 2005; Morris et al., 2019) or some form of submesoscale instability (Brannigan et al., 2017; Thomas et al., 2013).

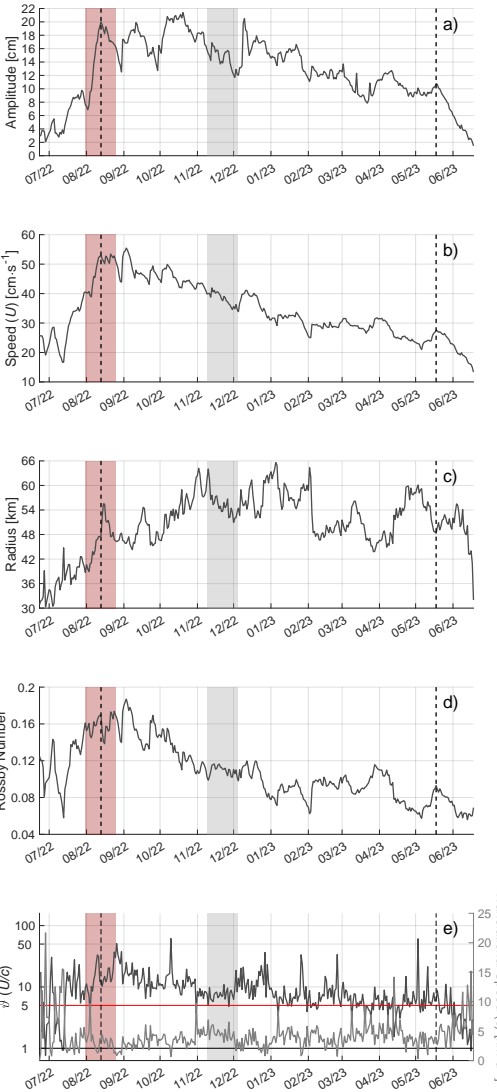

**Figure 3.** Time series of geometric and dynamic properties of the eddy from its first detection (23 June 2022) to its dissipation (18 June 2023). Panels show: a) amplitude, b) swirl speed, c) radius, d) Rossby number (calculated as the ratio between the swirl speed and radius, normalized by $f$), and e) the non-linearity parameter ($\vartheta$, estimated as the ratio of swirl to translational speeds). In panel e, the translational speed is indicated by the grey line, while black and red horizontal lines represent thresholds for non-linearity ($\vartheta > 1$) and strong non-linearity ($\vartheta > 5$) (Chelton et al., 2011). Pale red and grey shaded areas across all panels highlight periods when the eddy was near the NW African continental margin and during the eIMPACT2 survey, respectively. Vertical dashed lines indicate the onset and end of its mature phase.

## 3.2 Circulation and hydrography

The following section describes the circulation and hydrography along the OceT transect. As previously noted, this transect reflects a distorted view of the eddy structure due to the lack of synopticity; however, the general features of its core remain identifiable (see Supplementary Notes 2 and 3). The rationale for using this dataset lies in its vertical extent, which enables a comprehensive view of the water column down to intermediate depths, effectively capturing the full vertical structure of the Bentayga eddy (see Supplementary Notes 4 and 5).

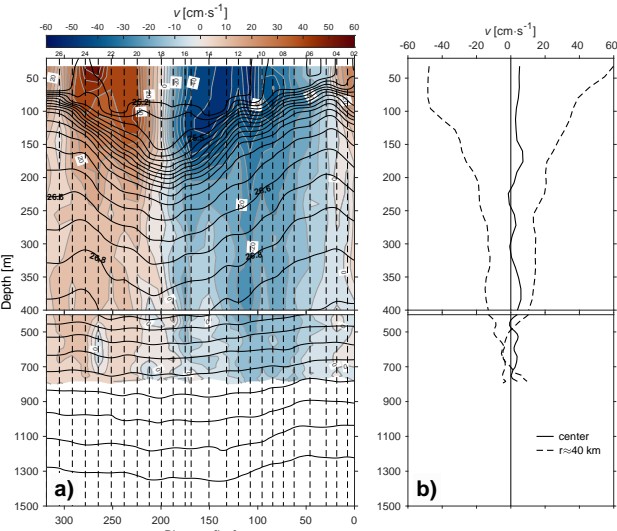

**Figure 4.** a) Vertical section of cross-transect velocities ($v$) from VMADCP measurements along the OceT transect. Grey contours represent velocity levels at 5 cm·s$^{-1}$ intervals, while black contours show isopycnals ($\sigma_\theta$) ranging from 25.2 to 27.7 kg m$^{-3}$ with 0.1 kg m$^{-3}$ intervals. Vertical dashed lines indicate the positions of oceanographic stations, with only even station numbers labeled for clarity. b) Vertical profiles of cross-transect velocities at the eddy's center (station 18), and at approximately the radial distance of maximum velocity ($r \approx 40$ km, refer to text for further details). Different vertical scales are applied for the 0–400 m and 400–1500 m depth ranges in both panels.

Cross-transect velocities along the OceT (Fig. 4a) were primarily influenced by the anticyclonic circulation induced by the Bentayga eddy (stations 12–24, 120–280 km from OceT's origin). Stronger velocities (>30 cm s$^{-1}$) extended from the surface to the $\sigma_\theta = 25.5$ kg m$^{-3}$ isopycnal depth, closely following its concave shape. Between stations 15–20 (175–225 km from OceT's origin), higher cross-transect velocities were observed at approximately 110–120 m depth (>45 cm s$^{-1}$). However, at station 18, velocities were nearly zero throughout the entire depth profile, indicating proximity to its center (Figs. 4a, b). Starting from stations 15 and 20 towards the east and west, respectively, the vertical segment of higher velocities became slightly shallower, with stations 12 and 24 roughly marking the eastern and western boundaries of the eddy exerted effect. This variation in the vertical position of the maximum cross-transect velocity layer suggests the presence of an inner core within, with a radius ($R_c$) of ~23-25 km and an external surrounding ring approximately 45 km wide. This segmentation is consistent

with the radial zonation presented in Supplementary Note 3, where maximum azimuthal velocities are reached in the outer sector of the eddy at approximately 40 km from its center, gradually decreasing outward.

The surrounding ring spans roughly between stations 20 and 24 to the west, and stations 12 and 15 to the east—corresponding to distances of up to 70 km from the eddy center—with its lower (deeper) boundaries closely following the spatial behavior of the aforementioned isopycnal (Fig. 4a). Vertical profiles of cross-transect velocity (Fig.4b) reveal a slight horizontal asymmetry, with stronger velocities occurring at deeper depths within the inner core, and at shallower depths in the ring area (Figs.4a, b). Although a closer examination of this sector indicates that it lacks spatial homogeneity (Fig. 4), the deformation noted at the beginning of this section and the inherently higher variability in this area make it prone to misinterpretation. As such, a more detailed analysis of this sector is not pursued in the present study. Nevertheless, its presence adds complexity to the overall azimuthal velocity distribution, highlighting radial variations that could impact the stability and mixing properties of the Bentayga eddy (McWilliams, 1988, 1985). These circulation patterns remained spatially coherent above the $\sigma_\theta = 26.6$ kg m$^{-3}$ isopycnal. However, at greater potential density anomalies (26.7–27.4 kg m$^{-3}$), the anticyclonic circulation weakened (<15 cm s$^{-1}$) and became more spatially disordered. Outside the observed horizontal boundaries of the Bentayga eddy, cross-transect velocities aligned with the external cyclonic circulation on either side (Fig. 2d).

From Fig. 5c, the deepening of the isopycnal surfaces is evident for $\sigma_\theta \geq 25.4$ kg m$^{-3}$ between stations 12 and 24, with the seasonal pycnocline (associated with $N$ values $\geq 5$ cph) exhibiting a concave shape along this horizontal segment. In contrast, lighter isopycnals displayed a slight convex shape between stations 15 and 20 (175–225 km from OceT's eastern origin) along the $\sigma_\theta = 25.2$ kg m$^{-3}$ isopycnal. The presence of a deep surface mixed layer (extending from the surface to a depth of 80 m), containing even lighter waters (<25.2 kg m$^{-3}$), along with a shallower and less intense pycnocline (with $N$ values of 3–4 cph), likely prevented doming of these lighter isopycnals. The core of the Bentayga eddy was centered between these shallower and deeper pycnoclines (~110 m depth), enclosing nearly homogeneous water and forming a pycnostad with a mean potential density anomaly of 25.3±0.05 kg m$^{-3}$ ($N \leq 2.5$ cph). As expected, outside the influence of the eddy (westward from station 24 and eastward from station 12), the density field was influenced by the cyclonic circulation on both sides, with a mesoscale elevation of water (isopycnal lifting) noticeable even at depths around 500 m (Fig. 5c).

Most of the structures for the distribution of $\sigma_\theta$ along the OceT described above are also observed in the first 600 m of the vertical sections of $\Theta$ and $S_A$ (Fig. 5a, b). Between stations 12 and 24, the 12.5–22°C isotherms and 36–36.9 g kg$^{-1}$ isohalines exhibited the previously mentioned deepening. Signs of minor elevation in the horizontal segment between stations 15 and 20 were also noticeable along the 23°C isotherm and the 36.9 g kg$^{-1}$ isohaline, as well as a core of nearly constant temperature and salinity, with mean values of 22.5±0.2°C and 36.83±0.01 g kg$^{-1}$. This well-mixed core was vertically bounded by two sharp positive thermoclines (~0.1°C m$^{-1}$), a sharp positive upper halocline (0.02 g kg$^{-1}$ m$^{-1}$), and a weaker negative lower halocline (-0.01 g kg$^{-1}$ m$^{-1}$). In the $S_A$ field, the structure of the eddy core was more clearly defined by the 36.8 g kg$^{-1}$ isohaline, forming a lens-like shape of relatively fresher water compared to the surrounding waters at the same vertical levels. This high degree of homogeneity is also discussed in Supplementary Notes 2 and 3 (see Supplementary Figs. 4-8), and matches well with the radial distance between the OceT data and the profiles used in those analyses.

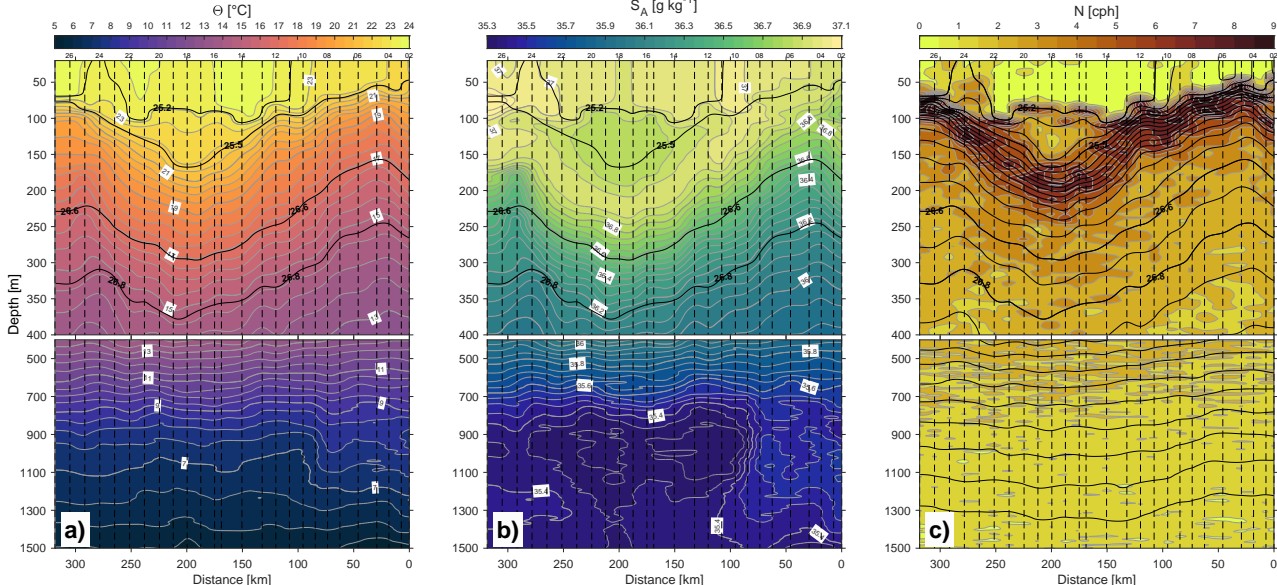

**Figure 5.** Vertical sections along the OceT of: a) Conservative temperature ($\Theta$), b) absolute salinity ($S_A$), and c) brunt-väisälä frequency ($N$). In panels (a) and (b), thick black contours correspond to $\sigma_\theta$ levels of 25.2, 25.5, 26.6, and 26.8 kg m$^{-3}$. In panel (c), black contours represent isopycnals ($\sigma_\theta$) ranging from 25.2 to 27.7 kg m$^{-3}$, with a contour interval of 0.1 kg m$^{-3}$. Vertical dashed lines indicate the positions of oceanographic stations as shown in Fig. 4a. Consistent with Fig. 4, different vertical scales were applied for the 0–400 m and 400–1500 m depth ranges.

It is important to note that, aside from the mesoscale perturbations in the thermohaline properties along the OceT, their overall hydrographic distribution corresponded to the expected water mass composition for the region, with a dominance of 12° and 15°C Eastern North Atlantic Central Waters (ENACW12 and ENACW15, respectively) (Fig. 6). At intermediate depths (700–1300 m), relatively colder and fresher waters were observed from station 7 westward (approximately 60 km from OceT's

origin) (Fig. 5a, b). These waters were consistent with the thermohaline values and the higher contribution of the diluted Antarctic Intermediate Water (AAIW) previously described in this region at these depths (Alvarez et al., 2005; Bashmachnikov et al., 2015; Jiménez-Rincón et al., 2023). In this study, the acronym AA, from the nomenclature proposed by Alvarez et al. (2005), will be used to refer to this water mass, as its thermohaline properties align with those observed locally in this region (Fig. 6), albeit with slight variations from the classical definition of AAIW (Tomczak and Godfrey, 2004).

No disturbances in the $\sigma_\theta$ field were observed at these intermediate depths despite the mentioned variability, indicating that density remained compensated (Fig. 5c). This compensation may also explain the localized enrichment of thermohaline fine structure, likely due to lateral intrusions, which were more noticeable in the $S_A$ field (Fig. 5b). Additionally, within the area thought to be influenced by the Bentayga eddy (stations 12–24), the relative contribution of AA appeared stronger and was accompanied by a higher presence of Sub-Polar Mode Water (SPMW) above it (Fig. 5a, b; Fig. 6). This suggests a possible

connection between the anticyclonic circulation driven by the eddy and the vertical extent of its trapping capacity (i.e., its

non-linearity ratio, $\vartheta$, see Fig. 3e). However, as explained in Sect. 2.1, reliable VMADCP velocity records were not obtained for these depths, preventing further analyses.

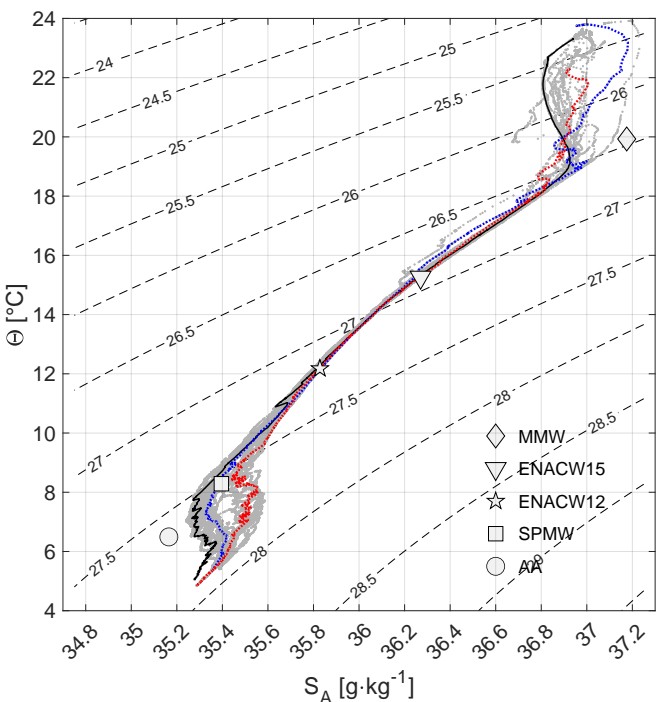

**Figure 6.** Conservative temperature ($\Theta$) versus absolute salinity ($S_A$) distribution along the OceT transect, presented as a $\Theta - S_A$ diagram. Grey dots represent data from all oceanographic stations. The thermohaline structure at the eddy center (station 18) is highlighted with a thick black continuous line. Thermohaline properties from regions outside eddy's influence, westward (station 26) and eastward (station 04), are depicted with thick blue and red dashed lines, respectively. Symbols mark typical source water masses for the area (Alvarez et al., 2005), including Madeira Mode Water (MMW), Eastern North Atlantic Central Water (ENACW15 and ENACW12, at 15°C and 12°C, respectively), Subpolar Mode Water (SPMW), and diluted Antarctic Intermediate Water (AA). Thin black dashed lines denote $\sigma_\theta$ levels from 24 to 29 kg m$^{-3}$ at intervals of 0.5 kg m$^{-3}$.

To further investigate the impact of the Bentayga eddy on the water column along the OceT, anomalies in $\sigma_\theta$, $\Theta$, and $S_A$ were analyzed (Fig. 7). As noted earlier, it was surrounded by several cyclonic mesoscale eddies, which caused perturbations outside its horizontal boundaries, affecting the water column from stations 02–11 and 25–27 (Fig. 2d). To account for these external influences, the mean profiles used as a reference state for anomaly calculations were derived using Djurfeldt (1989) method. First, an average distribution was calculated as a function of $\sigma_\theta$ for depth, $z$, $\Theta$, and $S_A$ (i.e., $\overline{z}(\sigma_\theta)$, $\overline{\Theta}(\sigma_\theta)$, and $\overline{S_A}(\sigma_\theta)$). These averaged profiles were then interpolated back to the original $z$-coordinate and subtracted from the respective vertical sections.

In terms of potential density anomaly and conservative temperature, the Bentayga eddy displayed a single core of lighter and warmer waters, extending from ∼80 m to 500 m depth ($<-0.2$ kg m$^{-3}$ and $>1°$C, respectively), horizontally spanning stations 12–24 (Fig. 7a, c), with the strongest anomalies located near the $\sigma_\theta = 25.5$ kg m$^{-3}$ isopycnal at ∼150–155 m depth. At this level, density and temperature anomalies reached about $-1$ kg m$^{-3}$ and 4°C, respectively. In contrast, the absolute salinity anomaly field exhibited a double-core structure: a shallower core of lower salinity between 80 and 140 m depth, enclosed between the $\sigma_\theta = 25.2$ and 25.5 kg m$^{-3}$ isopycnals (anomalies below $-0.05$ g kg$^{-1}$), and a deeper saltier core extending from ∼150 to 550 m depth, with maximum positive anomalies ($> 0.1$ g kg$^{-1}$) centered between 200 and 300 m depth (Fig. 7b). At intermediate depths (800–1300 m), colder and fresher anomalies were observed just below the core of positive thermohaline anomalies (Fig. 7a, b). These anomalies reflected the influence of SPMW and AA water masses (Fig. 6), with temperature anomalies below $-0.25°$C and salinity anomalies below $-0.1$ g kg$^{-1}$.

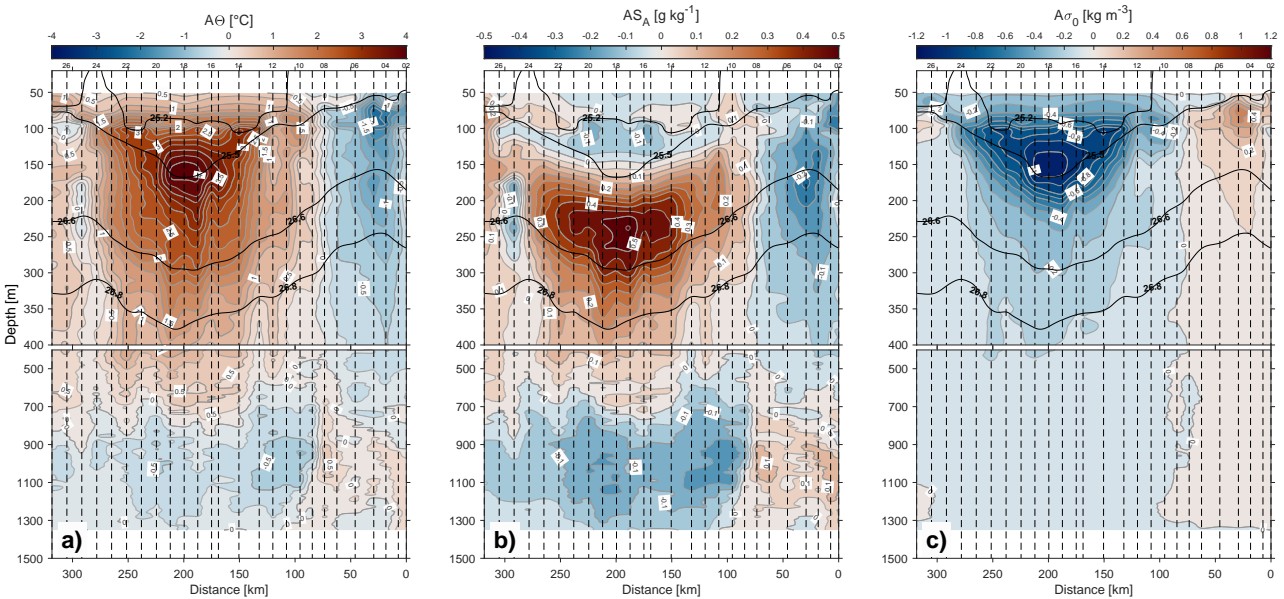

**Figure 7.** Vertical sections of anomalies in: a) conservative temperature (A$\Theta$), b) absolute salinity (A$S_A$), and c) potential density anomaly (A$\sigma_\theta$) along the OceT transect during the eIMPACT2 survey. Thick black contours in all panels denote $\sigma_\theta$ levels at 25.2, 25.5, 26.6, and 26.8 kg m$^{-3}$. Vertical dashed lines indicate the positions of oceanographic stations as shown in Fig. 4a. Consistent with Fig. 4, distinct vertical scales are used for the 0–400 m and 400–1500 m depth ranges.

### 3.3  3D view of the horizontal velocity field

The objectively interpolated fields of VMADCP horizontal velocity ($u$, $v$) and potential density anomaly ($\sigma_\theta$) (Fig. 8) clearly reveal the dominant influence of the Bentayga eddy in the sampled area, with its anticyclonic circulation prevailing across much of the region and remaining evident even at the shallower recorded depths ($<60$ m, Fig. 8a). Outside this prominent

circulation, the northwestern and southeastern corners of the interpolated field exhibited CEs with flow magnitudes comparable
to those of this anticyclonic system. Additionally, a smaller AE was identified northeast of the primary system (Fig. 2d). This
secondary structure, approximately half its size, displayed weaker anticyclonic circulation. However, since it was located near
the northeastern edge of the SeaSoar sampling grid, its geometry was likely not well resolved.

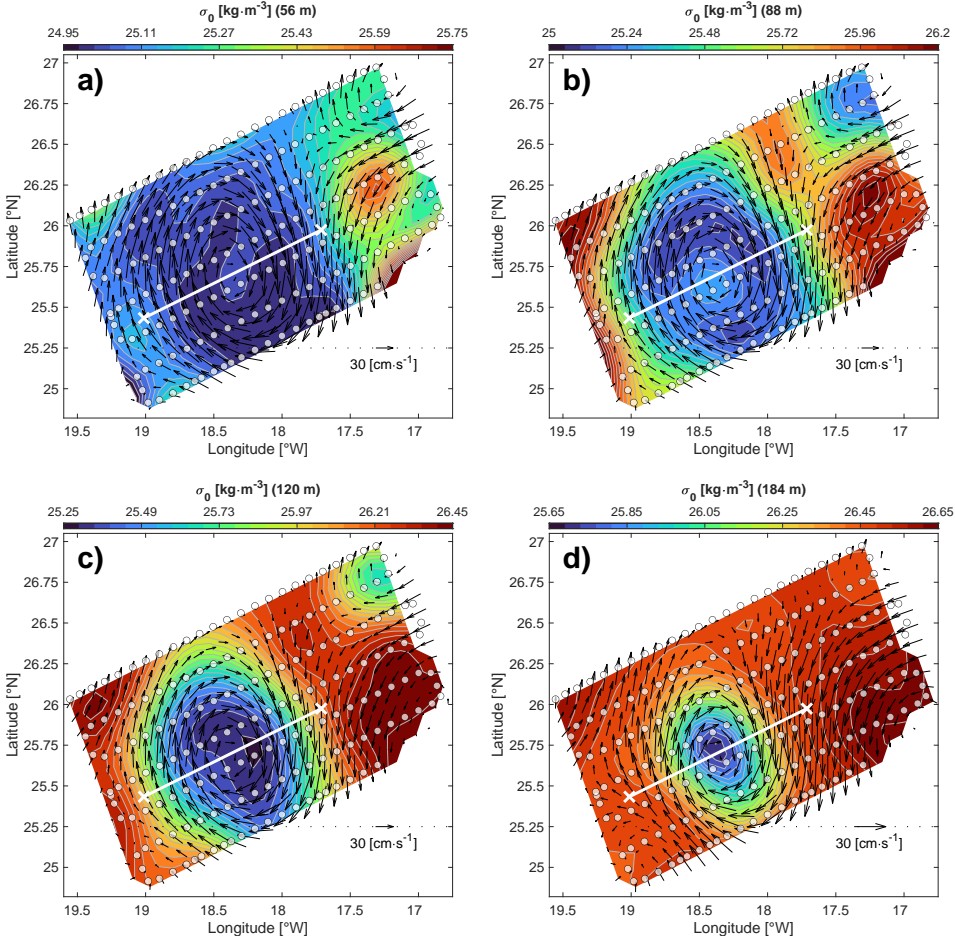

**Figure 8.** Objectively interpolated VMADCP velocity vectors (black arrows) superimposed on potential density anomaly ($\sigma_\theta$) fields at
different depths: (a) 56 m, (b) 88 m, (c) 120 m, and (d) 184 m. Each panel uses a distinct color scale representing $\sigma_\theta$ values, with 25 contour
levels uniformly spaced across the respective density range to highlight the isopycnal structure. Velocity vectors are scaled relative to the
magnitude at each depth (see legend in the bottom right corner of each panel). The ship's grid-like sampling trajectory during the eIMPACT2
Seasoar phase (pale white circles) and the virtual transect used for the vertical section in Fig. 9 (white black line) are also indicated. Objective
interpolation correlation scales were set to $L_x = L_y = 44$ km, with 3% uncorrelated noise applied.

Although the anticyclonic circulation induced by the Bentayga eddy was well-defined from depths as shallow as 30–60 m
(see Figs. 4 and 8), the expected doming of the upper isopycnals bounding the core was not clearly observed. Instead, it

appeared only subtly and in a spatially localized manner, as seen in the horizontal sections of the upper 100 m (Fig. 8a, b), and consistent with the structure depicted by the 25.2 kg m$^{-3}$ isopycnal along the OceT transect (Fig. 5). This localized doming of the upper isopycnals within the eddy may, in turn, reflect the advection of relatively lighter water from the surrounding ring, circulating around—and possibly penetrating into—the core. Moreover, the presence of this less dense "external" water may have locally intensified the vertical density contrast in that sector of the eddy, potentially hindering or even preventing the

upward displacement of the isopycnals native to that region. At these depths (<100 m depth), the eddy exhibited a generally circular, though slightly irregular, shape (horizontal aspect ratio of nearly 1.0), with meandering horizontal boundaries. With increasing depth, the structure became more regular and elliptical, maintaining a nearly constant horizontal aspect ratio of approximately 0.8 (Fig. 8c, d). Notably, the horizontal section at 120 m depth indicates that lighter water circulating around the core extended even to those depths (Fig. 8c). Below 120 m, the Bentayga eddy was characterized primarily by local isopycnal

deepening and exhibited a reduction in size with increasing depth (Figs. 8d, B1 and B2).

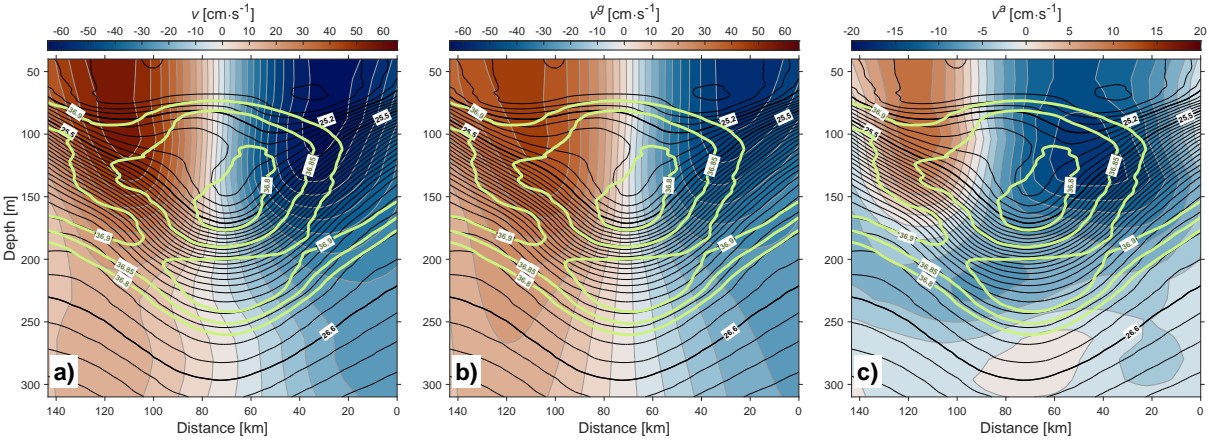

**Figure 9.** Objectively interpolated cross-transect velocity components along the virtual transect shown in Fig. 8. Panels represent: a) cross-transect velocities measured directly by the VMADCP ($v$), b) geostrophic velocities estimated using the thermal wind balance ($v^g$), and c) ageostrophic velocities calculated as the difference between VMADCP and geostrophic velocities ($v^a = v - v^g$). Black contours correspond to $\sigma_\theta$ levels (every 0.05 kg m$^{-3}$), with thicker lines highlighting the 25.2, 25.5, and 26.6 kg m$^{-3}$ isopycnals, which are also labeled for reference. Light green contours denote isohalines (36.8–36.9 g kg$^{-1}$, every 0.05 g kg$^{-1}$).

Vertical sections of the horizontal velocity components, $\sigma_\theta$, and $S_A$ were extracted along the virtual transect depicted in Fig. 8, offering a continuous depth-wise representation of the observed features and patterns (Fig. 9). To better represent the circulation around the eddy center, horizontal velocity vectors were rotated along the transect axis. The anticyclonic circulation associated with the Bentayga eddy dominated the entire water column along this transect, with peak cross-transect velocities exceeding 30 cm s$^{-1}$ in the upper 200 m (Fig. 9a). Within this vertical segment, just below the mixing layer ($z$ >70-80 m), the

430 isopycnals exhibited a steep inclination where the vertical density gradient was strongest. The highest velocities (>60 cm s$^{-1}$) were located at the eddy's edges, progressively converging towards the boundaries of the eddy core with increasing depth. This

subtle inward inclination was most pronounced between 80 and 150 m, where slight doming and deepening of the isopycnals delineating the core were also observed. Furthermore, within it, the isopycnals displayed small-scale irregularities, characterized by doming on one side and deepening on the other side. These patterns were mirrored by the isohalines delineating the relatively less saline eddy core (36.8–36.9 g kg$^{-1}$). While previous interpretations attributed these features to the lateral advection of lighter waters from the surrounding ring into the core, the observed isopycnal and isohaline perturbations may also be indicative of vertical displacements within the eddy structure.

Using the thermal wind balance relationship, the geostrophic current field was derived from the interpolated density field, with VMADCP records at 312 m depth serving as the reference level (Fig. 9b). Generally, VMADCP-measured velocities were more intense than the geostrophic currents, though their spatial patterns showed only minor discrepancies. Similar to the VMADCP observations, geostrophic currents were strongest at depths shallower than 200 m (>30 cm s$^{-1}$), with maximum values above 100 m depth (50-55 cm s$^{-1}$). The ageostrophic horizontal velocity was revealed by subtracting the geostrophic component from the VMADCP velocity field (Fig. 9c). These velocities were strongest within the upper 150 m of the water column (>5 cm s$^{-1}$), peaking at over 10 cm s$^{-1}$ between 80 and 120 m depth.

Water parcels that follow curved trajectories within an AE require a net centripetal acceleration to maintain their motion in equilibrium. In these conditions, the appropriate momentum balance is the cyclogeostrophic one, which includes this centripetal acceleration explicitly (Flierl, 1979; Kunze, 1986; Olson, 1980; Shakespeare, 2016). Assuming steady, inviscid, and axisymmetric flow, the radial component of the horizontal momentum equation becomes:

$$-\frac{v_\theta^2}{r} = f v_\theta - \frac{1}{\rho_0}\frac{\partial p}{\partial r},$$

where $v_\theta$ is the azimuthal velocity, $r$ is the radial distance to the eddy center, $\rho_0$ is the mean background density, and $p$ is the pressure. As shown in Shakespeare (2016), the left-hand side expresses the centripetal acceleration required to sustain the curved trajectory, balancing the physical forces that provide it—namely, an inward Coriolis acceleration and an outward-directed pressure gradient. This implies that, when streamline curvature becomes significant in an AE, part of the observed ageostrophic horizontal velocity arises as a compensating effect of the curvature itself.

To quantify the importance of curvature in the momentum balance, a nondimensional number can be defined as $C = v_\theta/(fr)$, known as the cyclogeostrophic Rossby number. Values of $C \gtrsim 0.1$ are typically considered sufficient to indicate that the centripetal acceleration required to sustain flow curvature becomes dynamically significant (e.g., Shakespeare, 2016; Ioannou et al., 2019; Penven et al., 2014; Douglass and Richman, 2015). Given that the virtual transect was nearly orthogonal to the eddy's mean flow, the azimuthal velocity component $v_\theta$ closely corresponds to the cross-transect velocity shown in Fig. 9a. This component, under the assumption $v \approx v_\theta$, was therefore used in the calculation of $C$ for the Bentayga eddy.

The diagnosed vertical distribution of $C$ reveals values exceeding 0.1 at depths $z \leq 160$ m when using either the mean or the upper 10% of $v_\theta$ within the eddy core. Peak values ($\gtrsim 0.15$) occurred between 80 and 160 m, coinciding with the depth range of strongest observed cross-transect ($v$) and ageostrophic ($v^a$) velocities. This suggests that curvature effects contribute significantly to the momentum balance in the core region, particularly for radial distances $r \lesssim 40$ km, promoting compensating ageostrophic velocities within the core that help sustain the anticyclonic circulation of the Bentayga eddy.

### 3.4 Oxygen distribution and biogeochemical signatures

The distribution of $O_2$ across the Bentayga eddy provides valuable insights into its biogeochemical imprint (Fig. 10). Vertical sections along the OceT transect reveal a well-defined subsurface minimum in absolute $O_2$ concentrations (Fig. 10a), with values below 210 $\mu$mol kg$^{-1}$ centered around 110–130 m depth and extending across stations 14–22. This oxygen-depleted layer coincides with the pycnostad core of the eddy, bounded by the $\sigma_\theta = 25.2$–25.5 kg m$^{-3}$ isopycnals. Anomalies relative to a reference vertical profile, derived from stations outside the eddy influence (as described in Section 3.2), further highlight this core as a region of moderate $O_2$ deficit ($AO_2 < -5$ $\mu$mol kg$^{-1}$) (Fig. 10b). Beneath this negative anomaly layer, $O_2$ concentrations exceed the reference values, producing positive anomalies ($AO_2 > 15$ $\mu$mol kg$^{-1}$) that closely follow the 25.5–25.6 kg m$^{-3}$ isopycnals.

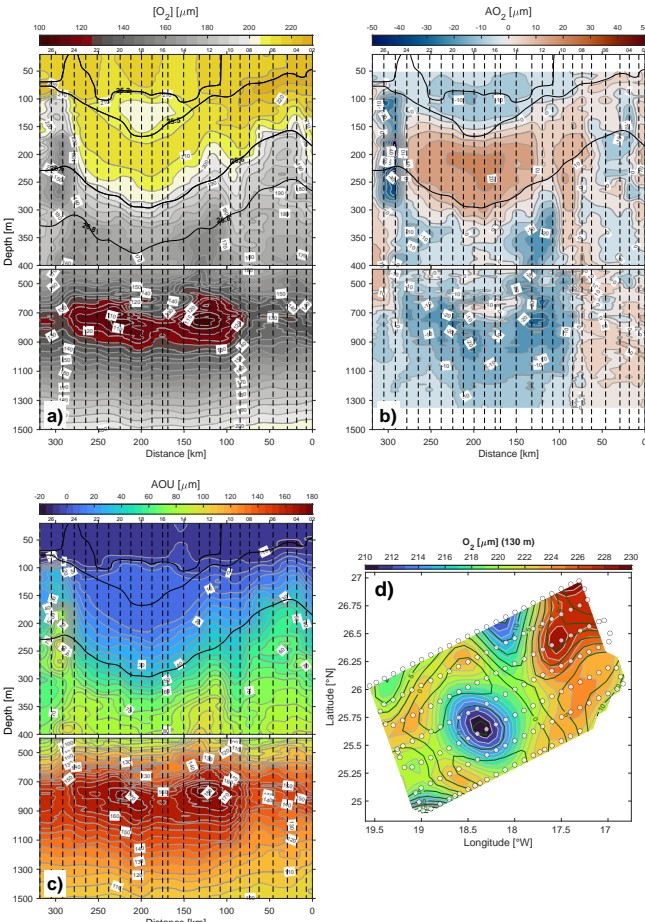

**Figure 10.** Dissolved oxygen structure of the Bentayga eddy. (a) Absolute $O_2$ concentrations [$\mu$mol kg$^{-1}$] along the OceT transect. (b) $O_2$ anomalies ($AO_2$) relative to the reference profile. (c) Apparent Oxygen Utilization (AOU). (d) Horizontal section at 130 m depth showing objectively interpolated $O_2$ (colormap) and AOU (dark green contours).

The Apparent Oxygen Utilization (AOU) field (Fig. 10c) offers additional support for this interpretation. Within the eddy core, AOU values remain relatively low, around 10 $\mu$mol kg$^{-1}$, slightly higher than surface waters (ranging from -10 to 0 $\mu$mol kg$^{-1}$) but well below typical values for thermocline or intermediate-depth waters. These low-AOU values extend deeper inside the eddy than in surrounding waters, with concentrations <35 $\mu$mol kg$^{-1}$ reaching depths of ~270 m and <80 $\mu$mol kg$^{-1}$ persisting to ~400 m. In contrast, values exceeding 120 $\mu$mol kg$^{-1}$ are observed below 500 m throughout the transect, outside the influence of the eddy. This vertical structure suggests that the eddy retains relatively young, recently subducted water masses, likely of surface or coastal origin. The limited oxygen consumption implied by the low AOU values indicates a weakly ventilated but not yet biogeochemically aged core.

A horizontal section at 130 m depth (Fig. 10d) further confirms the spatial coherence of this relatively oxygen-depleted, low-AOU core. However, it seems that during the SeaSoar phase, AOU values within the core were even lower, ranging between 0 and 2 $\mu$mol kg$^{-1}$, and were encircled by a weak negative-AOU ring (–4 to –1 $\mu$mol kg$^{-1}$). These slightly negative values indicate that the water was slightly oversaturated with respect to atmospheric equilibrium, consistent with the recent subduction of well-oxygenated surface waters into the eddy core and limited biological and microbial oxygen demand. This distribution supports the interpretation that the eddy trapped and transported oxygen-poor waters offshore, potentially sourced from coastal upwelling regions, as suggested by the satellite-derived eddy trajectory and its dynamical properties. The combination of hydrographic isolation and distinct $O_2$ and AOU signatures underscores the eddy's role in redistributing oxygen and modifying ventilation patterns at thermocline depths, pointing to biogeochemical impacts that extend beyond its dynamic and thermohaline structure.

## 3.5 Azimuthal velocity structure and eddy-core rotational dynamics

To investigate the rotational structure of the Bentayga eddy, we analyzed the radial distribution and vertical variability of azimuthal velocities ($u_\theta$) derived from multiple observational phases. This combined approach provided a comprehensive view of the eddy's internal dynamics, capturing both vertical coherence and horizontal structural transitions. Figure 11 shows the depth-resolved azimuthal velocity profiles extracted from the OceT transect. At each depth, a least-squares fit of the form $u_\theta = \omega_\theta r$ was applied to the observed velocity data to quantify the degree of solid-body rotation within the eddy core. Here, $\omega_\theta$ denotes the angular velocity, and the constraint $u_\theta = 0$ at the eddy center ($r = 0$) was imposed to ensure physical consistency. The linear fit was iteratively computed over increasing radial distances, and an effective core radius was defined as the radial distance at which the correlation coefficient ($r_{xy}$) between the fitted and observed data reached its maximum.

The results reveal a well-defined solid-body core extending laterally up to $\sim 23-28$ km (close to the observed R$_c$ in Sect.3.2 and in the Supplementary Note 3), and vertically from the shallowest observed level (~40 m) to approximately 350 m depth, where $r_{xy} \geq 0.9$. Below 350 m, the rotational coherence weakened, with $r_{xy}$ reaching a minimum of ~0.75 between 360 and 400 m. Interestingly, $r_{xy}$ increased again between 415 and 450 m, remaining above 0.8 down to ~600 m, after which it declined sharply, indicating the gradual erosion of solid-body characteristics with depth. The vertical structure of the rotation frequency $\omega_\theta$ (Fig. 11d) showed a nearly constant surface-averaged angular velocity down to 80 m, at approximately $1.35 \times 10^{-5}$ rad s$^{-1}$ (corresponding to a ~5-day period). A peak in $\omega_\theta$ was observed between 110 and 130 m, reaching $1.85 \times 10^{-5}$ rad s$^{-1}$ (~3.9-

510  day period), after which the rotation rate declined abruptly, reaching ∼70% of its peak by 210 m. This behavior suggests that the core is composed of vertically stratified layers, each rotating quasi-independently rather than as a uniform column, as similarly observed in meddies and other intrathermocline structures (Schultz Tokos and Rossby, 1991).

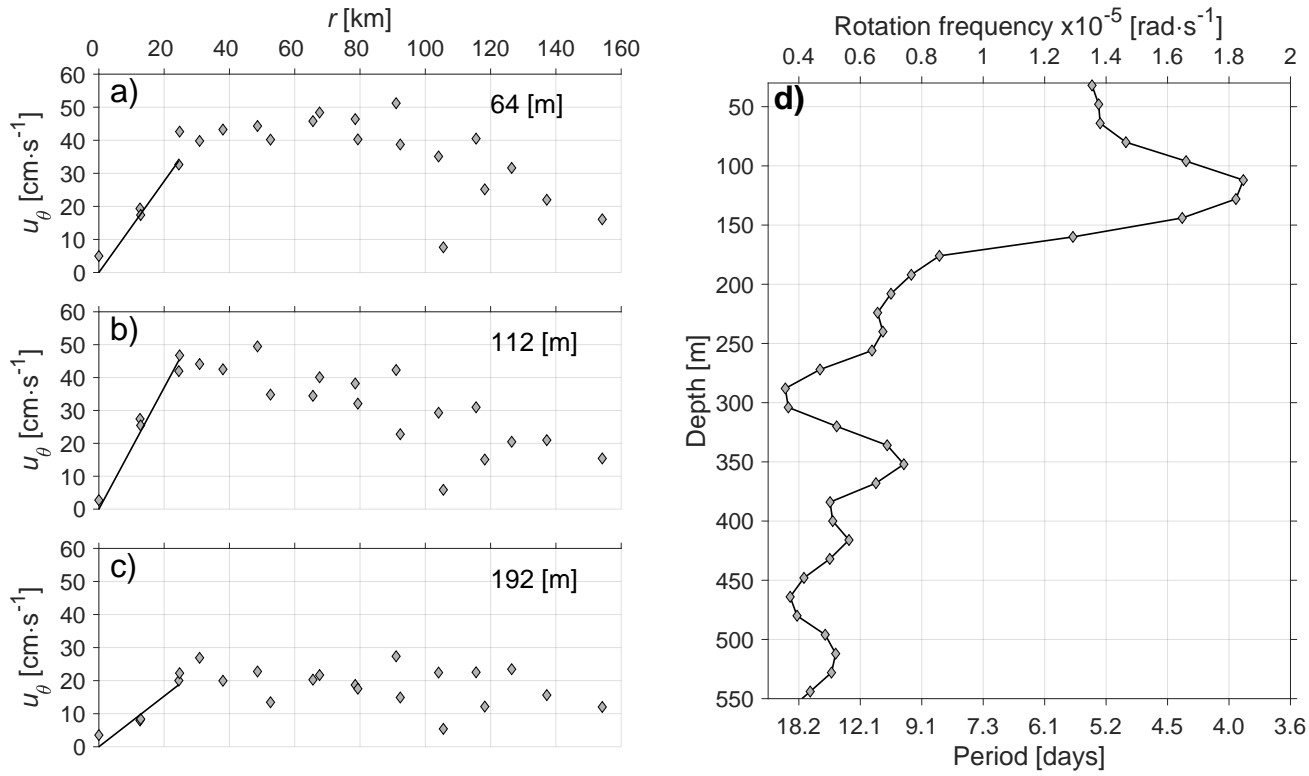

**Figure 11.** Radial profiles of azimuthal velocity ($u_\theta$) for the OceT transect at selected depths: a) 64 m, b) 112 m, and c) 192 m, derived from the OceT. Grey diamonds represent the observed data, while solid black lines correspond to the linear regression fit using $u_\theta = \omega_\theta r$, with $u_\theta = 0$ at $r = 0$. The slope of the regression ($\omega_\theta$) is proportional to the angular velocity of the eddy. d) Depth-wise variation of the solid-body core's rotation frequency ($\omega_\theta$) and its corresponding period.

Complementing this vertical analysis, the radial structure of azimuthal velocities in the 80–160 m depth range was evaluated across several eIMPACT2 survey phases, and is shown in Figure 12a. Despite minor asymmetries, the velocity profiles remain

consistent across them. Within the well-mixed core ($|r| \leq R_c = 23$ km), the profiles exhibit a nearly linear increase in $u_\theta$, consistent with solid-body rotation. Beyond this region, azimuthal velocities gradually increase to a maximum near $|r| \sim 35$–45 km before declining smoothly, a pattern reminiscent of Gaussian-type vortices. To capture this structural transition, a hybrid vortex model was developed by combining a Rankine and Gaussian profiles using a logistic weighting function (see Supplementary Note 3 for full formulation). This model, shown as a dashed black curve in Figure 12a, reproduces the linear

behavior within the core and the gradual decay in the outer region. The transition starts around $R_c$, and the hybrid formulation

remains valid up to $R_e \approx 3R_c \approx 70$ km, with $R_e$ corresponding to the outer edge of the previously defined surrounding ring, thereby capturing the eddy's full dynamical extent. In addition, Figure 12b shows the normalized relative vorticity ($\zeta/f$) and strain rate ($\eta/f$), computed from $u_\theta(r)$ assuming axisymmetry. While $\zeta/f$ follows the hybrid model closely—especially in the surrounding ring sector—the strain rate displays higher variability and deviates from theoretical expectations, possibly due to azimuthal asymmetries, resolution limits, or submesoscale deformation processes not accounted for in the axisymmetric model assumptions (see discussion in Supplementary Notes 3-5).

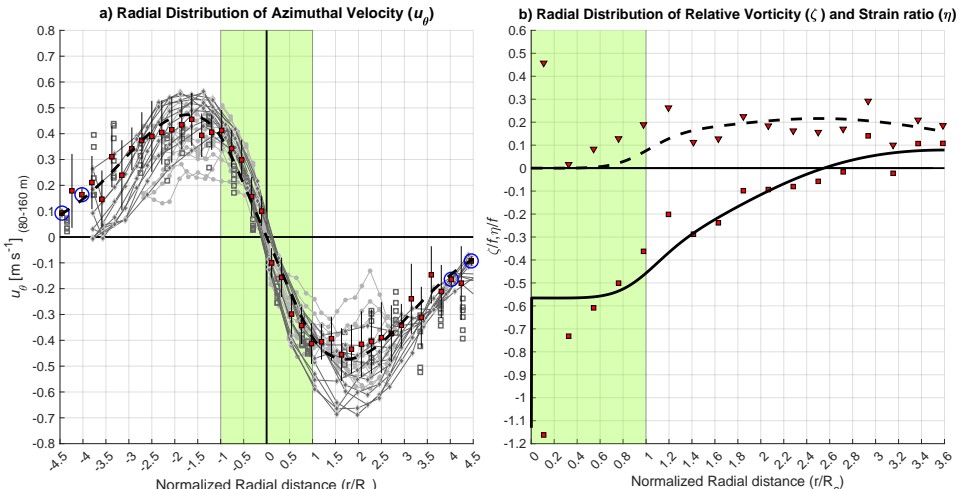

**Figure 12.** a) Radial distribution of azimuthal velocity ($u_\theta$) between 80 and 160 m depth, relative to the center of the Bentayga eddy normalized by $R_c$. Gray markers represent individual measurements from different survey phases: SeaSoar (T3 and T4; dark gray diamonds), Orthotransects (ZT and MT; light gray circles), and OceT (dark gray open squares) (see Supplementary Note 3 for further details). The binned mean profile is shown as red-filled black squares, calculated over 5-km radial intervals; vertical bars indicate $\pm 1$ standard deviation. Blue-circled markers highlight bins with fewer than 10 observations. An hybrid model-derived velocity profile is overlaid (black dashed). b) Normalized relative vorticity ($\zeta/f$, squares and solid lines) and strain rate ($\eta/f$, triangles and dashed lines) as a function of normalized radial distance. Symbols denote observed values, and curves follow the same color scheme as in panel a). The green-shaded region in both panels marks the extent of the well-mixed eddy core ($|r| \leq R_c = 23$ km).

To dynamically characterize the internal structure of the Bentayga eddy, the Rossby number (Ro) was calculated as the ratio of the vertical component of the relative vorticity field to planetary vorticity ($\zeta_{OI}$ and $f_{OI}$, respectively). The relative vorticity ($\zeta_{OI}$) was derived from the objectively interpolated VMADCP velocity fields using $\zeta_{OI} = \partial v/\partial x - \partial u/\partial y$. A horizontal section of the three-dimensional Ro field was extracted at a depth of 104 m, corresponding to the level of maximum VMADCP velocities recorded during the eIMPACT2 SeaSoar phase (Fig. 13a, see also Fig. 14). Additionally, a vertical section along the virtual transect shown in Fig. 13a was examined (Fig. 13b). These analyses revealed that its core was dominated by negative vorticity, with Ro values within the core reaching below -0.45 and a peak negative vorticity of approximately $-0.61f$ at its

center. Ro increased radially from the core toward the outer regions, transitioning to positive values in the central part of the
surrounding ring (45-50 km from the eddy center), where Ro ranged between 0.1 and 0.2.

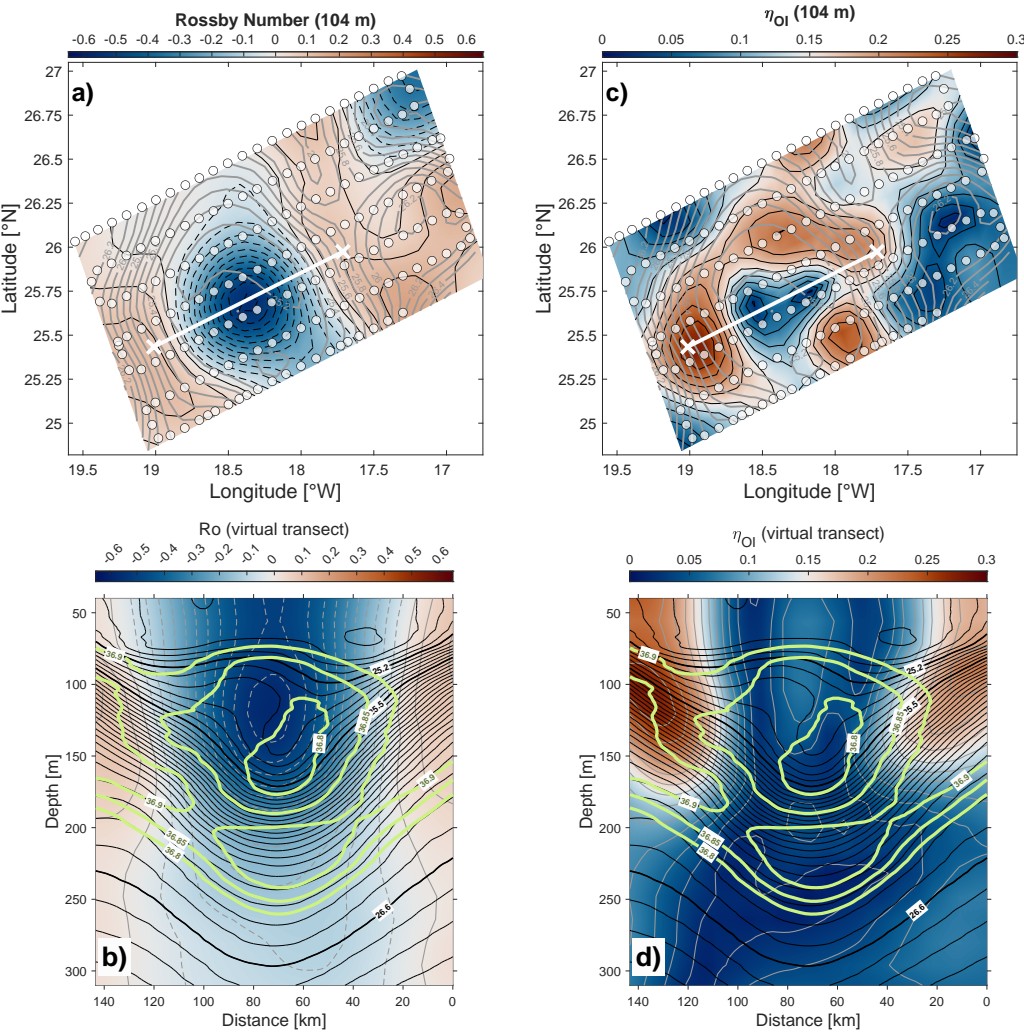

**Figure 13.** Rossby number (Ro) and strain rate ($\eta_{OI}$) derived from the objectively interpolated VMADCP velocity fields. a) Horizontal
section of Ro at 104 m depth, corresponding to the depth of maximum swirl speed of the eddy (refer to Fig. 14). Contours of Ro are shown
every 0.05, with negative values represented by dashed lines and positive values by solid lines. b) Vertical section of Ro along the virtual
transect in panel (a) (thick black line). Contours of Ro are displayed every 0.05, with dashed lines for negative values. c) and d) Same as a)
and b) but for the $\eta_{OI}$. Contours are shown every 0.025. Objectively interpolated $\sigma_\theta$ contours (every 0.1 kg m$^{-3}$) are shown as grey (a and
c) and black lines (b and d). In b) and d) $\sigma_\theta = 25.2$, 25.5, and 26.6 kg m$^{-3}$ are highlighted and labeled with thick black lines for clarity, and,
additionally, the 36.8–36.9 isohalines (every 0.05 g kg$^{-1}$) are included as thick light green contours.

In addition to Ro, the strain rate field ($\eta_{OI}$) was analyzed to further characterize the dynamical structure of the eddy and its surrounding environment (Figs. 13c, d). The strain rate was also computed from the interpolated VMADCP velocities using $\eta_{OI} = \sqrt{(\partial u/\partial x - \partial v/\partial y)^2 + (\partial v/\partial x + \partial u/\partial y)^2}$, and normalized by $f_{OI}$. The horizontal section at 104 m depth (Fig.13c) revealed a well-defined annular band of elevated strain rate (0.1–0.2) encircling the eddy core, especially concentrated near its western and northeastern flanks. These strain maxima occurred between 35–60 km radial distance from the eddy center, roughly aligning with the radial position of maximum azimuthal velocities (see Fig.12a), and indicating enhanced deformation at the transition between the inner core and surrounding ring. In the vertical section along the virtual transect (Fig.13d), strain values were lowest (<0.05) within the core region above $\sigma_\theta = 26.6$ kg m$^{-3}$, consistent with the nearly solid-body rotation observed in the core (Figs. 11 and 12). Notably, strain values inside the eddy core reached $\sim 0.1$ in localized regions where the $\sigma_\theta = 25.3$ kg m$^{-3}$ isopycnal displayed asymmetric behavior. A distinct patch of high strain ($\sim$0.3) emerged at depths of 80–200 m in the eddy's western and eastern flanks, suggesting strong horizontal shear possibly related to interactions with the surrounding flow or adjacent cyclonic structures.

Together, the vertical coherence of $\omega_\theta$ (Fig.11), the radial transition captured by the hybrid model (Fig.12), and the complementary patterns in the Rossby number and strain rate fields (Fig. 13) offer a consistent and physically grounded depiction of the eddy's rotational structure. These diagnostics collectively support the use of a solid-body approximation within the core and a smooth transition to a Gaussian-like regime in the surrounding ring, while revealing a rotationally coherent interior embedded within a broader region of enhanced deformation—hallmarks of a mature anticyclonic intrathermocline eddy. This provides a robust framework for defining the integration limits used in subsequent calculations of energy content, hydrographic anomalies, and transport properties.

The horizontal location where the Rossby number (Ro) changes sign—i.e., where $\zeta_{OI} = 0$—was used to dynamically identify the outer boundary of the eddy's spatially coherent horizontal structure. This occurred at a radial distance of approximately 62 km, closely aligning with the outer edge of the $\eta_{OI}$ maximum zone in the surrounding ring. Notably, this location corresponds to $\sqrt{2}R_{max}$, where $R_{max} \approx 41$ km marks the radius of maximum azimuthal velocity (Fig.12; see also Supplementary Note 3). Interestingly, $R_{max}$ is in excellent agreement with the climatological first baroclinic Rossby radius of deformation, estimated at 42.8 km for this latitude using the empirical polynomial from Chelton et al. (1998). As expected for Gaussian-like anticyclones, the Ro sign reversal occurs beyond the velocity peak. This consistency between theory and observation reinforces the validity of our dynamical interpretation and the robustness of our radial zonation. We therefore adopt this boundary to define the outer shell enclosing the rotationally coherent region of the eddy, which also coincides with the external edge of the strain-dominated sector, and use it as a reference to evaluate its vertical structure and water-trapping capacity.

Once this boundary was defined, circum-averaged azimuthal velocities were computed at each VMADCP depth, yielding a vertical profile of the average shell velocities along with their corresponding standard deviation (Fig. 14). Maximum shell velocities were observed near the surface, at depths shallower than 120 m, with a relative peak centered at 104 m (approximately $50\pm10$ cm s$^{-1}$). Below this peak, the shell velocity magnitude decreased sharply, extending down to 200 m depth, with a mean vertical shear of 0.004 s$^{-1}$. At greater depths, the velocity reduction was more gradual, with a mean shear of 0.002 s$^{-1}$ down to about 600 m depth. As shown in Fig. 14, velocities below 550 m approached the eddy's mean translational speed ($c$), and

between 550 and 750 m they fell below this threshold. This suggests that, at such depths, the eddy's water trapping capacity may no longer be sufficient to significantly limit exchange with the surroundings, thereby reducing its ability to transport and maintain these water masses along its trajectory.

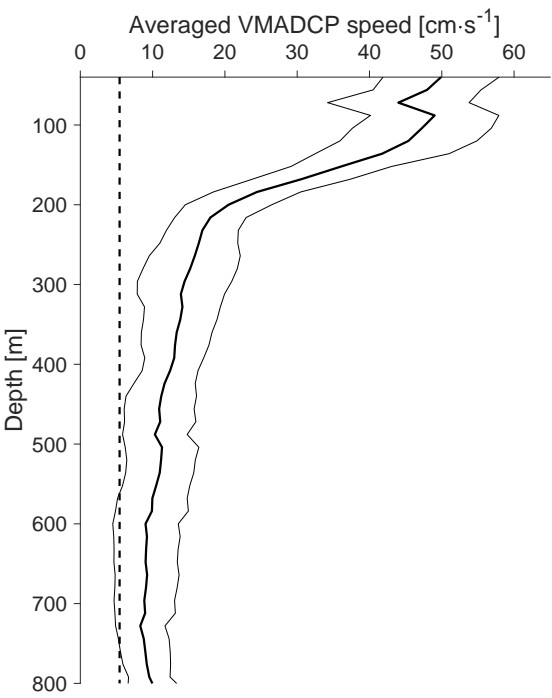

**Figure 14.** Vertical profile of the averaged VMADCP shell velocities (thick black line) with its corresponding standard deviation (represented by thin black lines) during the eIMPACT SeaSoar phase. The dashed vertical line indicates the average translational speed of the eddy (5.4 cm s$^{-1}$) during the eIMPACT2 survey.

### 3.6 Potential vorticity and stratification

The radial distributions of Ro and $\eta_{OI}$ provide a dynamical framework for interpreting the eddy's structure, with the former emphasizing rotational dominance and the latter indicating minimal deformation in the core and increasing strain toward the periphery—together confirming a highly coherent, vorticity-dominated interior. Complementing this, the vertical stratification and PV fields characterize the internal stability of the water column and the magnitude of PV anomalies within the eddy. Given the conservative nature of PV in the absence of external forcing and dissipation, such anomalies are robust indicators of water

mass retention. Combined, these analyses elucidate the interplay between the eddy's rotational dynamics and its stratification-controlled trapping capacity. As discussed earlier (Sect. 1 and 2.3.1), ITEs are characterized by a large reduction in PV within their core, primarily due to their reduced stratification and the homogeneous water they enclose, resulting in significant negative PV anomalies relative to the surrounding environment. The calculated PV of the Bentayga eddy related to isopycnal tilting was

found to be two orders of magnitude smaller than the PV associated with water column stretching. This aligns with previous findings, such as those for the PUMP eddy by Barceló-Llull et al. (2017), and reflects the characteristic behavior of ITEs, where the vertical density gradient, $\partial_z\sigma_\theta$, i.e., the stratification of the water column, predominantly controls the magnitude of their core's PV.

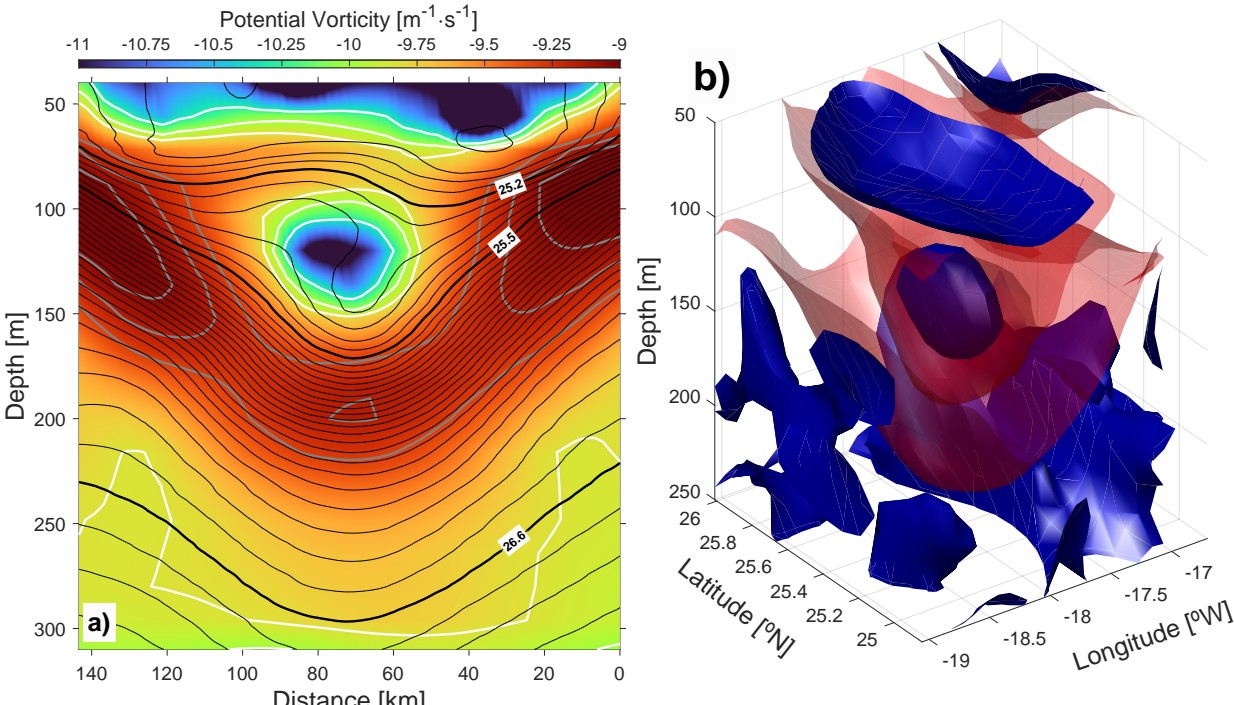

**Figure 15.** Ertel's potential vorticity (PV) derived from objectively interpolated VMADCP velocities and $\sigma_\theta$ fields. a) Vertical section of $\log_{10}$(PV) along the virtual transect indicated by the thick black line in Figs. 8 and 13a. Extreme PV levels are represented by thick white and grey contours, highlighting the lowest ($5\times10^{-11}$, $10\times10^{-11}$, and $15\times10^{-11}$ m$^{-1}$ s$^{-1}$) and highest PV values ($5\times10^{-10}$, $7\times10^{-10}$, and $9\times10^{-10}$ m$^{-1}$ s$^{-1}$), respectively. Thin black contours indicate $\sigma_\theta$ levels (spaced every 0.05 kg m$^{-3}$). For clarity, $\sigma_\theta$ levels at 25.2, 25.5, and 26.6 kg m$^{-3}$ are labeled and displayed as thick black lines. b) Three-dimensional view of the PV field, represented by the $15\times10^{-11}$ and $5\times10^{-10}$ m$^{-1}$ s$^{-1}$ surfaces, shown as dark blue and light yellow sheet-like patches, respectively.

As seen in Fig. 15, the lowest PV values ($\sim 10^{-11}$ m$^{-1}$ s$^{-1}$) were observed in the shallowest layers and within the inner core, where water exhibited high homogeneity and low stratification. As anticipated, higher PV values ($>5\times10^{-10}$ m$^{-1}$ s$^{-1}$) were detected in regions of greater stratification, where closely spaced isopycnal surfaces indicated compression of the water column in the vertical dimension. The highest PV values ($>9\times10^{-10}$ m$^{-1}$ s$^{-1}$) were detected near regions of strong isopycnal lifting at the periphery of the Bentayga eddy, possibly influenced by adjacent CEs. Furthermore, the spatial distribution of PV highlights the sharp stratification contrast between its anomalously low-PV core and the surrounding layers of relatively higher

PV (Fig. 15b). At greater depths, below these surrounding layers, PV values diminished gradually, reaching approximately
595 $1 \times 10^{-10}$ m$^{-1}$ s$^{-1}$, indicating a progressive decrease in PV distribution with depth.

## 3.7 Energy content

The net available potential energy (APE) and kinetic energy (KE) within the Bentayga eddy were estimated using Eqs. 2 and 3, integrating from the surface to a depth (H) of 550 m, corresponding to the eddy's effective trapping depth as inferred from hydrographic and velocity anomalies (see Section 3.5). This depth marks the lower boundary of the coherent thermohaline
structure, beyond which the eddy signature decays rapidly. Table 1 presents the detailed APE and KE values obtained using three physically grounded radial limits: the solid-body core radius ($R_c$ = 23 km), the radius of maximum azimuthal velocity ($R_{max}$ = 41 km), and the external edge of the surrounding ring ($R_e$ = 70 km). These limits were defined based on the rotational and structural characteristics of the eddy and are discussed in detail in the Supplementary Note 3.

Across this radial range, APE increased from 0.006 PJ to 0.031 PJ, while KE increased more sharply from 0.001 PJ to
605 0.021 PJ (1 PJ = $10^{15}$ J). At the solid-body core radius ($R_c$), APE was approximately 368% greater than KE; this contrast decreased to 126% at $R_{max}$, and further to 45% at $R_e$. As a result, the energy Burger number ($B_E$ = KE/APE) rose from 0.21 at $R_c$ to 0.69 at $R_e$, indicating a progressive shift toward kinetic energy dominance as larger portions of the eddy's surrounding ring are incorporated. This predominance of APE over KE is a hallmark of well-established intrathermocline eddies (ITEs) and is consistent with findings from previous studies (e.g., Schultz Tokos and Rossby, 1991; Schmid et al., 1995; Fernández-Castro
et al., 2020). Moreover, it reflects a mature stage in the eddy's life cycle, during which the balance between available potential and kinetic energy shifts as the structure adjusts and stabilizes (e.g., Schultz Tokos and Rossby, 1991).

**Table 1.** Energy content and Burger number estimates for the Bentayga eddy, evaluated across three horizontal scales ($L_x$ = 2R) corresponding to distinct integration radii: the solid-body core ($R_c$), the radius of maximum azimuthal velocity ($R_{max}$), and the outer edge of the surrounding ring ($R_e$). Kinetic energy (KE) and available potential energy (APE) were used to compute the energy Burger number ($B_E$ = KE/APE). The table also presents the length-scale Burger number ($B_L = N^2 L_Z^2 / f^2 L_x^2$) and the parametrized energy Burger number ($B_E^L = B_L/(1+Ro)^2$), estimated using two values of Rossby number: the average (Ro = 0.47) and the maximum (Ro = 0.65) vorticity within the eddy core. All calculations were performed using fixed values of squared Brunt–Väisälä frequency ($N^2 = 3.55 \times 10^{-5}$ s$^{-2}$) and Coriolis parameter ($f = 6.33 \times 10^{-5}$ s$^{-1}$), representative of the eddy's core conditions.

| $L_x$ [km] | KE [$10^{13}$ J] | APE [$10^{13}$ J] | $B_E$ | $B_L$ | $B_E^L$(Ro=0.47) | $B_E^L$(Ro=0.65) |
|---|---|---|---|---|---|---|
| $2R_c$ | 1.24 | 5.80 | 0.21 | 1.27 | 0.59 | 0.47 |
| $2R_{max}$ | 6.64 | 14.98 | 0.44 | 0.40 | 0.19 | 0.15 |
| $2R_e$ | 21.33 | 30.94 | 0.69 | 0.14 | 0.06 | 0.05 |

To explore the origin of this trend, we evaluated the length-scale Burger number, $B_L = N^2 L_Z^2 / f^2 L_x^2$, and its theoretical connection to the energy partitioning through the parameterized form $B_E^L = B_L/(1+Ro)^2$, where $L_Z = H = 550$ m and $L_x = 2R$. Ro values used in this parameterization were derived from the eddy core and correspond to the average (|Ro| = 0.47) and maximum (|Ro| = 0.65) relative vorticity. For consistency, all calculations were performed using fixed values of $N^2 = $

$3.55 \times 10^{-5}$ s$^{-2}$ and $f = 6.33 \times 10^{-5}$ s$^{-1}$, representative of the eddy's core. The results show that $B_L$ decreases with increasing radius, from 1.27 at $R_c$ to 0.14 at $R_e$, reflecting the changing aspect ratio of the integrated volume. $B_E^L$ also decreases accordingly and shows the closest agreement with $B_E$ at $R = R_{max}$. However, this apparent convergence must be interpreted with caution, as the estimations rely on data collected during the eIMPACT2 OceT phase, which is particularly susceptible to sampling-induced distortions. At larger radii, $B_E^L$ progressively underestimates the observed ratio, reflecting not only the expected sampling-related distortion but also the growing influence of processes not captured—such as the contribution of horizontal shear, flow strain, ageostrophic motions, and radial structural complexity to the relative increase in kinetic energy. At $R = R_c$, by contrast, $B_E^L$ overestimates $B_E$ by nearly a factor of two, although the version computed using the maximum core vorticity provides a closer approximation. This outcome reinforces the notion that energy partitioning within the eddy is fundamentally constrained by its geometry and dynamical state, while also highlighting the limitations of theoretical scalings when applied to observational data affected by sampling biases.

Supplementary Notes 4 and 5 demonstrate that the solid-body core remains the least affected by sampling-induced distortions, reinforcing the interpretation of $R_c$ as the most reliable spatial limit for estimating the eddy's intrinsic energetics. This supports the notion that the eddy's core retains the most coherent and energetically stable signature. However, the progressive divergence observed between theoretical expectations and empirical estimates at larger radii highlights the inherent difficulties of reconciling these approaches, particularly under these sampling conditions. Despite these uncertainties, the general pattern indicates that the internal energy balance of the eddy is primarily governed by geometric and dynamical constraints. Previous studies (e.g., McWilliams, 1985; Chelton et al., 2011) emphasized the role of aspect ratio in shaping eddy energetics, while Schultz Tokos and Rossby (1991) illustrated how strong vorticity can modulate the distribution of kinetic and potential energy within mesoscale ocean structures.

To contextualize these results, Table 2 summarizes the energy partitioning and dynamical indicators of several ITEs previously documented in the literature. Most exhibit $B_E$ between 0.5 and 1.5, although substantial variability arises depending on the eddy's age, stratification, and geographical setting. Within this framework, the Bentayga eddy's value of $B_E = 0.21$, when evaluated at $R_c$, lies at the lower end of the spectrum, consistent with a strongly baroclinic structure dominated by potential energy. In contrast, the PUMP eddy—also sampled within the CEC—displayed a KE value approximately 50% higher than its APE (Barceló-Llull et al., 2017). This divergence is particularly noteworthy considering that both eddies were of similar age at the time of their respective surveys, suggesting potential differences in their formation mechanisms, environmental interactions, or stages within their life cycles. In summary, both the empirical estimates and the theoretical parameterizations support the use of $R_c$ as the most representative integration radius for quantifying the energy content and dynamical state of the Bentayga eddy. This limit encompasses a region of high structural coherence and minimal sampling distortion, thus providing a conservative yet robust lower bound for evaluating the eddy's contribution to regional mesoscale energetics.

**Table 2.** Comparison of energy content and associated dynamical parameters for intrathermocline eddies (ITEs) from previous studies. The vertical extent (H) and characteristic radius (R) used to estimate the available potential energy (APE) and kinetic energy (KE) are listed alongside the resulting energy Burger number ($B_E = KE/APE$). Additionally, the length-scale Burger number ($B_L = N^2 L_Z^2 / f^2 L_X^2$), Rossby number, and the parameterized energy Burger number ($B_E^L = B_L/(1 + Ro)^2$) are presented. Here, the background squared Brunt-Väisälä frequency ($N^2$), the Coriolis parameter ($f$), and the eddy's vertical and horizontal scales ($L_Z$ and $L_X$, respectively), used for the calculations are also included. When $L_Z$ and $L_X$ correspond to the spatial domain used for APE and KE calculations, the associated values of H and R, as defined in this table, are referenced.

| ITE's properties | Schultz Tokos and Rossby (1991): Meddy (October 1984) | Schultz Tokos and Rossby (1991): Meddy (October 1985) | Prater and Sanford (1994): A two core newly formed Meddy | Schmid et al. (1995): AE _Vitória_ Brazil Current | Barceló-Llull et al. (2017): PUMP | Fernández-Castro et al. (2020): AE west off Great Abaco Island |
|---|---|---|---|---|---|---|
| H (m) | 900 | 600 | 650 | 400 | 500 | 1000 |
| R (km) | 65 | 40 | 9 | 50 | 46 | 80 |
| APE ($\times 10^{13}$ J) | 7.5 | 2.1 | 0.2 | 19.0 | 5.6 | 438 |
| KE ($\times 10^{13}$ J) | 7.9 | 1.2 | 0.4 | 9.6 | 8.9 | 36 |
| $B_E$ | 1.05 | 0.57 | 2.00 | 0.51 | 1.59 | 0.08 |
| $N^2$ ($\times 10^{-5}$ s$^{-2}$) | – | – | 0.8 | – | 2.8* | 2.5 |
| $f$ ($\times 10^{-5}$ s$^{-1}$) | 7.7 | 6.6 | 8.6* | 5.1* | 6.4* | 7.1* |
| $L_Z$ (m) | – | – | H | – | H | 500 |
| $L_X$ (km) | – | – | 2R | – | 2R | 120 |
| $B_L$ | – | – | 1.4 | – | 0.2 | 0.1* |
| Ro | -0.3 | -0.4 | -0.9 | -0.2 | -0.6 | -0.1 |
| $B_E^L$ | – | – | 61.54* | – | 1.25 | 0.09 |

The * symbol denotes values that were not explicitly provided in the respective studies.

These were inferred from figures, calculated using related information, or estimated based on the provided methodology.

Assumptions and approximations for these values are detailed in their respective references.

### 3.8 Thermohaline anomalies and transport

Distinct thermohaline anomalies were identified within the Bentayga eddy compared to the surrounding environment (Fig. 7). These anomalies formed the basis for computing the vertical profiles of available heat anomaly (AHA) and available salt anomaly (ASA). Radial integration from the eddy center to the core outer boundary ($R_c = 23$ km) revealed marked vertical differences between the $ASA(z)$ and $AHA(z)$ distributions (Fig. 16). As previously discussed in Sect. 3.7, the radial limit adopted here corresponds to the distance at which Doppler-like distortions arising from sampling asynopticity are minimized (see Supplementary Notes 1–5). Accordingly, the resulting estimates should be interpreted as conservative lower-bound values for the full eddy-induced effect. Within this boundary, thermohaline anomalies—and their associated eddy-driven fluxes—exhibit the highest degree of spatial and temporal coherence.

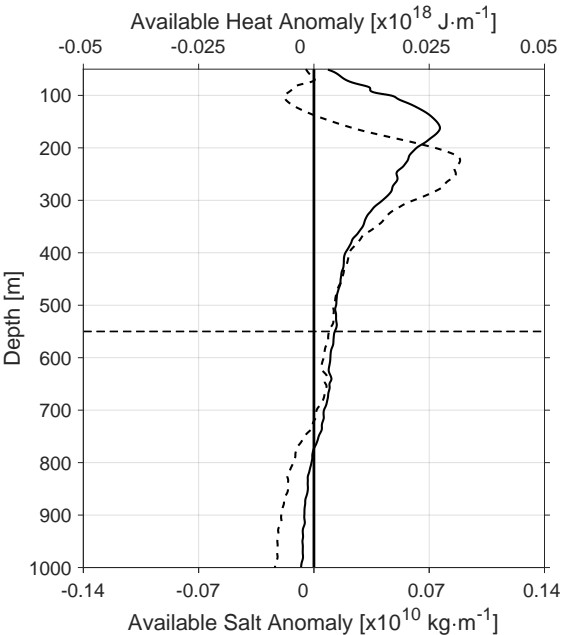

**Figure 16.** Vertical profiles of radially integrated available heat anomaly ($AHA(z)$) and available salt anomaly ($ASA(z)$) within the core of the Bentayga eddy ($R_c = 23$ km) during the eIMPACT OceT phase. The solid black line represents $AHA(z)$, while the dashed black line corresponds to $ASA(z)$. The horizontal dashed line indicates the effective trapping depth (H = 550 m), used as the lower limit for vertical integration.

$ASA(z)$ displayed predominantly positive values, except within the vertical range of 80–150 m and below 700 m. Negative anomalies in the upper layer, attributed to the salinity deficit in the core, reached a minimum of -0.18 Gkg m$^{-1}$ (1 Gkg = $10^9$ kg) at 112 m depth. Below this depth, $ASA(z)$ increased steadily, peaking at 0.89 Gkg m$^{-1}$ at 232 m. This was followed by a sharp decline with an average rate of 4.0 Mkg m$^{-2}$ (1 Mkg = $10^6$ kg) down to 400 m, transitioning to a more gradual decrease (0.8 Mkg m$^{-2}$) at greater depths. In contrast, $AHA(z)$ remained strictly positive from ~50 m to 750 m, with a

maximum value of 27.0 PJ m$^{-1}$ at 160 m - significantly shallower than the ASA($z$) peak. A minor reduction in AHA($z$) at 90 m briefly interrupted the general increasing trend, which rose with a slope of -0.2 PJ m$^{-2}$ up to the maximum. Below 160 m, the profile declined at 0.1 PJ m$^{-2}$ until 400 m, and then more gradually (0.02 PJ m$^{-2}$) at greater depths. Given that the critical trapping depth of the Bentayga eddy was 550 m, the negative anomalies in ASA($z$) and AHA($z$) below 700 m were unlikely to

be influenced by the eddy. Vertical integration of ASA($z$) and AHA($z$) to the trapping depth yielded total values of 0.015 Tkg (1 Tkg = $10^{12}$ kg) and 6.550 EJ (1 EJ = $10^{18}$ J), respectively. These positive totals underscore the persistence and intensity of heat and salt anomalies, despite localized negative segments in ASA($z$).

During the eIMPACT2 campaign, the Bentayga eddy followed a west-southwestward trajectory at an average speed of 4.5 km d$^{-1}$, consistent with the phase speed of nondispersive baroclinic Rossby waves (Chelton et al., 2007). Using this

translational speed ($c$), along with the eddy's geometric dimensions (R$_c$ = 23 km and H= 550 m), the eddy-driven volume transport ($V_e$) was estimated at 1.31 Sv. This value closely matches the transport attributed to the PUMP eddy (1.38 Sv), computed using Eq. (6) and the parameters reported by Barceló-Llull et al. (2017), and aligns with the mean mesoscale transport of 1.3 Sv estimated for the CEC by Sangrà et al. (2009). Compared to other Eastern Boundary Upwelling Systems (EBUS), the eddy-driven volume fluxes in the Canary Eddy Corridor exceed the average values reported for the Peru-Chile Current

System (PCCS) (Chaigneau et al., 2011; Hormazabal et al., 2013), but remain below those observed in the Eastern Indian Ocean (Dilmahamod et al., 2018). In terms of thermohaline transport, the Bentayga eddy contributed an eddy-driven heat flux ($Q_{eh}$) of 4.60 TW (1 TW = $10^{12}$ W), approximately one order of magnitude greater than average values reported for the PCCS (Chaigneau et al., 2011) and for both the Western and Eastern Indian Ocean (Dilmahamod et al., 2018). The eddy also transported salt at a rate of $Q_{es} \approx 0.42$ Gkg s$^{-1}$, and freshwater at $Q_{fw} \approx -0.012$ Sv. These fluxes are consistent with those

estimated for the PCCS under comparable upper-ocean conditions, while remaining roughly an order of magnitude lower than the typical values reported in the Indian Ocean (Dilmahamod et al., 2018).

## 4   Discussion

The CEC is renowned for its intense eddy activity and elevated levels of Eddy Kinetic Energy (EKE) (Barton et al., 1998; Sangrà et al., 2009). This high EKE primarily results from the interaction of the CC with wind-driven upwelling and the orographic

influence of the Canary Islands, generating mesoscale eddies that propagate westward into the open ocean (La Violette, 1974; Van Camp et al., 1991; Hernández-Guerra et al., 1993; Arístegui et al., 1994, 1997; Barton et al., 2000; Basterretxea et al., 2002; Pelegrí et al., 2005). Among these features, long-lived ITEs are mesoscale structures that persist for several months and play a critical role in maintaining the region's elevated EKE. The dense population of eddies in the CEC further promotes frequent eddy-to-eddy interactions (Sangrà et al., 2005).

Despite their significance, detailed hydrographic and dynamic studies of ITEs within the upper 500 m of the water column in the CEC are scarce. Prior to this study, only one ITE, the PUMP eddy (Barceló-Llull et al., 2017), had been comprehensively analyzed. Formed south of Tenerife Island and studied during its mature phase, this ITE exhibited dual cores: a shallow one at 80–100 m depth with azimuthal velocities exceeding 30 cm s$^{-1}$, and a deeper core at 250 m depth containing Madeira

Mode Water (MMW). By September 2014, it had traveled approximately 550 km southwestward over a period of nearly four months. In contrast, the Bentayga eddy, as examined in this study, exhibited a single cohesive core extending from 80 to 220 m depth, characterized by maximum azimuthal velocities exceeding 40 cm s$^{-1}$ and relatively homogeneous salinities between 36.8 and 36.95 g kg$^{-1}$. These differences with respect to the PUMP eddy reflect the influence of distinct physical processes and environmental conditions between their formation and subsequent observation.

For Bentayga, a combination of hydrographic and satellite-based evidence suggests the recent trapping of upwelling waters from filaments near Cape Juby and Cape Bojador during its intensification in August 2022. Specifically, the eddy's inner core exhibits relatively low oxygen concentrations but also low AOU values (Fig. 10), together with slightly cooler and fresher anomalies (Fig. 7)—signatures consistent with recently subducted waters that have not undergone substantial remineralization. This interpretation is further supported by satellite images of sea surface temperature and chlorophyll-a concentration from that period (Fig. 17), which show filaments spiraling toward the eddy's center, indicating surface convergence consistent with theoretical models of anticyclonic eddy growth (Sangrà et al., 2005, 2007). By contrast, the PUMP eddy showed no evidence of proximity to, or interaction with, upwelling waters along its trajectory prior to sampling (see Fig. 1 in Barceló-Llull et al., 2017).

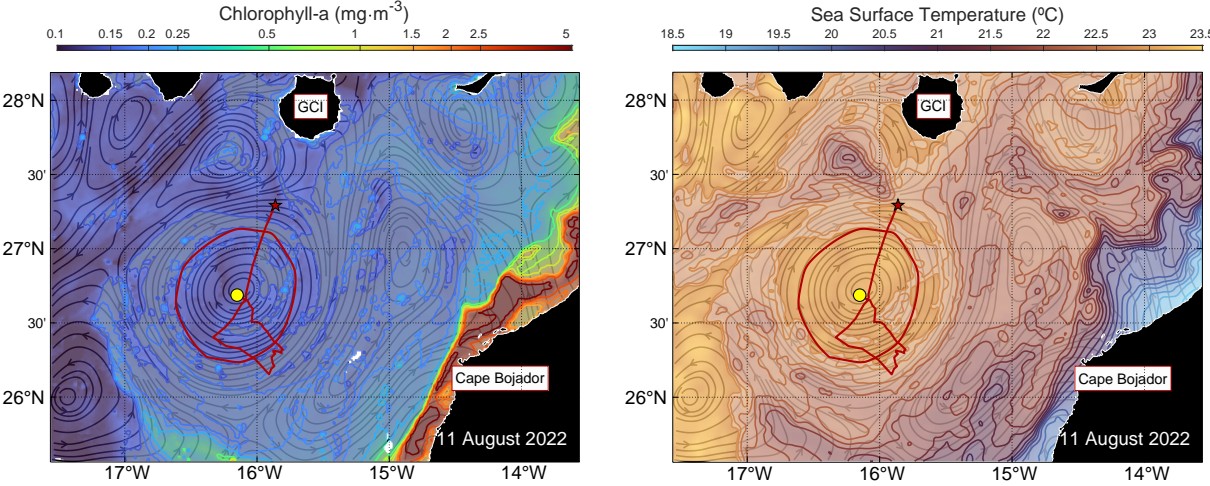

**Figure 17.** Interaction between coastal upwelling filaments and the mesoscale eddy. The left panel shows chlorophyll-a concentration, and the right panel displays sea surface temperature, both overlaid with geostrophic streamlines derived from satellite altimetry (black arrows). The snapshot corresponds to 11 August 2022, a key moment during the eddy's intensification phase, illustrating its interaction with upwelling filaments (shaded area) along the northwestern African margin. The thick red line indicates the eddy's trajectory, with its position at birth marked by a red star and its position on the depicted date indicated by a yellow circle.

Interactions between mesoscale eddies, a common feature in the CEC, have been shown to influence eddy hydrography through straining, merging, or deflection processes (Sangrà et al., 2005; Rodríguez-Marroyo et al., 2011; Ruiz et al., 2014).

Although we do not present direct observational evidence of such interactions affecting Bentayga, a video derived from satellite altimetry reveals that the eddy remained in close proximity to multiple vortices throughout its lifetime (see Valencia and Aguiar-González, 2025). This persistent mesoscale crowding suggests that eddy–eddy interactions may have contributed to the observed structural complexity of Bentayga, distinguishing it from the more isolated evolution of the PUMP eddy.

The Bentayga eddy displayed a well-defined anticyclonic circulation from the shallowest VMADCP-recorded levels (30 and 40 m depth, depending on the sampling phase) down to approximately 250 m (see Figs. 4a, 8, 9a, and B1a–b). Below this depth, horizontal sections of the objectively interpolated velocity field during the eIMPACT2 SeaSoar phase revealed increasingly complex circulation patterns, illustrated in Figs. B1 and B2. At 328 m (Fig. B1c), the core appeared elongated, with the minimum azimuthal velocities slightly displaced northward, whereas at 408 m (Fig. B1d), the circulation weakened but retained spatial coherence, now with the core slightly displaced to the south. At greater depths, the anticyclonic signal became progressively distorted and spatially irregular: although some rotational features persisted (Fig. B2), they were no longer dominant. These changes may reflect the increasing influence of other dynamical processes—such as ambient shear or nearby mesoscale structures (see Supplementary Note 6; Valencia and Aguiar-González 2025)—that may contaminate the eddy-induced circulation below approximately 330 m. Some of the observed features could also stem from limitations in the interpolation procedure. Additionally, previous studies in the CEC have reported the trapping of near-inertial waves at the base of intrathermocline eddies (Martínez-Marrero et al., 2019), which may dominate current variability at depth. The proximity of Bentayga to the Canary Archipelago, nearby seamounts, and the northwestern African continental margin (Fig. 1) could further favor the incidence of super-inertial internal waves (e.g., semidiurnal tides), potentially modulating circulation and hydrographic structure at these depths.

This observed weakening of the eddy-induced anticyclonic circulation below $\sim$330–350 m has direct implications for the vertically integrated diagnostics. While the correlation between the observed azimuthal velocity profiles and their solid-body fits ($r_{xy}$) remains relatively high ($\geq$0.9) down to 350 m, it decreases and fluctuates below this level, indicating reduced rotational coherence (see Sect. 3.5). However, the final vertical integration limit was not based on rotational coherence alone. Instead, it was defined dynamically as the depth at which the ratio between the shell velocity and the eddy's translational speed drops below unity. This approach captures the level beyond which the eddy can no longer coherently advect its inner properties as it propagates. Accordingly, a vertical limit of 550 m was used for integrations, balancing physical relevance, vertical coverage, and robustness against sampling-induced distortion. Reducing this limit to 340 m leads to differences of about 20% in APE, AHA, and ASA estimates, while KE remains comparatively less sensitive. These findings justify the use of the full 550 m depth for integration, provided that the results are interpreted as energetically conservative yet dynamically meaningful estimates of the eddy's influence. These complexities also highlight the importance of employing advanced interpolation techniques to accurately extract and characterize the signal of the eddy from the observed current and hydrographic data. Addressing these limitations and separating external influences would require a more detailed analysis, which is beyond the scope of this study (e.g., Candela et al., 1992).

The geometry of the Bentayga eddy is broadly consistent with previously documented AEs in the Macaronesian region (Table 3). However, its horizontal and vertical dimensions place it among the largest observed eddies in this area. With an

external radius of 70 km and a vertical extent of 550 m, Bentayga ranks within the upper 20th percentile in both parameters. Moreover, its core—characterized by solid-body rotation—exhibits a radius of ∼23 km and a vertical aspect ratio ($H/L_x$) of 0.012, substantially higher than the regional median (0.0033). This dual radial structure, composed of a compact and vertically coherent core surrounded by an extended ring, reinforces its classification as a mature and dynamically robust intrathermocline eddy. It is also noteworthy that, among the studies analyzed in Table 3, only Barceló-Llull et al. (2017) explicitly distinguishes between the core and the surrounding ring in their structural estimates. In contrast, the remaining studies typically report a single radius value, without differentiating between the inner and outer components of the eddy. Finally, the energetics of the Bentayga eddy support this interpretation: its energy Burger number ($B_E < 1$), defined as the ratio of KE to APE, is consistent with values expected for large, baroclinically dominated vortices (Table 1).

Beyond its structural properties, the eddy's energy distribution offers additional insight into its dynamical regime and maturity when compared to other ITEs (Table 2). Across the three integration radii considered (core radius $R_c$, radius of maximum azimuthal velocity $R_{max}$, and external radius $R_e$), Bentayga consistently exhibits a dominance of APE over KE, with corresponding $B_E$ of 0.21, 0.44, and 0.69, respectively (Table 1). These values suggest a baroclinically dominated structure, particularly within the solid-body core and middle ring, and are well aligned with expectations for mature ITEs. When compared to other observed AEs, Bentayga eddy's energetics differ markedly from those of the PUMP eddy (Barceló-Llull et al., 2017), which presented $B_E \sim 1.6$—indicating an energy balance tilted toward kinetic energy, consistent with a younger and more dynamically active eddy (McWilliams, 1985). Likewise, the meddy reported by Schultz Tokos and Rossby (1991) exhibited a $B_E \sim 1.05$, close to unity. In contrast, Bentayga eddy's core-scale value of $B_E = 0.21$ underscores its high APE content, consistent with an evolved, stratified structure that stores more energy in the vertical displacement of isopycnals than in rotational motion. The increase in $B_E$ observed at larger integration radii reflects the contribution of the outer ring to the total KE budget. However, this value ($B_E = 0.69$ at $R_e$) should be interpreted with caution, as the outer regions of the eddy are more sensitive to sampling-induced distortions and possible contamination from ambient flow, as detailed in Supplementary Notes 1-4.

In addition to its baroclinic character, the eddy's momentum structure reveals further dynamical complexity. The relatively strong anticyclonic circulation responsible for the observed KE levels included a non-negligible ageostrophic component. These velocities appear to act as a compensating response to the eddy's curved geometry, contributing approximately 10–20% of the total observed flow. The effect of curvature, along with the resulting ageostrophic velocities, suggests that the momentum balance within Bentayga is best described in cyclogeostrophic terms. A similar situation arises in the PUMP eddy, for which the estimated cyclogeostrophic Rossby number ranges from 0.17 to 0.26, indicating that both intrathermocline eddies exhibit comparable momentum balances. While such a regime is often expected in submesoscale vortices (McWilliams, 1985), increasing evidence indicates that mesoscale eddies can also exhibit significant ageostrophic velocity components arising from net centripetal forcing (e.g., Ioannou et al., 2019; Penven et al., 2014; Douglass and Richman, 2015; Kunze, 1986; Joyce et al., 2013). Taken together, these results reinforce the view that anticyclones in the CEC frequently deviate from strict geostrophic balance, thereby justifying the use of cyclogeostrophic corrections such as those proposed by Uchida et al. (1998), Penven et al. (2014), and Ioannou et al. (2019).

**Table 3.** General properties of previously studied anticyclonic eddies (AEs) in the Macaronesian region. The Table summarizes key characteristics of AEs reported in various studies, including the eddy type, study location, radius (R), vertical extent (H), vertical aspect ratio (H/L$_X$), temperature anomaly ($\Theta'$), salinity anomaly ($S_A'$), vertical displacement of a characteristic isopleth (h'), water mass types in the AE core (WM), and Rossby number (Ro = $\zeta/f$). For ensemble averages, properties represent statistical means, whereas for individual AEs, the specific values are provided. Where applicable, vertical displacement (h') values are listed without parentheses for sinking isopleths, while values in parentheses represent shoaling isopleths. Mean, standard deviation, mode, median, and 80th percentile values for all listed studies are included, along with the Bentayga eddy's* for comparison. See the respective references for additional details on study-specific methods and results.

| Study (authors) | Location | AE type | R (km) | H (m) | H/L$_X$ | $\Theta'$ (°C) | $S_A'$ (g kg$^{-1}$) | h' (m) | WM | Ro |
|---|---|---|---|---|---|---|---|---|---|---|
| Armi and Zenk (1984) | Canary Basin | Subsurface eddy from in situ observations | 50.0 | 900 | 0.0090 | 2.50 | 0.800 | - | MW | - |
| Aristegui et al. (1994) | Lee of Canary Archipelago | Surface eddy from in situ observations | 31.0 | 400 | 0.0065 | 1.00 | - | 45 | - | - |
| Pingree (1996) | Northeastern Atlantic Ocean | Subsurface eddy from in situ observations | 42.0 | 300 | 0.0014 | 1.40 | 0.350 | 80 (40) | - | -0.3 |
| Sangrà et al. (2009) | Canary Eddy Corridor | Eddy Demography based on satellite altimetry | 50.0 | 300 | 0.0030 | - | - | - | - | - |
| Caldeira et al. (2014) | Lee of Madeira Island | Surface eddy from satellite and in situ observations | 25.0 | 300 | 0.0060 | - | - | 100 | - | -0.7 |
| Ruiz et al. (2014) | Off Cape Bojador | Surface eddy from satellite and in situ observations | 62.5 | 250 | 0.0020 | 0.80 | 0.200 | 40 | ENACW | - |
| Pegliasco et al. (2015) | Canary Upwelling System | Ensemble mean eddy based on satellite altimetry and Argo observations | 52.0 | 325 | 0.0034 | 0.60 | 0.120 | - | - | - |
| Schütte et al. (2016) | Tropical northeastern Atlantic | Ensemble mean subsurface eddy based on satellite altimetry and Argo observations | 52.0 | 325 | 0.0034 | -4.00 | 0.720 | 52 (48) | SACW | -0.7 |
| Karstensen et al. (2017) | Tropical northeastern Atlantic | Subsurface eddy from in situ observations | 30.0 | 200 | 0.0033 | -2.00 | -0.600 | 50 (20) | SACW | -0.6 |
| Barceló-Llull et al. (2017) | Canary Eddy Corridor | Subsurface eddy from in situ observations | 46.0 | 500 | 0.0054 | 1.50 | 0.300 | 150 (40) | MMW | - |
| Kolodziejczyk et al. (2018) | Off Cap-Vert Peninsula | Subsurface eddy from in situ observations | 100.0 | 600 | 0.0030 | - | - | 40 (40) | SACW (NACW) | - |
| Estrada-Allis et al. (2019) | Canary Eddy Corridor | Surface eddy from a regional simulation | 40.9 | 800 | 0.0098 | - | - | 45 | SACW (NACW) | -0.7 |
| Cardoso et al. (2020) | Cabo Verde | Eddy demography based on satellite altimetry | 50.0 | - | - | - | - | - | - | - |
| Ioannou et al. (2022) | Northern Canary Current Upwelling System | Eddy demography based on satellite altimetry and Argo observations | 56.6 | 120 | 0.0011 | 0.45 | 0.140 | - | - | - |
| Dilmahamod et al. (2022) | Mauritanian Upwelling region | Subsurface eddy from a regional simulation | 31.6 | 300 | 0.0047 | -4.50 | -0.800 | 40 (50) | SACW | -0.7 |
| | Mean | | 53.8 | 414 | 0.0043 | | | | | -0.6 |
| | Standard deviation | | 29.9 | 182 | 0.0025 | | | | | 0.2 |
| | Mode | | 50.0 | 300 | 0.0030 | | | | | -0.7 |
| | Median | | 48.0 | 300 | 0.0033 | | | | | -0.7 |
| | 80th percentile | | 62.5 | 520 | 0.0064 | | | | | -0.7 |
| | Bentayga | | 70.0 (23.0) | 550 | 0.0039 (0.0120) | 1-4 | >0.1 | 75 (10) | ENACW | -0.5 (-0.7) |

*The values presented for Bentayga eddy are relative to its R$_e$ (R$_c$)

The core of the Bentayga eddy displayed positive temperature and salinity anomalies, with the exception of the 80–120 m layer, where negative salinity anomalies were detected (Figs. 7 and 16). Overall, these anomalies slightly exceeded those previously reported for AEs in the Macaronesian region (Table 3). Rather than indicating the presence of a distinct water mass, they are consistent with the typical properties of ENACW, which dominate the thermocline in this region. This contrasts with the PUMP eddy, whose core was composed of MMW (Barceló-Llull et al., 2017), and with subsurface ITEs formed near the Mauritanian coast, which may transport South Atlantic Central Water into open-ocean areas influenced by North Atlantic Central Water (Schütte et al., 2016; Karstensen et al., 2017). The volume, heat, salt, and equivalent freshwater fluxes associated with the Bentayga eddy's anomalies are consistent with values reported for other AEs in the CEC (Sangrà et al., 2009; Barceló-Llull et al., 2017), as well as in other EBUS (Chaigneau et al., 2011; Dilmahamod et al., 2018) and western boundary current regions (Dilmahamod et al., 2018). Its subsurface intensification—typical of ITEs—reflects its capacity to effectively act as a conduit for transferring thermohaline anomalies into the ocean interior along thermocline depths.

Satellite observations suggest that interactions with upwelling filaments extending from the northwestern African coast (Fig. 17) may have contributed to the thermal and saline properties observed in the Bentayga eddy. Although our data do not directly capture filament entrainment into the eddy's core, the visual evidence of filament convergence towards the eddy center supports the hypothesis that such incorporation could have occurred. As the eddy migrated offshore, these entrained waters may have progressively deepened within the eddy structure, driven by isopycnal adjustment and potential vorticity conservation, as described for similar processes in previous studies (e.g. D'Asaro, 1988; Pingree, 1996). This behavior is reminiscent of the subduction-driven evolution proposed for a shallow subtropical subducting westward-propagating eddy ("Swesty") (Pingree, 1996), although a definitive classification for the Bentayga eddy cannot be established with the available observations. If so, it could have contributed to the injection of coastal upwelling waters and their associated properties into the ocean interior.

Beyond their physical transport of heat and salt, ITEs such as the Bentayga eddy play a pivotal role in biogeochemical exchanges by potentially delivering nutrient-rich upwelling waters to oligotrophic regions. These processes can stimulate offshore primary productivity, highlighting the ecological importance of ITEs in shaping regional ocean dynamics (e.g., Hormazabal et al., 2013; Thomsen et al., 2016; Cornejo et al., 2016; Karstensen et al., 2017; Bosse et al., 2017; Cerdán-García et al., 2024). In the case of the Bentayga eddy, our in situ observations of dissolved oxygen and AOU reveal a subsurface oxygen-deficient core with relatively low AOU values—features consistent with the recent incorporation of Saharan upwelling waters during the eddy's intensification, thus reinforcing its potential biogeochemical impact on the open ocean interior.

Extrapolating observed eddy-driven fluxes to estimate annual eddy-driven transport is a common practice once the average frequency of eddy formation in a region is established. For instance, using satellite altimetry and previous estimates, Sangrà et al. (2009) calculated an annual volume transport of 1.3 Sv by long-lived eddies in the CEC, representing approximately one-quarter of the Canary Current's total annual transport, based on an average of 17 eddies per year. However, the capacity of eddies to transport heat, salt, and biogeochemical properties can vary widely depending on their lifespan, size, intensity, and interactions with surrounding waters. Applying transport values from the Bentayga eddy to the broader context of the CEC presents certain challenges. Nevertheless, its distinct trajectory and unique characteristics underscore its significant contribution to understanding eddy variability in the region. An analysis of the climatological eddy detection atlas (META3.2 DT *allsat*;

Aviso+, 2022), spanning 1993–2022, identified only six AEs with trajectories resembling the one of the Bentayga eddy's, each remaining near the continental shelf for over 10 days. This rare behavior positions this ITE as a valuable case for investigating eddy-driven transport, offering new insights into the dynamics that facilitate the movement of coastal upwelling waters into the ocean interior.

During the eIMPACT2 campaign, the Bentayga eddy was flanked by two cyclonic eddies CEs, consistent with a generally crowded mesoscale environment. A video synthesis of daily altimetry from June to November 2022 (Valencia and Aguiar-González, 2025) reveals that Bentayga remained in close proximity to multiple anticyclonic and cyclonic vortices throughout its trajectory across the CEC. This persistent mesoscale crowding supports our hypothesis that eddy–eddy interactions may have contributed to the structural variability observed within the Bentayga eddy. Such interactions—via straining, merging/stacking, or filament exchange—can modify the hydrographic properties of the eddy core and enhance lateral mixing (e.g., de Marez et al., 2020; Rykova and Oke, 2022; Fu et al., 2023; Barboni et al., 2023; Garreau et al., 2018; Ruiz et al., 2014). These processes increase the likelihood that coastal water masses, once entrained, can be subducted and advected into the ocean interior, potentially amplifying the transport efficiency of coherent eddies such as Bentayga.

The findings presented in this study offer a comprehensive understanding of the physical characteristics—such as hydrography and dynamical properties—of the Bentayga eddy and its role within the CEC as an ITE. This eddy demonstrated the capacity to transport substantial volumes of water, heat, and salt. Its subsurface core, characterized by oxygen-deficient waters with relatively low AOU, further suggests a recent biogeochemical imprint from upwelling filaments that were likely entrained and transported during its intensification. However, the marked differences observed when compared to other ITEs, such as the PUMP eddy, highlight that each eddy may exhibit distinct features that critically shape its transport dynamics and regional influence. These differences underscore the need to study individual eddies in detail to fully capture their specific behaviors and contributions. While broader patterns can be inferred through the integration of satellite and in situ observations (e.g., Sangrà et al., 2009; Chaigneau et al., 2009; Dong et al., 2014; Pegliasco et al., 2015; Ioannou et al., 2022), high-resolution and quasi-synoptic field surveys—such as those achieved during specific phases of the eIMPACT2 campaign—are essential. These allow for the characterization of fine-scale physical structures while preserving the coherence of their spatial and temporal context, providing insights into the coupled physical–biogeochemical dynamics that unfold within eddies. Such detailed observations are fundamental to improving our understanding of the cumulative role of ITEs in modulating regional circulation and biogeochemical cycling.

## 5  Concluding remarks

This study provides a comprehensive analysis of the ITE Bentayga, observed in the CEC, highlighting its hydrographic structure, dynamical properties, transport characteristics, and broader implications for regional oceanography. Using high-resolution, multi-phase data collected during the eIMPACT2 campaign, the study emphasizes the eddy's complex structure, evolution, and interactions with surrounding mesoscale features.

The methodological approach adopted—combining satellite tracking with multiple observational phases and idealized zonation models—enabled us to characterize the eddy's three-dimensional structure, quantify its energetics, and assess eddy-driven volume and thermohaline fluxes. The application of different integration limits, supported by a detailed analysis of its hydrography, rotational coherence and azimuthal velocity patterns, further enhanced the robustness of our estimates. This multi-faceted strategy provides a valuable blueprint for future ITE studies in other regions.

The Bentayga eddy exhibited a compact solid-body core surrounded by a wide outer ring of lower coherence, together transporting substantial volumes of water, heat, and salt. Its subsurface intensification, along with the presence of oxygen-deficient waters with low AOU, suggests a recent biogeochemical imprint from coastal upwelling filaments that were likely entrained during its intensification in August 2022. Compared to other regional ITEs—such as the PUMP eddy—Bentayga exhibited notable structural and dynamical differences, including a single-core configuration and distinct hydrographic anomalies. These differences reinforce the need for case-specific studies to fully understand ITE variability and assess their impact.

This work also underscores the sensitivity of eddy diagnostics to the temporal coherence of the observations. Although the eIMPACT2 campaign aimed to achieve quasi-synoptic sampling, not all phases met this objective. The resulting spatial and temporal misalignments were addressed through comparative analyses and conservative integration strategies, but they highlight the need for improved sampling designs to minimize distortion in future eddy surveys.

The present study contributes valuable insights to the growing body of research on mesoscale eddies, offering a detailed case study of an intrathermocline eddy and its role in regional oceanic processes. However, it also highlights the limitations of generalizing eddy-driven transport based on a single event. Capturing the full complexity of ITE dynamics—particularly their interactions with surrounding mesoscale features and external forcings such as upwelling filaments—requires extended high-resolution observational efforts and the support of advanced numerical modeling. Future efforts should prioritize understanding the variability of these structures across broader temporal and spatial scales, with particular attention to the mechanisms driving their intensification and the long-term consequences for ocean circulation, heat and salt budgets, and ecosystem functioning.

**Appendix A: Wavelet analysis of the Lagrangnian evolution of the eddy geometric properties of the Bentayga eddy**

As previously seen, in Sect. 3.1 (Fig. 3), the geometric properties of the Bentayga eddy reveal significant variability throughout its life cycle, spanning from June 23, 2022, to June 18, 2023. To further investigate the temporal variability and dominant frequencies, a wavelet analysis was conducted (Fig. A1) (Torrence and Compo, 1998). In this study, a Morlet wavelet was employed, which is well-suited for detecting oscillatory signals and provides good resolution in both time and frequency domains. The dimensionless frequency used was six, which balances the trade-off between time and frequency resolution. The analyzed periods ranged from 4 to 350 days, enabling the detection of both long-term signals and short-term variations.

Figures A1a-c present the wavelet power spectra for amplitude, speed, and radius detrended standardized anomalies, with solid black contours indicating the 90% and 95% significance levels. These significance levels were determined through a Monte Carlo simulation with 10,000 iterations using white noise as a reference for identifying statistically significant fluctuations. The dashed black and white contour represents the cone of influence, which indicates the regions where edge effects

reduce the reliability of the results. The global wavelet power spectra, averaged over the entire time series, are shown in
Fig. A1d.

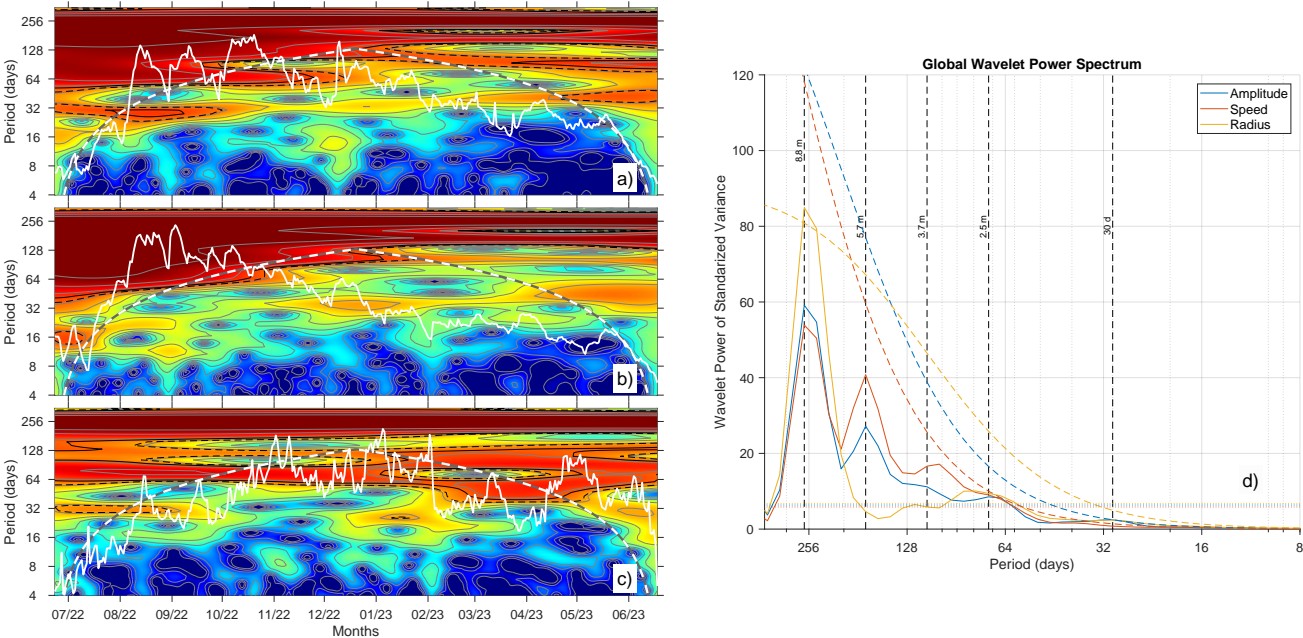

**Figure A1.** Wavelet analysis of altimetric features exhibited by the Bentayga eddy from the META3.2 NRT product. Panels (a) to (c) display the wavelet power spectrum (colormap) for the detrended standardized anomalies of amplitude, swirl speed, and radius, with thick white lines indicating the corresponding anomalies. Panel (d) illustrates the global wavelet power spectrum, averaged over time. Dashed black/white contours in panels (a) to (c) represent the cone of influence. Also, solid and dashed black contours indicate the 90% and 95% significance levels, determined through a Monte Carlo experiment with 10,000 iterations. Dashed (dotted) lines in panel (d), indicate the 95% red (white) noise levels.

The wavelet analysis reveals several dominant periods in all three properties, with significant variability at both low and high frequencies. Remarkably, a strong periodic signal at approximately 256 days (∼8.8 months) is observed for speed and amplitude, corresponding to large-scale seasonal variability during the formation, growth, maturity, and the slow decay phases of the Bentayga eddy, as seen in Fig. 3. The radius shows a similar periodicity, though the peak is less pronounced. In addition,
a secondary peak is observed at shorter periods, around 64 days (∼2 months), indicating higher-frequency fluctuations likely linked to shorter-scale dynamical processes, such as interactions with other mesoscale structures or variations in external forcing (e.g. wind stress). These significant fluctuations, particularly for radius and speed, suggest that the Bentayga eddy experienced intermittent episodes of intensification and relaxation, as reflected in the variability shown in Fig. 3.

The initial rapid growth and transition into the mature phase are captured by shorter-period fluctuations in Figs. A1a-c,
where the wavelet power shows more localized variability. However, during the slow decay and the eventual rapid dissipation

phase, particularly post-May 2023, the wavelet analysis, using the Morlet wavelet optimized for periodic signals, struggles to resolve these processes fully. This is especially evident in the wavelet's difficulty in resolving the eddy's rapid decay phase. The inherent shape and resolution of the chosen wavelet may smooth out sharp transitions associated with the dissipation of the Bentayga eddy, blending them into longer-period oscillations. Therefore, while the wavelet analysis offers important insights,
complementary methods may be needed to fully capture the complete life cycle of the eddy, particularly the rapid transitions at both ends of its evolution.

In summary, the combined results from the time series and wavelet analyses provide a comprehensive picture of the life cycle of the Bentayga eddy, characterized by alternating periods of intensification and decay. External factors, such as interactions with the continental margin and surrounding mesoscale features, likely contributed to these fluctuations, particularly during
the early and mature phases. These findings emphasize the importance of multi-scale variability in the life cycle of mesoscale eddies and underscore the need for further investigation into the mechanisms driving these processes.

## Appendix B:  Depth-resolved VMADCP velocity field during eIMPACT2 SeaSoar phase

Figures B1 and B2 present horizontal sections of the velocity field recorded by the VMADCP during the eIMPACT2 SeaSoar phase. The figures group specific depth ranges, with Fig. B1 covering depths from 104 m to 408 m and Fig. B2 showing depths
from 472 m to 744 m. These sections illustrate the depth-dependent structure of the circulation within the Bentayga eddy, as well as the potential influence of neighboring eddies.

In the shallow layers (104 m to 248 m), a well-defined and coherent anticyclonic circulation is observed. Between 328 m and 408 m, the eddy core becomes elongated, and signs of a possible bifurcation into dual nuclei are noted at 328 m. This structural change may have resulted from internal deformation of the eddy or interpolation artifacts during data processing. At greater
depths (472 m to 744 m), the circulation weakens significantly and becomes increasingly irregular. Although anticyclonic features persist, the influence of external mesoscale eddies appears to dominate, disrupting the eddy's coherence.

To complement the interpolated fields shown in Figs. B1 and B2, this appendix also presents the original VMADCP velocity observations recorded during the eIMPACT2 SeaSoar phase (Figs. B3 and B4). Horizontal sections at the same depth levels are displayed as scatter plots using identical color palettes and velocity magnitude limits. These figures allow for direct visual
comparison between the objectively interpolated fields and the underlying measurements. By overlaying the original ship's sampling trajectory and retaining consistent color scaling, these plots provide context for assessing the spatial coverage and reliability of the interpolated fields used in the main analysis.

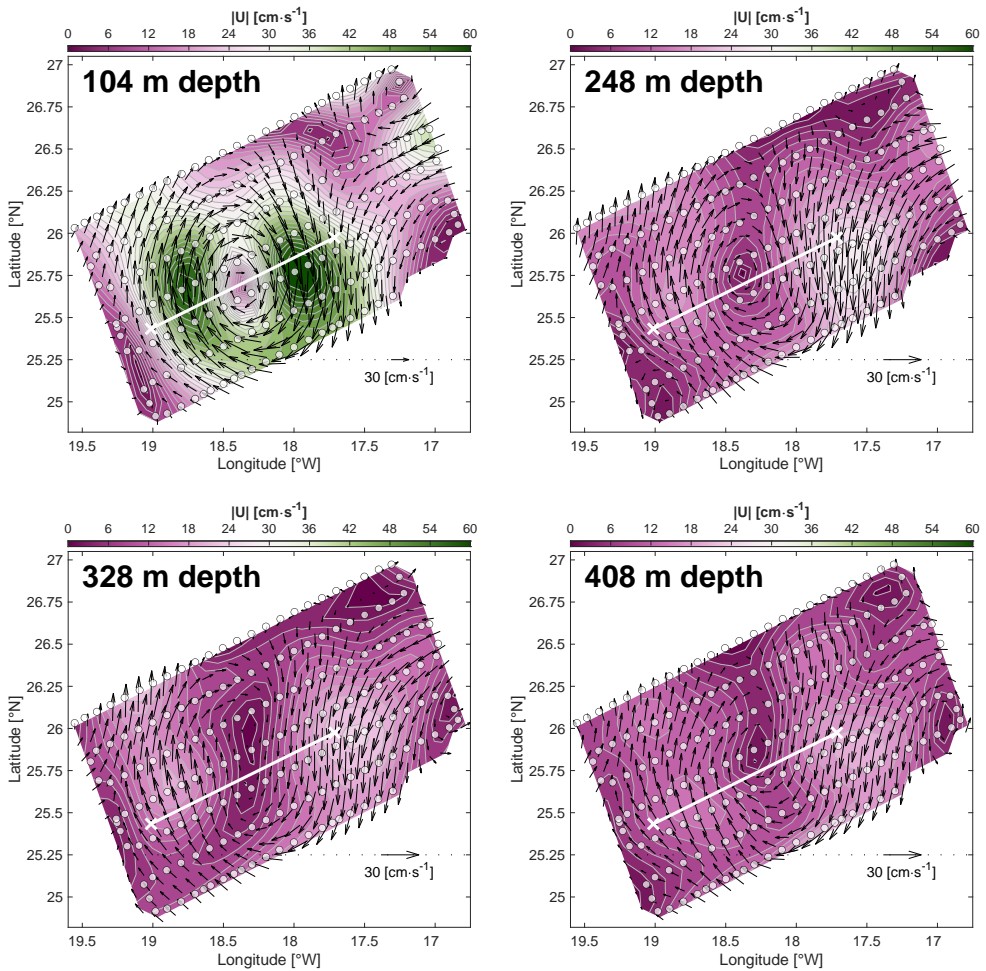

**Figure B1.** Objectively interpolated VMADCP velocity vectors (black arrows) superimposed on its magnitude contours at various depths: 104, 248, 328, and 408 m. Each panel uses a consistent color scale and speed range with 30 contour levels to highlight the velocity structure. Velocity vectors are scaled proportionally to the respective depth (see legend in the bottom right corner of each panel). The ship's grid-like sampling trajectory during the eIMPACT2 SeaSoar phase (pale white circles) and the virtual transect used for the vertical section in Fig. 9 (solid white line) are also indicated. Objective interpolation correlation scales were set to $L_x = L_y = 44$ km, with 3% uncorrelated noise applied.

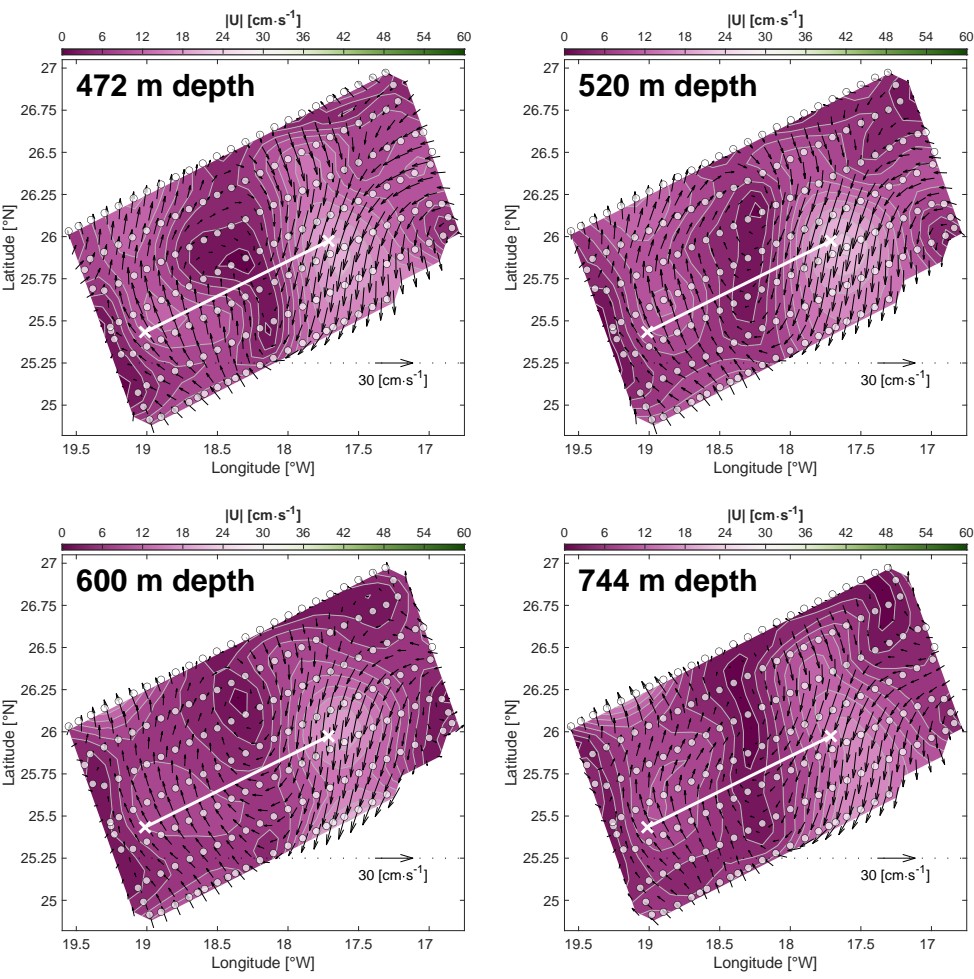

**Figure B2.** Same as Fig. B1 but for depths: 472, 520, 600, and 744 m.

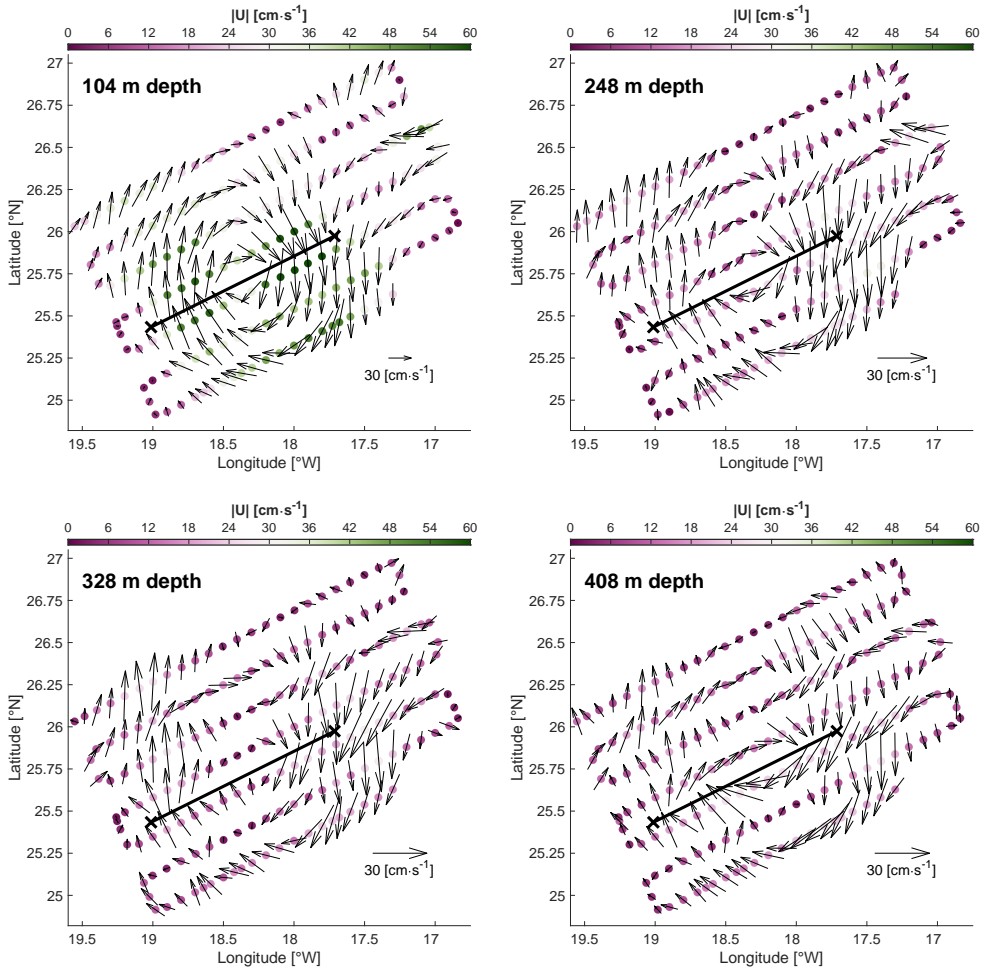

**Figure B3.** Original VMADCP velocity vectors (black arrows) superimposed on velocity magnitude from ungridded measurements at the same depth levels showed in Fig. B1, enabling visual comparison with the interpolated fields. Velocity vectors are scaled proportionally to the respective depth (see legend in the bottom right corner of each panel). The virtual transect used for the vertical section in Fig. 9 is also shown (solid black line).

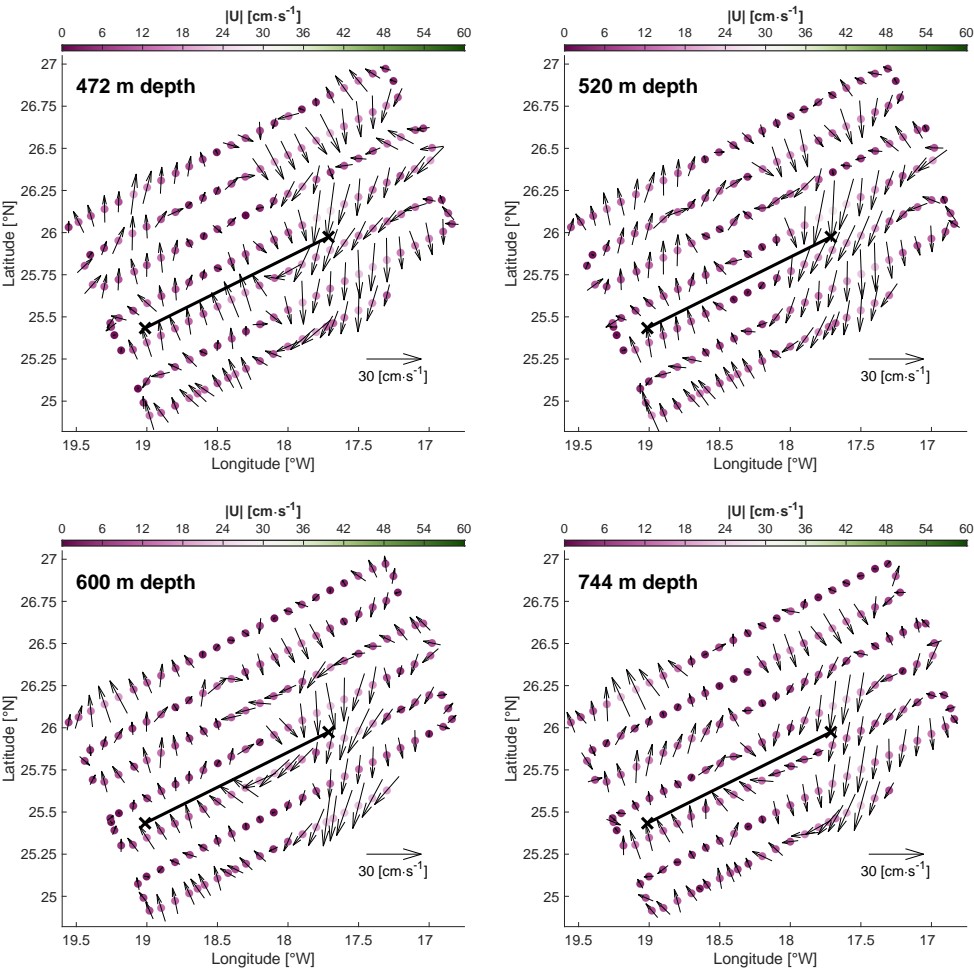

**Figure B4.** Same as Fig. B3 but for the depths showed in Fig. B2.

*Code availability.* The software codes used for the analyses in this study, written in MATLAB, are available upon request from the corresponding author. The specific MATLAB version used for the analyses is R2023b.

*Data availability.* All in situ datasets used in this study are publicly available on Zenodo and can be accessed at https://doi.org/10.5281/zenodo.14911706. The META3.2exp NRT and DT *allsat* products are freely available from AVISO+ at https://www.aviso.altimetry.fr/. The DT *allsat* dataset can be accessed via DOI: https://doi.org/10.24400/527896/a01-2022.005.220209, while the NRT product does not have an associated DOI. The European Seas Gridded L4 Sea Surface Heights and Derived Variables NRT altimetry product (https://doi.org/10.48670/moi-00142) and the Global Ocean Colour (Copernicus-GlobColour) Bio-Geo-Chemical Level 3 product (https://doi.org/10.48670/moi-00280) are publicly available through the Copernicus Marine Service (CMEMS) at https://marine.copernicus.eu/. The GHRSST Level 4 MUR Global Foundation Sea Surface Temperature Analysis product (https://doi.org/10.5067/GHGMR-4FJ04) is provided by PO.DAAC and can be accessed at https://podaac.jpl.nasa.gov/.

*Video supplement.* A video representation of the Bentayga eddy's life cycle is available at: https://doi.org/10.5446/69688 (Valencia and Aguiar-González, 2025). This animation, based on the daily altimetric product META3.2exp and geostrophic current fields derived from the 930 CMEMS near-real-time product `SEALEVEL_EUR_PHY_L4_NRT_008_060`, illustrates the eddy's evolution from its generation southwest of Gran Canaria Island in June 2022 to its structure during the eIMPACT2 survey in November. The video also reveals the persistent proximity of other mesoscale vortices along the eddy's trajectory, providing qualitative support for the hypothesis of eddy–eddy interactions within the Canary Eddy Corridor.

*Author contributions.* L.P.V.: Conceptualization, Investigation, Data Curation, Methodology, Software, Formal Analysis, Validation, Visu-935 alization, Writing – Original Draft, and Writing – Review & Editing. Á.R.S.: Conceptualization, Investigation, Methodology, Supervision, Validation, and Writing – Review & Editing. B.A.G.: Methodology, Resources, Software, Formal Analysis, Validation, and Writing – Review & Editing. J.A.: Conceptualization, Investigation, Resources, Project Administration, Funding Acquisition, and Writing – Review & Editing. X.A.Á.S: Conceptualization, Investigation, Resources, Project Administration, Funding Acquisition, and Writing – Review & Editing. J.C.: Investigation, Resources, and Writing – Review & Editing. M.D.G.C: Investigation, Resources, and Review & Editing. A.M.M.: 940 Conceptualization, Investigation, Data Curation, Methodology, Supervision, Validation, and Writing – Review & Editing.

*Competing interests.* The authors declare that they have no conflict of interest.

*Acknowledgements.* This research was funded by the e-IMPACT project (PID2019-109084RB-C21 and C22) led by J.A. and X.A.Á.S., supported by the Spanish government under the Ministry of Science and Innovation (MCIN/AEI/10.13039/501100011033). L.P.V. acknowledges financial support from the Chilean government through the BECAS CHILE ANID Doctorado en el Extranjero program (Grant No.

72210549) of the National Agency for Research and Development (ANID), under the Ministry of Science, Technology, Knowledge, and Innovation. L.P.V. also expresses gratitude to Bàrbara Barceló-Llull and Francisco Machín for their valuable feedback and insightful discussions throughout the development of this research. Special thanks are extended to the captain and crew of the R/V *Sarmiento de Gamboa*, as well as the Marine Technology Unit (UTM-CSIC) and the scientific staff for their dedicated support during the cruise. To enhance data visualization and promote accessibility for readers with color-vision deficiencies, this study utilized the Scientific color maps (Crameri,

2023) and `cmocean` (Thyng et al., 2016), alongside conventional MATLAB color palettes. These colormaps were selected to prevent visual distortion and improve inclusivity, following best practices in scientific visualization (Crameri et al., 2020). Acknowledgment is given to the use of ChatGPT, developed by OpenAI, for language proofreading and clarity enhancement during the preparation of an earlier draft of this manuscript. Finally, we sincerely thank Dr. Ilker Fer (editor), Dr. Anthony Bosse, and the anonymous reviewer for their constructive comments and thoughtful suggestions during the peer-review process, which significantly strengthened the quality and clarity of the

manuscript.

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
