# Peer review of "Mesoscale dynamics of an intrathermocline eddy in the Canary Eddy Corridor"

_EGUsphere, 2025_

## Author Comment (AC4)

**Assessment of Sampling Bias in the Bentayga Eddy: Technical Response to Reviewer Comments**

Luis P. Valencia et al.

April 2025

**1 Introduction**

During the eIMPACT2 survey, an intrathermocline eddy named *Bentayga* was extensively sampled aboard the R/V *Sarmiento de Gamboa*. The sampling strategy comprised several phases, two of which are included in the study presented by Valencia et al. [2025].

One of the main concerns raised by an anonymous referee (`https://doi.org/10.5194/egusphere-2025-99-RC3`) was the lack of synopticity in the dataset selected to evaluate the eddy's geometry and its derived properties—specifically, kinetic energy (KE), available potential energy (APE), available heat anomalies (AHA), and available salt anomalies (ASA).

Here, building on the framework proposed by Allen et al. [2001], we evaluate the degree of spatial distortion introduced during the sampling phase used for those calculations: the Oceanographic Transect (OceT) phase. The objective is to assess how accurately the eddy's true structure was captured under non-synoptic sampling conditions.

To this end, we compare the OceT results with quasi-synoptic sections that provide a more instantaneous view of the mesoscale structure. These comparisons, both visual and quantitative, help estimate the potential bias introduced by spatial distortion and are intended to directly address the methodological concerns raised during the review process.

**2 Description of the non- and quasi-synoptic phases**

**2.1 eIMPACT OceT phase**

The non-synoptic phase corresponds to the **eIMPACT OceT phase**, conducted between 19 and 28 November 2022. It consisted of a series of 26 oceanographic stations along a nearly zonal transect (OceT) that crossed the eddy from east to west (Fig. 1). At each station, vertical profiles of CTD-O and VMADCP were obtained. The stations were spaced approximately $12.7 \pm 2.8$ km apart, and the profiles extended from the surface to a depth of around 1500 m.

As seen in Fig. 1, some degree of distortion in the eddy's shape can be inferred from the sampling. This is explained by the fact that *Bentayga* was translating with an average velocity $c$ of $0.051 \pm 0.012$ m s$^{-1}$, while the R/V was moving in the same direction at a mean speed $v_v$ of $0.454 \pm 0.167$ m s$^{-1}$. As the anonymous referee pointed out, and as also discussed in Allen et al. [2001], this distortion resulting from the lack of synopticity may compromise the structural integrity of the eddy, introducing a significant bias in all calculations involving its spatial dimensions.

**2.2 eIMPACT SeaSoar phase**

The quasi-synoptic phase used in Valencia et al. [2025] corresponds to the **eIMPACT SeaSoar phase**, conducted during 9–14 November 2022. Continuous measurements were obtained using a towed CTD-O mounted on an undulating SeaSoar vehicle, combined with VMADCP observations. This phase employed a southwest–northeast sampling grid consisting of seven transects (each approximately 278 km long), spaced by ~22 km, and sampled vertical layers between 30 and 320 m depth (Fig. 2). However, this dataset was primarily used to construct an objectively interpolated three-dimensional field, rather than to evaluate the structural shape of *Bentayga*.

In the present report, we use two of the seven along-eddy transects, selecting those closest to the eddy's center. For consistency, we retain their original numbering, with transect T3 located to the south

and T4 to the north of *Bentayga*'s center. Transect T3 was conducted on 11 November, approximately between 01:00 and 17:00 UTC, with the R/V moving in the same direction as the eddy at a mean velocity of $v_v = 4.31 \pm 0.092$ m s$^{-1}$, while *Bentayga* exhibited a translational velocity of $c = 0.041$ m s$^{-1}$. T4 was recorded between 19:00 UTC on 11 November and 10:00 UTC on 12 November, during which the R/V moved against the eddy's direction at a speed of $v_v = 4.29 \pm 0.127$ m s$^{-1}$, while the eddy translated at a mean velocity of $c = 0.051 \pm 0.014$ m s$^{-1}$.

[Figure]

Figure 1: Study area showing the **eIMPACT OceT phase** sampling (19–28 November 2022). White stars indicate the 26 oceanographic stations where CTD-O and VMADCP profiles were collected. The thick dark red line traces the trajectory of *Bentayga* over time, with selected dates annotated. Colored dashed ellipses depict the eddy's perimeter on specific days, defined by the location of maximum circum-averaged velocity, and corresponding colored dots mark its center. Thin black contours represent isobaths every 500m, ranging from 500 to 5000m depth.

[Figure]

Figure 2: Same as Fig. 1, but for the spatial coverage during the **eIMPACT SeaSoar phase** (9–14 November 2022). Thick black lines indicate the grid-like trajectory of the R/V as it towed the undulating SeaSoar vehicle and simultaneously recorded VMADCP data. Diamond markers highlight the paths of transects T3 and T4, both colored by date

**2.3   eIMPACT Ortho-transects phase**

In addition to the main survey phases, a distinct configuration of two orthogonal transects—hereafter referred to as the **ortho-transects**—was conducted across *Bentayga*, forming a cross-shaped sampling pattern centered on the eddy (Fig. 3). These transects were carried out immediately after the end of the **eIMPACT SeaSoar phase**, between 02:00 UTC on 14 November and 03:00 UTC on 15 November.

During the **eIMPACT ortho-transects phase**, only VMADCP currents were continuously recorded. Since both transects intersected the eddy close to its center, they were used to evaluate the radial structure of its azimuthal velocities and to compare it with that obtained during the **eIMPACT OceT phase** [Valencia et al., 2025].

The transects—hereafter named the Zonal Transect (ZT) and the Meridional Transect (MT)—crossed the eddy from west to east and from south to north, respectively. The ZT was conducted on 14 November 2022 between 02:00 and 08:00 UTC, spanning a total distance of 116.07 km. During this period, the R/V moved at an average speed ($v_v$) of $4.60 \pm 0.22$ m s$^{-1}$, opposite to the eddy's propagation direction, which exhibited a translational speed ($c$) of 0.0704 m s$^{-1}$. The MT was carried out between 19:00 UTC on 14 November and 03:00 UTC on 15 November, covering 122.43 km from south to north. In this case, the vessel moved at a similar speed of $4.66 \pm 0.24$ m s$^{-1}$, approximately orthogonal to the eddy's translational path.

[Figure]

Figure 3: Same as Fig. 1, but for the **eIMPACT ortho-transects phase** (14–15 November 2022). The color-coded circles represent the full trajectory of the R/V, with colors indicating hours elapsed since the beginning of the phase (14 November at 02:00 UTC). Stars and diamonds mark the segments defining ZT and MT, which were selected for the analysis presented in this study.

**3 Mathematical Formulation**

Equation (10) from Allen et al. [2001] describes the effect of asynoptic sampling on the apparent wavelength of a propagating signal $m$, as observed from a moving vessel:

$$\lambda' = \lambda \left(1 - \frac{c}{v_v}\right)^{-1}$$

where:

- $\lambda$ is the true spatial wavelength of the feature (e.g., the intrinsic horizontal scale of the eddy),

- $c$ is the propagation speed of the feature (e.g., the eddy's translational velocity),

- $v_v$ is the vessel speed along the same axis as the feature's motion,

- $\lambda'$ is the apparent wavelength inferred from the moving platform.

This relationship captures a Doppler-like distortion introduced by the relative motion between the sampling platform and the moving feature. When the vessel travels in the same direction as the propagating feature ($c > 0$), the observed wavelength increases ($\lambda' > \lambda$); conversely, if the vessel moves against the propagation direction, the observed wavelength is shortened ($\lambda' < \lambda$).

This Doppler-induced deformation can significantly affect the interpretation of mesoscale features, particularly when diagnosing their spatial structure or computing derived quantities that depend on horizontal gradients.

To formalize this distortion, we define a **Doppler Factor** $D$:

$$D = 1 - \frac{c}{v_v^{(\parallel)}}$$

where:

- $c > 0$ is the propagation speed of the feature, defined as positive in a fixed reference direction,

- $v_v^{(\parallel)}$ is the component of the vessel's velocity along the same direction.

Depending on the sign of $v_v^{(\|)}$, the Doppler Factor can take two regimes:

- $v_v^{(\|)} > 0$: vessel moves in the same direction as the propagating feature (downstream),

- $v_v^{(\|)} < 0$: vessel moves in the opposite direction (upstream).

The Doppler Factor modifies the apparent wavelength as:

$$\lambda' = \frac{\lambda}{D} = \lambda \left( 1 - \frac{c}{v_v^{(\|)}} \right)^{-1}$$

Thus:

**Downstream sampling:** $D < 1 \Rightarrow \lambda' > \lambda$. The structure appears elongated.

**Upstream sampling:** $D > 1 \Rightarrow \lambda' < \lambda$. The structure appears compressed.

**Effective along-propagation velocity in cross-stream transects**

In cases where the vessel traverses the feature along a path that is not aligned with its direction of propagation (e.g., a cross-stream transect), the vessel's effective speed in the direction of the feature's movement must be projected onto the propagation axis. As shown in Equation (12) of Allen et al. [2001], this projection can be expressed as:

$$v_f = v_s \frac{S}{S + \Lambda}$$

Where:

- $v_v$ is the vessel's speed along the actual sampling path (e.g., the cross-stream transect),

- $v_f \equiv v_v^{(\|)}$ is the effective component of the vessel's speed projected along the direction of the feature's propagation,

- $S$ is the track leg separation,

- $\Lambda$ is the length of each cross front leg of the vessel's track.

This expression allows us to convert a cross-stream transect velocity $v_v$ into an equivalent along-stream velocity $v_f$, which can then be inserted into the Doppler factor and apparent wavelength expressions defined previously.

In our specific case, the vessel executed a single, long transect nearly orthogonal to the direction of the eddy's propagation. This situation corresponds to the limit where the cross-track leg length $\Lambda$ is much greater than the leg separation $S$, i.e., $S \ll \Lambda$. Under this condition, the projection of the vessel's speed onto the eddy's propagation direction—used to compute the Doppler factor—can be expanded as follows.

Starting from the projection formula given by Equation (12) of Allen et al. [2001]:

$$v_f = v_v \cdot \frac{S}{S + \Lambda}$$

we introduce the small parameter $\epsilon = \frac{S}{\Lambda} \ll 1$, and rewrite the expression as:

$$v_f = v_s \cdot \frac{\epsilon}{1 + \epsilon}$$

Expanding in powers of $\epsilon$ yields:

$$v_f \approx v_v \cdot \left( \epsilon - \epsilon^2 + \epsilon^3 - \dots \right) = v_v \cdot \left( \frac{S}{\Lambda} - \left( \frac{S}{\Lambda} \right)^2 + \left( \frac{S}{\Lambda} \right)^3 - \dots \right)$$

Since $S \to 0$ in our case, this shows that:

$$v_f \to 0 \quad \Rightarrow \quad D = 1 - \frac{c}{v_f} \to -\infty$$

That is, the **effective along-propagation velocity becomes negligible**, which implies a **strong Doppler-like distortion** when computing apparent wavelengths from a cross-eddy transect. In this regime, the classical Doppler factor and gradient-based diagnostics must be interpreted with caution or corrected using time-adjusted spatial coordinates.

Despite the theoretical sensitivity of cross-stream transects to asynoptic distortion, it is important to contextualize the impact in our specific case along MT. The eddy's translational velocity was relatively small, with $c = 0.0704\,\mathrm{m \cdot s^{-1}}$, while the R/V moved at an average speed of $v_v \approx 4.66\,\mathrm{m \cdot s^{-1}}$. The full cross-eddy transect (MT) lasted approximately 7.3 hours, during which the eddy displaced by only about 1.8 km. Given the eddy's horizontal scale—exceeding 100 km in diameter—this displacement is minor and unlikely to produce a significant distortion in the inferred structure.

Moreover, while the rotational motion of fluid parcels could also contribute to internal distortion, previous results from Valencia et al. [2025] show that the azimuthal velocities in the inner core (within a radius of $\sim 25$ km) result in rotation periods of approximately 4 days. Since this rotational timescale is significantly longer than the duration of the MT transect, the relative motion of water parcels during the sampling window is minimal. Additionally, this rotational period increases with radius, further reducing the likelihood of substantial deformation at larger distances from the eddy center.

Therefore, although the theoretical framework suggests a strong potential for Doppler-like distortion in cross-stream transects, both the slow drift of the eddy and the relatively slow internal rotation imply that, in our case, the effect is likely to be small. This supports the validity of using MT to compare the radial structure of azimuthal velocities, as theoretical considerations suggest that asynoptic distortions should be limited in this case.

**4 Assessment of Apparent Structure Deformation Due to Sampling Strategy**

**OceT Transect: Along-Eddy Sampling**

Based on the Doppler factor analysis presented above, we expect that the eddy structure observed during the **eIMPACT OceT phase** was subject to spatial distortion due to the asynoptic nature of the sampling. In particular, *Bentayga* translated westward at a mean speed of $c = 0.051 \pm 0.012\,\mathrm{m \cdot s^{-1}}$, while the R/V advanced in the same direction with a mean velocity of $v_v = 0.454 \pm 0.167\,\mathrm{m \cdot s^{-1}}$. Over the course of the $\sim$9-day transect, the eddy displaced approximately 40 km, a distance comparable to its inner-core radius ($\sim 25$ km) and substantial relative to its total horizontal scale ($\sim 100$ km). This motion is expected to stretch the apparent horizontal structure of the eddy in the direction of sampling, as predicted by the Doppler framework.

Figure 5 of Valencia et al. [2025]—showing vertical sections of potential density anomaly, conservative temperature, and absolute salinity along OceT—presents signatures consistent with this elongation. In that study, however, we interpreted the structure as a physical feature of the eddy: an inner core surrounded by two distinct velocity rings, together spanning a radial width of $\sim 55$ km. Interestingly, this scale is close to the mean displacement of the eddy during the OceT sampling phase, suggesting that the observed ring-like structure may, at least in part, result from Doppler-induced deformation.

As derived earlier, the Doppler Factor $D$ should quantify the distortion in the observed spatial scale due to asynoptic sampling. To express this distortion as a percentage change in the apparent spatial scale, we define:

$$\Delta_\lambda = \left( \frac{1}{D} - 1 \right) \times 100$$

where:

- $\Delta_\lambda$ is the percentage of deformation in the apparent wavelength,

- $D = 1 - \frac{c}{v_v}$ is the Doppler Factor.

For the **eIMPACT OceT phase**, the eddy drift ($c$) and vessel speed ($v_v$) were:

$$c = 0.051 \pm 0.012 \ \mathrm{m \cdot s^{-1}}, \qquad v_v = 0.454 \pm 0.167 \ \mathrm{m \cdot s^{-1}}$$

This yields:

$$D = 1 - \frac{c}{v_v} = 1 - \frac{0.051}{0.454} \approx 0.888$$

Thus, the expected distortion is:

$$\Delta_\lambda = \left( \frac{1}{0.888} - 1 \right) \times 100 \approx 12.6\%$$

We can also estimate the uncertainty in $D$ given by:

$$\sigma_D = \left| \frac{\partial D}{\partial c} \right| \sigma_c + \left| \frac{\partial D}{\partial v_v} \right| \sigma_{v_v} = \left| -\frac{1}{v_v} \right| \sigma_c + \left| \frac{c}{v_v^2} \right| \sigma_{v_v}$$

Then, inserting the numerical values:

$$\sigma_D = \left( \frac{1}{0.454} \cdot 0.012 \right) + \left( \frac{0.051}{(0.454)^2} \cdot 0.167 \right) \approx 0.026 + 0.023 = 0.049$$

To estimate the propagated uncertainty in $\Delta_\lambda$, we use:

$$\sigma_{\Delta_\lambda} = \left| \frac{d}{dD} \left( \frac{1}{D} - 1 \right) \times 100 \right| \cdot \sigma_D = \left( \frac{100}{D^2} \right) \cdot \sigma_D$$

$$\sigma_{\Delta_\lambda} = \left( \frac{100}{0.888^2} \right) \cdot 0.049 \approx 126.8 \cdot 0.049 \approx 6.2\%$$

So, the sampling strategy during the **eIMPACT OceT phase** likely induced a spatial distortion of:

$$\Delta_\lambda = 12.6 \pm 6.2\%$$

Representing a non-negligible elongation of the apparent horizontal structure of the eddy, primarily along the sampling direction.

**SeaSoar Transects T3 and T4: Opposing Sampling Directions**

Transects T3 and T4 from the **eIMPACT SeaSoar phase** were conducted on consecutive days, representing contrasting sampling configurations: T3 was aligned with the eddy's propagation direction (downstream), while T4 was performed in the opposite direction (upstream).

**T3** was carried out on 11 November 2022, from 00:48 to 16:33 UTC, lasting approximately 15.8 hours. During this transect, the vessel advanced at a mean speed of $v_v = 4.314 \pm 0.092 \, \mathrm{m \cdot s^{-1}}$, in the same direction as *Bentayga*, which translated at $c = 0.041 \, \mathrm{m \cdot s^{-1}}$, moving approximately 2.33 km, which is minimal compared to its diameter ($\sim 100$ km). The Doppler factor was:

$$D = 1 - \frac{c}{v_v} = 1 - \frac{0.041}{4.314} \approx 0.9905$$

and the corresponding percentage of deformation:

$$\Delta_\lambda = \left( \frac{1}{D} - 1 \right) \times 100 \approx 0.96\%$$

To propagate the uncertainty:

$$\sigma_D = \left| \frac{\partial D}{\partial v_v} \right| \sigma_{v_v} = \left| \frac{c}{v_v^2} \right| \sigma_{v_v} = \left( \frac{0.041}{(4.314)^2} \right) \cdot 0.092 \approx 0.0002$$

$$\sigma_{\Delta_\lambda} = \left( \frac{100}{D^2} \right) \cdot \sigma_D = \left( \frac{100}{(0.9905)^2} \right) \cdot 0.0002 \approx 0.02\%$$

Thus:

$$\Delta_\lambda = 0.96 \pm 0.02\%$$

**T4** was conducted from 18:38 UTC on 11 November to 10:30 UTC on 12 November, with a duration of 15.9 hours. In this case, the vessel moved against the eddy's propagation direction at $v_v = 4.293 \pm$

$0.128\,\mathrm{m\cdot s^{-1}}$, while *Bentayga* translated at $c = 0.051\,\mathrm{m\cdot s^{-1}}$, displacing approximately 2.92 km during the transect. The resulting Doppler factor was:

$$D = 1 - \frac{c}{-v_v} = 1 + \frac{c}{v_v} = 1 + \frac{0.051}{4.293} \approx 1.0119$$

leading to:

$$\Delta_\lambda = \left(\frac{1}{D} - 1\right) \times 100 \approx -1.18\%$$

Uncertainty:

$$\sigma_D = \left(\frac{0.051}{(4.293)^2}\right) \cdot 0.128 \approx 0.0004$$

$$\sigma_{\Delta_\lambda} = \left(\frac{100}{D^2}\right) \cdot \sigma_D = \left(\frac{100}{(1.0119)^2}\right) \cdot 0.0004 \approx 0.039\%$$

Thus:

$$\Delta_\lambda = -1.18 \pm 0.04\%$$

Although T3 and T4 were performed with opposite orientations, both exhibit very small Doppler-induced deformation—less than 1.2%—with minor uncertainty. These results confirm that the SeaSoar transects can be considered quasi-synoptic, introducing negligible spatial distortion in the observed eddy structure.

**ZT: Counter-Propagation Sampling**

The Zonal Transect (ZT) of the **eIMPACT ortho-transects phase** was carried out on 14 November 2022, from 01:15 to 08:15 UTC, lasting approximately 7.0 hours. During this time, the vessel advanced nearly zonally in the opposite direction to the eddy's translation (i.e., upstream), at an average speed of $v_v = 4.604 \pm 0.223\,\mathrm{m\cdot s^{-1}}$, while *Bentayga* translated westward at $c = 0.070\,\mathrm{m\cdot s^{-1}}$.

Over the duration of the transect, the eddy displaced a horizontal distance of approximately 1.77 km. This displacement is relatively small compared to the transect length (116.07 km) and the horizontal scale of the eddy itself, indicating limited structural advection during the sampling.

The Doppler factor for this transect is:

$$D = 1 - \frac{c}{v_v} = 1 - \frac{0.070}{4.604} \approx 0.9848$$

The percentage of deformation in the apparent wavelength is then:

$$\Delta_\lambda = \left(\frac{1}{D} - 1\right) \times 100 \approx \left(\frac{1}{0.9848} - 1\right) \times 100 \approx 1.54\%$$

The uncertainty in $D$ is computed using first-order error propagation:

$$\sigma_D = \left|-\frac{1}{v_v}\right| \sigma_c + \left|\frac{c}{v_v^2}\right| \sigma_{v_v} = \frac{1}{4.604} \cdot 0.000 + \frac{0.070}{(4.604)^2} \cdot 0.223 \approx 0.00073$$

Then, the uncertainty in the percentage deformation becomes:

$$\sigma_{\Delta_\lambda} = \left(\frac{100}{D^2}\right) \cdot \sigma_D \approx \frac{100}{(0.9848)^2} \cdot 0.00073 \approx 0.075\%$$

Thus, the expected distortion due to asynoptic sampling during ZT is:

$$\Delta_\lambda = 1.54 \pm 0.08\%$$

This result confirms a negligible Doppler-induced deformation in the observed eddy structure. The short duration of the transect and the minimal displacement of the eddy—less than 2% of the transect length—further support the reliability of this section for analyzing the radial distribution of the eddy's azimuthal velocities.

To summarize the results presented throughout this section, Table 1 compiles the key parameters used in the Doppler distortion analysis for each sampling transect. The table includes the direction of sampling relative to the eddy's propagation, the average vessel and eddy speeds, total sampling duration, the eddy's displacement during each transect, and the corresponding Doppler factor and wavelength deformation. This synthesis allows for a direct comparison of the magnitude of Doppler-like effects across phases. Notably, the strongest deformation is associated with the eIMPACT OceT phase, where the eddy's slow but sustained displacement over nine days likely contributed to a non-negligible spatial bias. In contrast, transects T3, T4, and ZT exhibit minimal deformation, reinforcing their value as quasi-synoptic references. Although Doppler metrics were not formally computed for MT due to its orthogonal orientation, our earlier discussion supports its reliability as a reference transect given the minimal eddy movement and slow internal rotation during its sampling.

Table 1: Summary of Doppler distortion analysis for the different sampling phases. For each transect, we show the direction of sampling relative to the eddy's propagation, average vessel speed ($v_v$), mean eddy translation speed ($c$), total duration ($\Delta t$), eddy displacement during sampling ($\Delta x$), Doppler Factor ($D$), and the estimated deformation in observed wavelength ($\Delta_\lambda$).

| Transect | Direction | $v_v$ [m/s] | $c$ [m/s] | $\Delta t$ | $\Delta x$ [km] | $D$ | $\Delta_\lambda$ [%] |
|---|---|---|---|---|---|---|---|
| OceT | Downstream | 0.454 | 0.051 | 9.0 d | 40.0 | $0.888 \pm 0.049$ | $12.6 \pm 6.2$ |
| T3 | Downstream | 4.314 | 0.041 | 15.8 h | 2.33 | $0.990 \pm 0.002$ | $1.0 \pm 0.2$ |
| T4 | Upstream | 4.293 | 0.051 | 15.9 h | 2.92 | $1.012 \pm 0.003$ | $-1.2 \pm 0.3$ |
| ZT | Upstream | 4.604 | 0.070 | 7.0 h | 1.77 | $1.015 \pm 0.005$ | $-1.5 \pm 0.5$ |
| MT | Orthogonal | 4.662 | 0.070 | 7.3 h | 1.84 | — | — |

**5 Hydrographic Structure Along SeaSoar Transects T3 and T4**

The SeaSoar transects T3 and T4 were conducted during the eIMPACT2 survey between 11 and 12 November 2022. Both sections crossed the Bentayga eddy along near-parallel paths, slightly offset from its geometric center—T3 sampling the southern region and T4 the northern. The undulating SeaSoar platform profiled the upper ocean (30–320 m depth) with a vertical resolution of approximately 2 m and a horizontal resolution of 1.5 km.

Figure 4 presents vertical sections of potential density anomaly ($\sigma_\theta$), conservative temperature ($\Theta$), and absolute salinity ($S_A$) along T4 (upper panels) and T3 (lower panels). These profiles reveal the internal hydrographic structure of the eddy and provide a reference for evaluating its spatial symmetry and vertical stratification. These quasi-synoptic observations are central to the ongoing effort to assess the potential spatial distortion introduced by the non-synoptic sampling performed during the OceT phase.

In particular, they are being used to construct an idealized radial representation of the eddy, which serves as a baseline for comparison against the potentially deformed patterns observed in OceT. This reconstruction is being approached by interpolating the hydrographic properties from T3 and T4 onto a common radial coordinate system centered on the estimated eddy core. Each data point is assigned a radial distance from this center, enabling the aggregation of values into a smoothed, azimuthally symmetric structure. The resulting fields aim to represent a reference eddy minimally affected by Doppler-like distortion, against which the features observed in OceT can be contrasted. This process is intended to help disentangle genuine physical asymmetries from sampling-induced artifacts and to clarify whether features such as elongation or displaced cores are intrinsic or methodological in origin.

[Figure]

Figure 4: Vertical sections of (a,d) potential density anomaly ($\sigma_\theta$), (b,e) conservative temperature ($\Theta$), and (c,f) absolute salinity ($S_A$) for transects T4 (top row) and T3 (bottom row). The distance axis is defined along the transect track, with pressure as the vertical coordinate. Data were collected using the SeaSoar platform during the eIMPACT2 survey.

**6    Conclusions**

This study highlights the importance of sampling strategy in capturing the mesoscale structure of eddies. The differential movement of the eddy and the survey grid affects data interpretation, requiring correction factors such as the Doppler shift to properly reconstruct dynamic features.

**7    Conclusions**

This report provides a focused assessment of the spatial distortion introduced by non-synoptic sampling during the eIMPACT OceT phase, in response to concerns raised during the EGU Discussion process. Using the Doppler framework proposed by Allen et al. [2001], we quantified the expected deformation of the observed eddy structure and compared it to quasi-synoptic references obtained during the eIMPACT SeaSoar and ortho-transects phases.

Our analysis shows that the sampling configuration during OceT likely introduced a non-negligible elongation of the eddy's apparent structure, with an estimated deformation of $12.6\pm6.2\%$ in the sampling direction. In contrast, the SeaSoar transects T3 and T4 and the ortho-transect ZT present much lower distortion values (all $< 1.6\%$), supporting their use as quasi-synoptic references. The MT transect, while not formally evaluated due to its orthogonal orientation, was also deemed reliable based on the low drift and slow internal rotation of the eddy during its sampling.

To further evaluate the impact of distortion, we are constructing an idealized, azimuthally averaged structure of the eddy using interpolated data from T3 and T4. This reconstructed structure will serve as a baseline to compare with the eddy as observed during the OceT phase, helping to distinguish between genuine physical asymmetries and sampling-induced artifacts. Additionally, a comparison of azimuthal velocity distributions has been initiated using data from the ortho-transects (ZT and MT), which intersect the eddy near its geometric center. These will be contrasted with the azimuthal velocities derived from the OceT sampling to assess the spatial consistency of the rotational structure, particularly the radial extent of the solid-body core. This dynamical cross-check provides an independent validation of the eddy structure and will help determine whether the kinematic features captured in OceT are robust or artifacts of spatial distortion.

Although the analysis is ongoing, the results obtained so far suggest that the non-synoptic sampling in OceT may introduce biases in the computation of dynamical metrics such as APE, KE, and ASA. Future work will aim to quantify the magnitude of these biases and determine whether corrections or

reinterpretations are necessary for the affected estimates. Overall, this report confirms the relevance of considering synopticity when interpreting oceanographic structures and highlights the value of multiple sampling approaches to validate the robustness of derived conclusions.

**Appendix: Altimetric Eddy Metrics**

The altimetric eddy metrics used in this study—including the eddy's trajectory, center position, and the perimeter corresponding to the maximum circum-averaged geostrophic speed—were obtained from the near-real-time (NRT) product META3.2exp distributed by AVISO+ (SSALTO/DUACS) (`https://www.aviso.altimetry.fr`). Further details can be found in Valencia et al. [2025].

**References**

J.T. Allen, D.A. Smeed, A.J.G. Nurser, J.W. Zhang, and M. Rixen. Diagnosis of vertical velocities with the qg omega equation: an examination of the errors due to sampling strategy. *Deep Sea Research Part I: Oceanographic Research Papers*, 48(2):315–346, 2001. ISSN 0967-0637. doi: https://doi.org/10.1016/S0967-0637(00)00035-2. URL `https://www.sciencedirect.com/science/article/pii/S0967063700000352`.

L. P. Valencia, Á. Rodríguez-Santana, B. Aguiar-Gonzaléz, J. Arístegui, X. A. Álvarez-Salgado, J. Coca, and A. Martínez-Marrero. Mesoscale dynamics of an intrathermocline eddy in the canary eddy corridor. *EGUsphere*, 2025:1–46, 2025. doi: 10.5194/egusphere-2025-99. URL `https://egusphere.copernicus.org/preprints/2025/egusphere-2025-99/`.

---

## Author Response (AR1)

**Response to Referee Comments**
**on the manuscript entitled:**
**"Mesoscale dynamics of an intrathermocline eddy in the Canary Eddy Corridor"**
**(doi: 10.5194/egusphere-2025-99)**

We thank both referees for their thoughtful reviews and constructive comments. We have carefully revised the manuscript accordingly and provide below a detailed, point-by-point response. Each comment is introduced with the line or figure it refers to, followed by our response in bold.

**Response to Referee #1**
**(doi: 10.5194/egusphere-2025-99-RC1/RC2)**

**l60:** Bosse et al. (Scientific Reports, 2019) could be considered to describe the role of PV as a barrier for horizontal exchanges.

**Response:** We have now incorporated a direct reference to Bosse et al. [2019] to strengthen the discussion on the role of PV anomalies in trapping water masses. Specifically, we note that the low-PV core of ITEs enhances their capacity to isolate and transport water, and that the associated PV gradient acts as a dynamic barrier to lateral exchanges, particularly below the mixed layer. We also highlight the seasonal modulation of this barrier, which enriches our discussion of seasonal processes influencing the evolution of the Bentayga eddy. The revised paragraph is located in the Introduction (lines 62–66 of the revised manuscript).

**l68-74:** More broadly speaking subthermocline or intrathermocline eddies have been long known to form via a mechanism of diapycnal mixing (causing the low stratification) identified in two main scenarios: diapycnal mixing followed by restratification and geostrophic adjustment (McWilliams, 1988) and friction in the bottom boundary layer along continental slope (D'Asaro, 1988). When a prexisting deep eddy merges with a surface mesoscale eddy, this process can lead to the stacking of cores and formation of an ITE (eg, Belkin et al 2020; or Garreau et al 2018). Similarly, the history of winter mixing can help understand the formation of isolated deep core in anticyclonic mesoscale eddies (eg, Barboni et al 2023; or Yu et al. 2017). Since the described eddy is living for one year, it could experience winter mixing and restratification. Are they any Argo float sampling in the eddy that could be used to describe the vertical structure of the eddy?

**Response:** We thank the referee for this detailed and insightful comment. In the revised manuscript, we have expanded the introduction to incorporate the theoretical mechanisms proposed by McWilliams [1988] and D'Asaro [1988], which are now discussed in lines 71–75 and 75–79, respectively. We also integrated recent observational studies that support more complex vertical eddy structures. In particular, the formation of double-core and vertically layered ITEs via winter convection and vertical alignment processes is now discussed in lines 89–95 [Yu et al., 2017; Barboni et al., 2023]. Additional mechanisms involving the stacking of distinct water masses, such as those described in Garreau et al. [2018] and Belkin et al. [2020], are included in lines 105 and 107-108. Collectively, these additions reflect the growing recognition that multiple processes may contribute to ITE formation and evolution.

Regarding the availability of Argo float data within the eddy, we performed a targeted search and identified two profiles that intersected the outer boundary of Bentayga during its life cycle. These profiles, recorded on 25 February and 28 May 2023, were located approximately 56 km and 43 km from the eddy center, respectively—well outside the core. Given their peripheral position within the eddy's surrounding ring, a region typically marked by high variability and reduced coherence, the hydrographic properties observed do not reliably reflect the eddy's internal structure. Consequently, these profiles were not included in the present analysis aimed at characterizing the eddy's wintertime interior.

**l138:** Please justify the scales chosen for filtering (12km seems to be the spacing between stations? What does 8m correspond to?).

**Response:** We appreciate the referee's attention to this methodological detail. The filtering scales applied were selected based on an analysis of the spatial autocorrelation structure of the noise field, as reconstructed from EOF modes associated with high-wavenumber variability (modes 4 to N) (lines 181-184 of the revised manuscript). The horizontal decorrelation length scale was estimated from the EOF component representing horizontal structure and yielded an integral scale of approximately 12 km, consistent with the average spacing of the OceT stations. The vertical scale was derived from the component capturing vertical variability, resulting in an integral scale near 8 m. These values were therefore used to guide the Gaussian filter parameters applied to the hydrographic fields.

**l145:** Could you please justify the choice of NRT product versus DT? (it seems not to be available for the period, but it could be important to mention).

**Response:** We have now clarified in the manuscript that the use of the Near Real Time (NRT) product from the META3.2exp atlas was necessary, as the Delayed Time (DT) version does not fully cover the period associated with the evolution of the Bentayga eddy. To complement this and place the eddy's trajectory in a broader climatological context, we additionally used the META3.2 DT *allsat* product, particularly to explore its interaction with filaments from the northwestern African upwelling system. These clarifications have been incorporated in the revised manuscript (lines 192-196).

**l204–207:** What does "r" represent in the equation here? and if it is the radial extension in cylindrical coordinate system, how can it provide values of fluxes?

**Response:** The term $r$ represents the radial distance from the eddy center to the outer boundary of the inner core, and $2r$ is used to approximate the width of this vertical face. The fluxes of volume, heat, and salt are thus computed as the product of the eddy's translational speed ($c$) and the vertical integral of the corresponding anomaly (e.g., salinity, temperature) across this frontal section. This approach assumes that the core region dominates the net transport and simplifies the cross-section as approximately rectangular. These clarifications have been added to the beginning of the "Eddy-driven fluxes" subsection (lines 252–255 of the revised manuscript).

**l215:** *Bentayga* could be written in italic throughout the manuscript.

**Response:** We thank the referee for this suggestion. However, after reviewing the house standards of *Ocean Science*, we found no requirement to italicize the names of oceanographic features. Therefore, we have opted to retain the standard (roman) font for "Bentayga" throughout the manuscript, in line with the journal's editorial conventions.

**Figure 2:** This figure is very heavy, please consider using transparency and better choice of marker in order to improve its readiness.

**Response:** We thank the referee for this helpful comment. We opted to retain a single multi-panel figure (Fig. 2), while applying visual adjustments to reduce visual load. These include increased background transparency, simplified contours, and improved marker contrast to better distinguish key features such as eddy trajectories and sampling stations.

**l229, 244 and Fig 3:** It might be more instructive to describe omega in terms of Rossby number (ie normalized by f).

**Response:** In the revised manuscript, we have incorporated the Rossby number ($Ro = \zeta/f$) as an additional metric of the eddy's nonlinearity. Its temporal evolution is now shown in Figure 3d and discussed in the accompanying text (lines 280, 285-287, and 294 of the revised manuscript).

**l287:** This could be the sign of increasing relative importance of high frequency wave compared to mesoscale signal. Do you have means to check this?

**Response:** We appreciate the referee's suggestion. While our primary analysis focused on the mesoscale evolution, we conducted a dedicated time series experiment during the eIMPACT2 survey, involving over 24 hours of VMADCP sampling at the eddy center. Preliminary results revealed a clear presence of high-frequency motions, including near-inertial wave trapping between 300–500 m and internal waves with 6–8 h periods centered near 200 m depth. These findings support the idea that high-frequency variability became more prominent at the base of this mature eddy. However, we note that the interpretation of

time-evolving properties from OceT data may also be affected by distortions due to the lack of full synopticity, as discussed in our response to Referee #2. A more comprehensive wave–eddy interaction analysis will be presented in a follow-up study incorporating the mentioned datasets.

**l307:** lens?-like
**Response:** We followed this helpful suggestion. The original expression "bag-like" has been replaced with "lens-shaped," which is more appropriate and commonly used in the oceanographic literature to describe subsurface anticyclonic cores such as ITEs (line 361 of the revised manuscript).

**l349:** was likely not well resolved.
**Response:** We agree with the referee that "not well resolved" more accurately conveys the limitation imposed by the sampling coverage. The original sentence has been revised accordingly to reflect this more objective and technically precise phrasing (line 405 of the revised manuscript).

**Figure 8:** Please find a way to better highlight the section.
**Response:** We revised Figure 8 to enhance the visibility of the transect and to ensure that the ship's path is more clearly represented.

**l361:** It looks to me that the radius is actually increasing with depth... Could be maybe hightlight the position of velocity maximum?
**Response:** While an apparent outward expansion of the radius of maximum velocity is visible with depth, as shown in the horizontal sections of current vectors and velocity magnitude (Figs. B1 and B2), this pattern is also accompanied by a deformation of the core and a lateral displacement toward regions influenced by adjacent cyclonic structures. For this reason, and to ensure internal consistency with our hydrographic analysis, our description of the eddy's geometry is based primarily on the density field, which provides a more coherent framework for characterizing its stratified core and boundaries.

**Figure 9:** Please add labels for SA. The thicker contours are not visible enough... one label per sigma value is enough.
**Response:** In the revised version of Figure 9, we improved the visibility of the isopycnal contours by increasing their thickness and contrast, and we added one label per $\sigma_\theta$ value to facilitate interpretation of the density structure within the SA panels. These adjustments enhance the readability of the eddy's internal hydrographic features.

**l413:** The location of sign change in vorticity does not usually match with the position of maximum velocity. For instance, a Gaussian eddy with a velocity maximum at R would have a change in sign of vorticity at sqrt(2)R.
**Response:** We thank the referee for this pertinent observation. Indeed, in anticyclonic vortices with Gaussian-like structures, the radius of maximum azimuthal velocity ($R_{max}$) does not coincide with the radius at which relative vorticity—and thus the Rossby number (Ro)—changes sign. In our analysis, we found $R_{max} \approx 41$ km, while the Ro sign change occurs at approximately 62 km, closely matching $\sqrt{2}R_{max}$, as expected theoretically. This distinction has now been explicitly clarified in the revised manuscript (lines 524-531), and we have corrected the original phrasing to avoid conceptual ambiguity. Furthermore, we note that $R_{max}$ agrees well with the climatological first baroclinic Rossby deformation radius estimated from the polynomial fit of Chelton et al. [1998], further supporting the consistency of our dynamical characterization.

**l415:** I understand that the "shell" is defined by the zero vorticity contour, am I right?
**Response:** Yes, the referee is correct. In our revised manuscript, we explicitly define the "shell" as the region bounded by the radius at which the Rossby number (Ro) changes sign—i.e., where relative vorticity becomes zero. This radius ($\sim 62$ km) corresponds to the dynamically defined outer edge of the inner coherent ring, and also aligns with the outer limit of the strain maximum sector. This region serves as a physically meaningful boundary enclosing the rotationally coherent part of the eddy. The manuscript has been updated accordingly to clarify this definition and avoid ambiguity (lines 526–533).

**l417:** An Eulerian way to look at the ability for the eddy to trap water would be to look at the strain rate (which should be near zero in the eddy center and increase ouside the solid-body core). Did you

have a look at this diagnostics?

**Response:** We appreciate the referee's insightful suggestion. While the strain rate diagnostic was not included in the original manuscript, this comment—along with the referenced literature—prompted us to incorporate it into the revised analysis. We now explicitly compute the strain rate field from the horizontal and azimuthal velocity gradients of the VMADCP data. As expected for coherent mesoscale vortices, the resulting strain field displays a pronounced minimum at the eddy center and a clear increase beyond the solid-body core, reaching its maximum in the sector of the surrounding ring. This behavior supports the use of this diagnostic to characterize the eddy's water-trapping capacity from an Eulerian perspective. The corresponding description has been added to Section 3.5 of the revised manuscript (lines 505-516).

**General:** The definition of radius should be clarified. I see at least three definition : location of solid body rotation core, of velocity maximum, and of zero vorticity. It might be usefull to introduce specific notation of each of these for sake of clarity.

**Response:** We appreciate the referee's suggestion and have clarified this point in the revised manuscript. Three distinct but complementary radii are now explicitly defined to characterize the eddy's radial structure: (1) the core radius ($R_c$), corresponding to the region exhibiting solid-body rotation; (2) the radius of maximum azimuthal velocity ($R_{max}$), where the peak swirl speed is attained; and (3) the outer radius ($R_e$), delineating the external edge of the eddy, beyond which velocity and hydrographic gradients cease to reflect the eddy's coherent dynamics. Although these definitions arise from different diagnostics—rotational fits, velocity profiles, and vorticity or strain-based metrics—they tend to converge in magnitude. In our study, initial estimates from visual inspection were refined using quantitative methods (as detailed in Supplementary Note 3), and the resulting values are applied consistently throughout the manuscript.

**l430:** The inherent property of PV to be conserved in absence of forcing and dissipation makes it also an important parameter to caracterise the trapping of water.

**Response:** We agree with the referee's observation and have revised the text accordingly to better emphasize the conservative nature of potential vorticity in the absence of forcing and dissipation. This revision highlights why PV anomalies are particularly informative for characterizing the eddy's water-trapping capacity (line 546-549 of the revised manuscript).

**Figure 13:** A log scale for PV might show more clearly the two orders of magnitude described in the text.

**Response:** We thank the referee for this helpful suggestion. In the revised version of Figure 13 (Figure 15 in the revised manuscript), the color scale for Ertel's potential vorticity has been converted to logarithmic units to better highlight the wide dynamic range of PV values, particularly the low-PV core of the eddy.

**Figure 15:** Same remarks as Figure 2.

**Response:** Following the same approach applied to Figure 2, we revised Figure 15 (Figure 17 in the revised manuscript) to improve clarity and visual balance. Specifically, we adjusted the background colormaps, refined the streamline overlays, and simplified graphic elements to ensure that the key features—namely, the interaction between upwelling filaments and the eddy—are clearly discernible. The caption was also edited for precision and consistency. These changes improve the readability of the figure while preserving its interpretative value.

**l445:** Are they any location of negative PV near the surface? This could be linked to symmetric instability described by Brannigan et al 2017 (see also Thomas et al., 2013 : https://doi.org/10.1016/j.dsr2.2013.02.025).

**Response:** We thank the referee for this insightful question. Using the original (non-reconstructed) fields, we identified localized regions of negative PV near the surface, particularly in areas where isopycnals exhibit pronounced tilting. While these features are consistent with the type of PV structure that may allow for symmetric instability, a comprehensive analysis of this possibility is still underway. We plan to address this topic in a follow-up study that will incorporate additional datasets collected during the same campaign, including high-resolution microstructure profiles (e.g., vertical shear).

**l576:** About Biogeochemical impacts : since the Seasoar was carrying an oxygen sensor, how does its

distribution look like across the eddy? It could be informative to discuss the possible biogeochemical impact in the light of these measurements.

**Response:** We thank the referee for this pertinent observation and for encouraging us to explore and incorporate this dimension of the dataset. In response, we have added a new figure (Fig.10 in the revised manuscript) and an accompanying subsection in the Results (Section 3.4) that specifically addresses the distribution of dissolved oxygen and Apparent Oxygen Utilization (AOU) across the Bentayga eddy. This analysis reinforces the interpretation of the eddy core as an isolated reservoir of relatively unprocessed, oxygen-deficient waters that have been transported offshore into the open ocean interior. These additions strengthen the manuscript by directly linking the eddy's physical coherence to its potential biogeochemical impact, particularly with respect to oxygen inventory redistribution and ventilation anomalies at thermocline depths.

**l603:** quantities of water.

**Response:** We appreciate the referee's suggestion. The original phrase "quantities of volume" has been revised to "volumes of water," which more accurately and clearly reflects the intended meaning in this context (line 786 of the revised manuscript).

**Response to Referee #2**
**(doi: 10.5194/egusphere-2025-99-RC3)**

**l36, l191, l436, l491:** Check the location of these references: Barceló-Llull et al., 2017a (Vertical velocity) and Barceló-Llull et al., 2017b (Anatomy...). For instance, in L. 36, 191, 436, 491 the reference should be 2017b (Anatomy...) instead of 2017a.

**Response:** We thank the referee for pointing out the incorrect attribution. This issue has been fully resolved following the comprehensive revision of the manuscript in response to the comments and suggestions from both referees. Specifically, the reference to Barceló-Llull et al. [2017a, JPO] is no longer cited in the revised version, as the content it referred to is no longer part of the study. As a result, the confusion regarding its attribution has been eliminated.

**l122:** Explain what "Continuous yo-yo tows" is.

**Response:** We thank the referee for pointing out the ambiguity. The original expression has been revised to more accurately describe the observational strategy used during the eIMPACT2 SeaSoar phase. Specifically, we now refer to a towed undulating vehicle (SeaSoar Mk II) equipped with a CTD-O system that continuously profiled the upper water column along the ship track (lines 153-154 of the revised manuscript).

**l128:** give depth of VMADCP sampling.

**Response:** To address this comment and improve clarity, we have revised the manuscript to explicitly state the vertical coverage of the VMADCP measurements. Horizontal velocity profiles were obtained from approximately 30 to 800 m depth, with a vertical resolution of 16 m (lines 159-160 of the revised manuscript). This update ensures consistency with the description of the sampling grid and better informs the reader about the data coverage.

**l129:** "Another phase used in this study, the eIMPACT2 OceT phase (19–28 November 2022)," – rephrase.

**Response:** In the revised manuscript, the beginning of this paragraph has been rephrased for clarity and precision (line 161).

**l129:** "Involved vertical CTD-O profiles" – with the SeaSoar or rosette?

**Response:** We have clarified that a rosette system was used for the collection of vertical CTD-O profiles during the eIMPACT OceT phase (line 162 of the revised manuscript).

**l129:** How much time took to complete the 26 stations of the OceT transect? If 9 days, how can you

use these data to calculate APE, KE, ASA and AHA? Are you considering these observations synoptic? Explain and justify.

**Response:** We thank the referee for raising this critical point regarding the synopticity of the OceT transect, which indeed spanned nine days (19–28 November 2022). This limitation is central to interpreting the eddy's energetics and property transports, and we have addressed it in detail across Supplementary Notes 1–5. In Supplementary Note 1, we quantify the spatial distortion introduced by the lack of synopticity using the framework of Allen et al. [2001], showing that the vessel's downstream motion relative to the eddy induced a Doppler-like elongation of approximately $12.6 \pm 6.2\%$. To assess the implications of this distortion, we developed an idealized eddy reconstruction based on quasi-synoptic orthogonal transects (Supplementary Note 2) and redefined the radial eddy structure using hydrographic and velocity criteria (Supplementary Note 3). Supplementary Note 4 presents sensitivity tests indicating that the energetic and anomaly budgets remain within acceptable uncertainty bounds when varying the integration domain. Ultimately, as discussed in Supplementary Note 5, we justify the use of OceT profiles for vertical integration on the basis that this dataset provides the most complete vertical coverage of the eddy. While we acknowledge the temporal limitations, the combined methodological framework enables us to quantify, correct for, and contextualize the associated distortions. Integrations were constrained to the eddy's core, which exhibited high temporal and spatial coherence throughout the survey, and the resulting estimates are thus interpreted as conservative lower bounds. This methodological clarification has also been incorporated in the revised manuscript (lines 316–320), as an introductory paragraph to Section 3.2.

**l152:** "(relative to the average field for the period 1993–2012)" – maybe clearer to provide the version of the altimetric product (e.g., vDT2021?).

**Response:** We thank the referee for this helpful observation. We now specify that the sea level anomaly and geostrophic current fields were obtained from the CMEMS near-real-time altimetry product `SEALEVEL_EUR_PHY_L4_NRT_008_060` (see lines 198-201 of the revised manuscript). Since this is a Near Real Time (NRT) product, no specific version such as vDT2021 applies.

**l167:** "Given the geometry of the Bentayga eddy", explain briefly the geometry you are referring to.

**Response:** We appreciate the referee's observation. In the revised manuscript, we now clarify that the use of this form of Ertel's potential vorticity is motivated by the eddy's small aspect ratio. This clarification has been added in line 213.

**Figure 2 and Figure 15:** Both figures could be cleaner: remove or lighten grey contours and arrows; adjust black contours. In Fig. 15, improve visibility of filaments in SST/chl-a maps.

**Response:** We followed the referee's suggestions and revised both figures accordingly. Please note that the figure originally referred to as Figure 15 is now Figure 17 in the revised manuscript. We hope these changes have improved their overall readability and clarity.

**Figure 2:** Add panel labels (a, b, c, ...).

**Response:** We have added panel labels (a, b, c, ...) to Figure 2 as suggested.

**l247–248:** "It appears that the onset of these shorter-scale fluctuations coincided with a slight northwestward deflection observed in its trajectory (Fig. 1)." where can we see this deflection? Maybe Fig. 2 1-May?

**Response:** Following a detailed review, we confirm that the onset of the shorter-scale fluctuations coincided with a slight northwestward deflection of Bentayga's trajectory, starting around 6 December 2022, shortly after the end of the eIMPACT2 survey (4 December 2022). This deflection is observable in Figure 1 of the manuscript, particularly along the trajectory between the December and January markers. We have clarified this point in the revised text to better guide the reader (lines 297–298).

**Figure 1:** For clarity and consistency, add year to date labels (dd/mm/year).

**Response:** We thank the referee for the suggestion. The date labels in Figure 1 have been updated to include the full year, following the dd/mm/yy format, to improve clarity and consistency.

**l256–259:** Expand these concepts: "reflecting the dominance of geostrophic characteristics even at these scales." "However, the relative energy contributions of these fluctuations did not always align, with

greater intensity observed in the variability of the SLA amplitude and radius."

**Response:** In the revised manuscript, we have expanded this section to clarify that the dominance of geostrophic dynamics at the timescales considered (weeks to a few months) is supported by the observed relationship between SLA amplitude and swirl speed: increases in SLA were systematically accompanied by increases in swirl speed, consistent with a geostrophically balanced eddy. We also clarified that, according to the wavelet analysis, the energy associated with fluctuations was greater in SLA amplitude and eddy radius than in swirl speed, suggesting that shorter-scale variability more strongly modulated the eddy's geometry and surface expression than its velocity field (lines 306–311).

**l639:** Lagrangian – capitalize.
**Response:** We have corrected the capitalization of "lagrangian" to "Lagrangian" as suggested (line 825 of the revised manuscript).

**Figure 4 caption:** "Vertical dotted lines indicate the positions of oceanographic stations,", dashed?
**Response:** We thank the referee for this observation. We have corrected the description in the captions of Figures 4, 5, and 7 by replacing "dotted" with "dashed" to accurately describe the vertical lines indicating station positions.

**l278–283:** Reference a figure.
**Response:** We acknowledge the referee's suggestion to reference a figure here. However, we have revised the analysis of this structure due to concerns regarding the non-synoptic nature of the OceT transect. In the revised manuscript, we clarify that the structure referred to in the original version of this study (the *external ring*) was in fact an interpretation influenced by a distorted view of the eddy. This structure no longer appears in the revised version, where the radial zonation of the eddy now includes only a core and a surrounding ring sector (see Supplementary Note 3). A note of caution regarding this reinterpretation is also included in the revised manuscript (lines 336–338).

**Figure 4:** The conclusions extracted from this figure should consider also the lack of synopticity of the observations.
**Response:** We thank the referee for this important observation. The interpretation of Figure 4 has been revised to explicitly consider the non-synoptic nature of the OceT sampling strategy. In the updated manuscript, we caution that the observed heterogeneity and structural deformation in the surrounding ring sector may result from sampling distortions. This limitation is now acknowledged directly in the text (lines 336–338), and no further interpretation of that sector is pursued. These clarifications are consistent with the revised radial zonation discussed in Supplementary Note 3 and ensure that the conclusions drawn from this figure remain robust despite the temporal distortion.

**l333:** "Associated with it" → "associated with the eddy". Also clarify "lower half of the core" with specific depth.
**Response:** Following the referee's suggestion, we revised the entire paragraph (now lines 388–397 in the revised manuscript). The expression "associated with it" has been removed, and references are now made explicitly to "the Bentayga eddy." Additionally, the phrase "lower half of the eddy core" was eliminated. Instead, specific isopycnal surfaces are now referenced along with their corresponding depths, ensuring greater clarity and precision in the description.

**l334–335:** Clarify "its base"; mention the two cores before describing the anomalies.
**Response:** Following the referee's suggestion, we revised the description to clearly distinguish between the patterns observed in different variables. Specifically, we now state that the potential density anomaly and conservative temperature fields exhibit a single-core structure (lines 388-389), while the absolute salinity anomaly field displays a double-core structure (line 392). This modification improves the clarity and logical flow of the hydrographic description of the eddy-induced anomalies.

**l339–340:** Clarify what is meant by "higher presence"; rephrase awkward temperature/salinity threshold phrasing.
**Response:** Following the referee's suggestion, we restructured the sentence to first describe the location of the colder and fresher anomalies, and then indicate their association with the SPMW and AA water masses. Additionally, we clarified the figure references, specifying that the thermohaline anomalies are

shown in Fig.7b, c and the water mass classification in Fig.6. These improvements enhance the clarity and precision of the description (lines 394–397 of the revised manuscript).

**Figure 8 caption:** please rephrase "Each panel displays a unique color scale and density range with 25 contour levels to highlight the isopycnal structure."
**Response:** Following the referee's suggestion, the caption of Figure 8 has been revised to improve clarity. The description now explicitly states that each panel uses a distinct color scale representing potential density anomaly ($\sigma_\theta$) values, with 25 uniformly spaced contour levels across the respective density range. Additional clarifications were added regarding the scaling of velocity vectors and the meaning of the dashed magenta and solid black lines (revised Figure 8 caption).

**l345:** "even at the shallowest recorded depths (Fig. 8a).", but the shallowest recorded depth is 30 m (L. 127) and Fig. 8a is at 56 m, why?
**Response:** We thank the referee for pointing out this discrepancy. We have revised the sentence accordingly, replacing "shallowest" with "shallower" to indicate that the reference is to depths shallower than 60 m.

**l350:** Do you refer to Fig. 9? Paragraph is confusing.
**Response:** We appreciate the referee's concern. The sentence has been rephrased to improve the clarity and coherence of the paragraph.

**l355:** at which depths are you referring to?
**Response:** Following the referee's suggestion, we clarified that the description refers to the upper layers, specifying "<100 m depth" to indicate the depth range at which the eddy exhibited a generally circular and slightly irregular shape. This clarification has been incorporated into the revised manuscript (line 413 of the revised manuscript).

**l357:** "The horizontal section at 120 m depth indicates that lighter water circulating around the inner core extended even at those depths (Fig. 8c)." I find difficult to see this in these horizontal sections.
**Response:** We thank the referee for their observation. In Fig. 8c, we refer to the presence of lighter waters ($\sigma_\theta < 25.3$ kg m$^{-3}$) that appear in a spatially localized patch on the eastern side of the eddy core. Although subtle, this feature coincides with the anticyclonic circulation inferred from the velocity vectors, which seem to organize the surrounding 25.3–25.4 kg m$^{-3}$ isopycnal contours into an elongated structure aligned with the flow. While we agree that this signal may not be immediately evident, we interpret it as a coherent, circulation-conditioned anomaly at this depth, consistent with the suggested advection and partial entrainment of lighter waters.

**l359–360:** Which patterns? I don't see here the reason to mention the vertical velocities associated with ITEs. I suggest removing this sentence.
**Response:** As suggested by the referee, we have removed the sentence from the manuscript.

**General:** authors write many short paragraphs in the manuscript, I recommend that they revise them throughout the manuscript, because sometimes these new paragraphs seem a continuation of what was said before and it is not justified to use a new paragraph.
**Response:** We thank the referee for this observation. We have revised the paragraph structure throughout the manuscript to ensure consistency and improve the overall flow. We hope that the revised version enhances the clarity and readability of the text.

**l360–361:** It would be easier to see the isopycnal deepening in vertical sections. Also, the biconvex shape of the eddy was already described in detail in the previous section 3.2.
**Response:** We thank the referee for this comment. While it is true that the biconvex shape of the eddy was previously described in Section 3.2 based on the OceT transect, the aim of this section is to assess the spatial consistency of those features using the quasi-synoptic SeaSoar dataset. The advantage of this dataset lies in its improved temporal coherence and higher horizontal resolution, which allow us to identify subtle horizontal patterns—such as localized deepening and shoaling of isopycnals—and assess their spatial alignment with the eddy circulation. This cross-validation across different observational phases strengthens the robustness of our interpretation and reduces the uncertainty associated with

non-synoptic sampling.

**l362:**  why salinity, and why not also temperature?
**Response:**  We thank the referee for this question. In the depth range discussed, the spatial distribution of conservative temperature closely mirrors that of potential density anomaly, offering limited additional insight. In contrast, the salinity field reveals localized structures and gradients that are not as evident in the temperature or density fields. These features are important for interpreting the water mass variability within the eddy. This distinction is illustrated in Figure 5 of the revised manuscript.

**l365:**  " Within this vertical segment, just below the mixing layer, ", give exact depth layer limits.
**Response:**  In the revised manuscript, we have explicitly indicated the mixed layer depth at the mentioned location, thereby clarifying the vertical segment referred to in the sentence (line 422).

**l367:**  The highest velocities ($>50$ cm s$^{-1}$) were located at the eddy's edges, progressively converging towards the boundaries of the inner core with increasing depth. ..." this is confusing, what do you mean by eddy's edges? avoid introducing new terms. Indeed, your definitions of inner core, inner ring, etc. in Fig. 4b are based on observations that cannot be considered synoptic. I see that the complete Seasoar grid was sampled in 5 days, much less time that the OceT transect, and hence the Seasoar sampling could be considered quasi-synoptic*. From Fig. 9a, the velocity associated with the eddy show a similar pattern than in previous ITE studies, and the speed-based radius is coherent in depth. On the other hand, Fig. 4a cannot be considered a snapshot of the velocity of the eddy due to the lack of synopticity of the observations, and the high horizontal resolution can be contaminated by the low temporal resolution.
**Response:**  We agree that the geometric definitions used to characterize the eddy's structure must be supported by robust and coherent diagnostics, especially given the limitations of non-synoptic sampling. This concern is directly addressed in Supplementary Note 3, where we refined the radial structure of the eddy by integrating information from both non-synoptic (OceT) and quasi-synoptic (SeaSoar and orthogonal transects) datasets. In the revised analysis, the boundaries of the (well-mixed) core and the surrounding ring are no longer based solely on idealized azimuthal velocity profiles. Instead, they are defined using spatially coherent features in the fields of relative vorticity, strain ratio, and radial hydrographic gradients, which together delineate dynamically consistent regions. These improved criteria mitigate the potential bias introduced by non-synoptic sampling and provide a more physically grounded framework for interpreting the eddy's geometry.

**l370:**  "Furthermore, within the inner core, the isopycnals displayed small-scale irregularities, ", where do you see this? what do you mean by irregularities? And how can you see small-scale irregularities after applying optimal interpolation with L =44 km?
**Response:**  We thank the referee for this important clarification request. As noted in the manuscript, the observed small-scale perturbations correspond to localized features in the objectively interpolated fields of density and salinity. These features are not artifacts introduced by the objective mapping procedure. Indeed, the interpolation was performed *horizontally* within each depth layer, following the original structure of each isobaric surface. This approach preserves vertical heterogeneities and does not impose any vertical smoothing across adjacent layers. Therefore, any vertical discrepancies or asymmetries present in the original profiles—such as localized doming or deepening—remain intact after interpolation. These features were verified against the original profile data and are consistent with the complex internal structure of the eddy, particularly in the core and surrounding region.

**l370:**  "characterized by doming on one side and deepening on the other side", are you referring to the typical vertical structure of ITEs: doming of the upper isopycnals and deepening of the deeper isopycnals?
**Response:**  We appreciate the referee's question. In this case, we are *not* referring to the canonical vertical biconvex structure of ITEs—i.e., doming of upper isopycnals and deepening of lower ones. Instead, we refer to a lateral asymmetry observed in the $\sim 25.3$ kg m$^{-3}$ isopycnal located within the eddy's well-mixed core. As shown in Figure 9, this isopycnal appears elevated in the western (left) portion of the core and deepened in the eastern (right) side, forming an acute internal zonal tilt. Importantly, this pattern coincides with the inclination of the 36.8 g kg$^{-1}$ salinity contour that defines the innermost homogeneous saline nucleus. The alignment between these two tracers supports the interpretation of an internally sloping structure, rather than a vertically symmetric one. This localized asymmetry likely

reflects internal deformation within the eddy core and further reinforces the spatial coherence of our observations.

**l371:** why do you introduce now salinity if you already have density that has the expected vertical structure of ITEs?

**Response:** We thank the referee for this question. In this particular case, we are *not* observing the canonical vertical structure of an ITE (i.e., a symmetric doming and deepening of isopycnals enclosing a convex core). Therefore, complementing the density field with another conservative tracer—such as salinity—provides valuable information. The isohalines help reveal subtle internal gradients and contribute to a more complete description of the eddy's core structure, especially under non-ideal or asymmetric conditions.

**l372-373:** remove or justify. What you see is the typical vertical structure of density in ITEs. See e.g. Barceló-Llull et al. 2017, DSR. Indeed, there you can find an explanation of the relation between the vertical structure in density and velocity: " This subsurface intensified anticyclonic flow implies a vertical shear that is consistent with the biconvex isopycnal shape through thermal wind balance. Above (below) this subsurface speed maximum a negative (positive) vertical shear of the horizontal velocity will adjust with a negative (positive) radial gradient of density leading to shoaling (deepening) of the isopycnals."

**Response:** We respectfully disagree with the referee's interpretation. While the vertical structure of the eddy does exhibit subsurface intensification, our observations do not reflect the canonical biconvex pattern typically associated with ITEs—i.e., symmetric shoaling and deepening of isopycnals around a central velocity maximum, as described in Barceló-Llull et al. [2017]. Instead, we identify a laterally asymmetric feature within the well-mixed core, most clearly captured by the ~25.3 kg m$^{-3}$ isopycnal and the 36.8 g kg$^{-1}$ isohaline. As previosly mentioned, these tracers exhibit a coordinated slope across the zonal dimension of the eddy core, forming an internally tilted structure rather than a symmetric dome-and-bowl geometry. This distinction, which is supported by our quasi-synoptic dataset, justifies the inclusion and interpretation of these features in the revised manuscript.

**l377:** "with a gradual inward movement towards the inner core at greater depths." remove.

**Response:** As suggested by the referee, this sentence has been removed from the manuscript.

**l379:** give values of maximum velocities.

**Response:** As suggested by the referee, we have now included the maximum geostrophic velocity values (50–55 cm s$^{-1}$) observed above 100 m depth in the revised sentence.

**l380:** "ageostrophic secondary circulation", ageostrophic horizontal velocity.

**Response:** We have modified the expression "ageostrophic secondary circulation" to "ageostrophic horizontal velocity" as suggested by the referee.

**l382:** "near the edges of the inner core," remove

**Response:** As suggested by the referee, the phrase "near the edges of the inner core" has been removed from the manuscript.

**Section 3.4:** OceT transect has non-synoptic observations. Justify the use of these observations instead of the quasi-synoptic Seasoar data for Fig. 10 and the corresponding analysis. I suggest to use the Seasoar observations for this analysis instead of the OceT transect, the radial section of the azimuthal velocity will be easier to interpret, without contamination due to the lack of synopticity, and probably only showing an inner core in solid-body rotation, without the inner/outer rings suggested by authors that may appear due to the lack of synopticity (that by the way, they should explain how they detect them in Fig. 10). Fig. 12 is from Seasoar or OceT data? Add this information in caption, and check that the other captions specify which data is shown in the corresponding figures.

**Response:** We acknowledge the referee's concern and thank them for the constructive suggestion regarding the potential use of quasi-synoptic SeaSoar data for the eddy's structural characterization. As rightly pointed out, the original identification of concentric inner and outer rings around the eddy core was influenced by a distorted view of the structure resulting from the non-synoptic nature of the OceT transect. This issue has been addressed in the revised manuscript. Our updated analysis incorporates

both the quasi-synoptic SeaSoar transects and the orthogonal sections. These reveal that, surrounding the nearly stationary and vertically coherent eddy core, there exists a dynamically active region characterized by significant spatial and temporal variability. Based on this improved understanding, we now refer to this area simply as the surrounding ring, without subdividing it into inner and outer sub-rings.

In addition, we have revised the caption of Figure 12 (now Figure 14 in the updated manuscript), as well as those of other relevant figures, to explicitly specify the dataset or sampling phase (e.g., SeaSoar or OceT) used in each case. This improves clarity and ensures full traceability for the reader.

The corresponding analysis is now presented in Section 3.5, which now also includes: (1) a revised radial zonation based on the azimuthal velocity distribution, (2) a comparison with an idealized theoretical model, and (3) the inclusion of strain rate estimates. We also refer the reader to Supplementary Note 3, which provides a detailed explanation of the revised radial structure. While we fully recognize the advantages of quasi-synoptic sampling, the integrated use of both datasets—each serving clearly defined and complementary purposes—offers a more robust and complete characterization of the eddy's radial dynamics.

**l432:** "extremely low-PV values ", extremely: use a more appropriate word.
**Response:** Following the referee's suggestion, we rephrased the description by replacing "extremely low-PV values" with "a large reduction in potential vorticity" and explicitly mentioned that this reduction results from the reduced stratification and homogeneous properties of the ITE core (line 550 of the revised manuscript).

**l436:** "general expectation for ITEs ", expectation: use a more appropriate word
**Response:** Following the referee's suggestion, we replaced "reflects the general expectation for ITEs" with "reflects the characteristic behavior of ITEs" to avoid subjective wording and to describe objectively the role of vertical stratification in controlling the PV magnitude within the ITE core (lines 554 of the revised manuscript).

**Figure 13a:** seems too busy, are necessary the white contours? Homogenize all values of PV in text and in the figure to have the same factor x10.
**Response:** We thank the referee for the observation. We revised Figure 13a (now Figure 15a in the revised manuscript) to improve clarity while preserving essential information. We decided to retain the contours as they effectively highlight the regions of greatly reduced PV within the eddy's inner core and surface layer. To ensure consistency, PV values are now expressed using a logarithmic scale, which facilitates interpretation across orders of magnitude and is now homogenized between the figure and the main text. These changes were made in conjunction with suggestions from both referees to enhance readability and interpretability.

**l442:** "were associated with isopycnal lifting induced by CEs adjacent to the Bentayga eddy." how do you differentiate between isopycnal lifting induced by the ITE vs. the adjacent CEs?
**Response:** We revised the text to avoid directly attributing the observed isopycnal lifting solely to adjacent cyclonic eddies. We now state that the highest PV values were detected near regions of strong isopycnal lifting at the periphery of the Bentayga eddy, possibly influenced by nearby cyclonic structures (lines 560–561 of the revised manuscript). This adjustment acknowledges the complexity of the surrounding environment and the limitations of the available transect coverage.

**l443:** Change "extremely".
**Response:** Following the referee's suggestion, we replaced "extremely low-PV core" with "anomalously low-PV core" to provide a scientifically precise and objective description, emphasizing the core's distinctively low PV relative to its surroundings (line 562 of the revised manuscript).

**l453:** which specific stage?
**Response:** We revised the text to explicitly state that the predominance of APE over KE suggests that the ITE is in a mature stage of its life cycle (lines 579 of the revised manuscript).

**l455–456:** you should mention at least here the lack of synopticity of the observations you used for these computations. Also, the Bentayga radius was different than for the PUMP eddy. What is the climatological first baroclinic Rossby radius of deformation for the region of study (Chelton et al., 1998)?

**Response:** We thank the referee for this insightful observation. Following the suggestion, we have now included an estimate of the climatological first baroclinic Rossby radius for the study region, calculated using the parameterization from Chelton et al. [1998], which yields a value of approximately 42.8 km. For comparison, the PUMP eddy presented a total radius of 46 km and a core radius of 30 km, which are broadly comparable—though not identical—to those estimated for Bentayga (total radius of 70 km, radial distance of maximum velocity of 41 km and core radius of 23 km based on our revised diagnostics). Importantly, we now clarify that the energetic computations discussed in this section were derived from the OceT transect, which is not fully synoptic. This limitation and its potential implications for the interpretation of integrated quantities are discussed in depth in Supplementary Notes 1-5. As noted there, we consider our final values to be conservative estimates given the constraints of the dataset.

**l505:** "Among these features, ITEs are long-lived mesoscale structures that persist for several months and play a critical role in maintaining the region's elevated EKE. " ITEs can be long-lived or not. This is not an intrinsic characteristic of them. Rephrase.
**Response:** We revised the text to specify that long-lived ITEs are mesoscale structures that persist for several months, thus avoiding a generalization across all ITEs (line 656 of the revised manuscript).

**l517:** "its", what eddy are you referring to here?
**Response:** We thank the referee for pointing out this ambiguity. In the revised manuscript, we clarified that the possessive "its" refers specifically to the Bentayga eddy. The sentence now reads: "For Bentayga, a combination of hydrographic and satellite-based evidence suggests the recent trapping of upwelling waters from filaments..." This modification ensures that the subject is explicitly stated and eliminates potential confusion (line 668).

**l517:** "The hydrographic properties of its inner core suggest significant trapping of upwelling waters from filaments", how? Explain.
**Response:** We thank the referee for requesting clarification. In response, we have revised the discussion to include a more detailed justification of the proposed trapping of upwelling waters. Specifically, we now highlight the combined use of satellite imagery and in situ hydrographic properties to support this interpretation. Satellite observations from August 2022 (Fig. 17 of the revised manuscript) show filaments of cool sea surface temperature and elevated chlorophyll-a concentrations extending from the upwelling zone near Cape Juby and Cape Bojador and spiraling toward the eddy center, consistent with surface convergence and filament entrainment. Complementarily, hydrographic measurements from November 2022 reveal that the eddy's inner core exhibits relatively low oxygen concentrations, low apparent oxygen utilization (AOU), and slightly cooler and fresher properties relative to the surrounding thermocline (Figs. 7 and 10 of the revised manuscript). These are characteristic of recently subducted upwelled waters that have not undergone extensive remineralization. This updated interpretation is now clearly stated in the revised manuscript (Section 4, lines 668-674).

**Figure 15:** should be improved for clarity.
**Response:** The figure referred to as Figure 15 in the original manuscript has been revised for improved clarity and is now presented as Figure 17 in the updated version. The new figure focuses on a key moment during the eddy's intensification phase (11 August 2022) and illustrates the interaction between the Bentayga eddy and upwelling filaments using chlorophyll-a and sea surface temperature fields, overlaid with geostrophic streamlines. Compared to the original, the revised figure reduces visual clutter, enhances the spatial resolution of key features, and includes clearer labeling of coastal landmarks and eddy position. These changes provide a more focused and interpretable representation of the hypothesized convergence dynamics.

**Figure 15:** In Fig. 15 I don't see any trapping of waters from filaments. I see high chl and low temperature waters advected by filaments near the eddy, it seems that these waters are moving around the eddy under the influence of the swirling velocities. During eddy formation, eddies trap water within their cores, and this water can be isolated from the surroundings over long distances if the eddy is non-linear (Chelton et al., 2011). In Fig. 15 I see interaction between the eddy and filaments, but I don't see trapping. Indeed, the eddy core characteristics seem quite constant in all snapshots.
**Response:** We thank the referee for this insightful observation. We agree that the satellite snapshot shown in Figure 17 (formerly Fig. 15) does not directly demonstrate the full process of water trapping

within the eddy core. Rather, it illustrates the interaction between coastal upwelling filaments and the Bentayga eddy, showing chlorophyll-rich and cold filaments spiraling toward the eddy center. These patterns are consistent with surface convergence, a process widely recognized as a precursor to subduction and filament trapping in anticyclonic eddies [e.g., Sangrà et al., 2005, 2007] While the eddy core appears relatively stable in terms of surface properties across the shown period, the interpretation of filament trapping is additionally supported by subsurface hydrographic observations from November 2022, which reveal relatively low oxygen concentrations and low AOU values within the eddy core, along with slightly cooler and fresher anomalies. These characteristics are indicative of recently subducted waters that have not undergone significant remineralization. Therefore, although the figure does not provide direct evidence of trapping, it documents surface convergence consistent with filament entrainment, and when considered together with the hydrographic anomalies observed months later, supports the interpretation of lateral incorporation of upwelled waters into the Bentayga eddy during its intensification phase.

**l522–524:** "Interactions between mesoscale eddies, ... Such processes likely contributed to the variability observed in the Bentayga eddy, ", provide evidence of this, or remove.
**Response:** We thank the referee for the comment. In the original version, the claim that eddy–eddy interactions contributed to the Bentayga eddy's variability lacked explicit support in that section. We have now reorganized the discussion to connect more clearly with later content, where we qualitatively describe the presence of multiple mesoscale vortices flanking the eddy during the eIMPACT2 campaign. Moreover, we now reference a newly added supplementary video (DOI provided in the revised manuscript and Supplementary Notes) showing the full satellite-derived trajectory of Bentayga, which highlights the persistent proximity of other eddies throughout its lifetime. While this does not constitute direct observational evidence of structural merging or water exchange, it strengthens the plausibility of eddy–eddy interactions as a contributing factor to the observed hydrographic complexity. The manuscript has been revised accordingly to clarify the nature and limitations of this interpretation.

**l526–531:** This is interesting. Provide exact depths in text (they are different than in Figure B1). Maybe plot more depth layers in Figure B1 if needed. Can you show original data together with interpolated data to check at least visually the interpolation? It seems that in the layers shown in B1 the circulation is anticyclonic, but the eddy seems to be elongated along the north-south direction at 328 m, having an elliptical shape. Do you have an explanation for this? "possibly representing a center with two nuclei" how do you know this? if no evidence, then remove. In text, guide the reader to the exact subplots. In B2, the larger velocities coming from the east are the signal of interaction with another feature that alter the circular shape of the eddy? What remote-sensing observations detect about this?
**Response:** We thank the referee for this detailed and insightful comment. In response, we have revised the corresponding paragraph in the Discussion (lines 680–694) to include the exact depth levels and to guide the reader more precisely to the relevant subplots in Figs. B1 and B2. We now explicitly address the elliptical shape of the eddy observed at 328 m (Fig. B1c), oriented along a north–south axis, and have removed the previous speculative reference to a "center with two nuclei," as there is no direct evidence supporting that interpretation. Regarding interpolation, we clarify in the captions of Figs. B1 and B2 that the fields shown are derived from objectively mapped SeaSoar VMADCP data. To improve transparency, we now include additional figures (Figs. B3 and B4) showing the original velocity measurements along the ship's track. This allows for a visual comparison between the interpolated fields and the underlying observations. The elongated pattern at 328 m is interpreted as the result of flow deformation caused by nearby mesoscale features. This interpretation is further supported by enhanced eastward velocities at 408 m (Fig.B1d), which may reflect ambient shear or interaction with a neighboring eddy. Although direct remote-sensing observations do not resolve structures at these depths, the persistent clustering of mesoscale features observed in the supplementary video product (Supplementary Note 6) lends support to this interpretation. All these clarifications have been incorporated into the revised text of the Discussion section.

**l532–537:** How is this impacting your integrations?
**Response:** We thank the referee for this important question. The revised analysis now addresses the vertical coherence of the Bentayga eddy and its implications for integrated diagnostics more rigorously. Specifically, the original interpretation previously presented in lines 532–537 has been removed and replaced with a more robust framework developed in Section 3.5 and Supplementary Note 4. In Section 3.5, we characterize the vertical coherence of the eddy using the correlation between observed azimuthal

velocities and solid-body rotation fits ($r_{xy}$), along with the structure of the angular velocity $\omega_\theta$. While high rotational coherence ($r_{xy} \geq 0.9$) is maintained down to $\sim$350 m, it decreases and becomes more variable below this depth. However, this metric alone was not used to define the vertical integration limit. Instead, we apply a physically grounded criterion based on the ratio between the shell velocity and the eddy's translational speed. This approach captures the depth range over which the eddy can coherently advect its outer ring during propagation. Following this criterion, we used 550 m as the upper integration limit. To quantify the effect of this choice, in Supplementary Note 4 we performed a sensitivity test showing that reducing the integration depth from 550 m to 340 m results in changes of approximately 20% in APE, AHA, and ASA, while KE remains comparatively less sensitive due to weaker circulation below 340 m. These findings demonstrate that vertical integration limits have a non-negligible impact on energy and tracer anomaly estimates. Therefore, the 550 m limit represents a conservative yet dynamically consistent upper bound, ensuring that the derived integrated values remain physically meaningful. This methodology is further contextualized in Supplementary Note 5, where we discuss how to cautiously interpret these diagnostics given the vertical extent and temporal sampling characteristics of the OceT dataset.

**l538–542:** What is the relation of this to your results?
**Response:** We appreciate the referee's request for clarification. The paragraph in lines 538–542 aims to place the observed weakening and irregularity of the eddy-induced circulation below 330 m in a broader dynamical context. While our results show that the anticyclonic structure remains coherent down to approximately 250 m, it becomes progressively less defined at greater depths. We refer to the study by Martínez-Marrero et al. [2019]—which reported intense near-inertial wave activity at the base of another ITE in the same region—as a plausible physical mechanism that could contribute to the observed weakening and variability. Additionally, the geographic proximity of Bentayga to energetic internal wave sources (e.g., the Canary Archipelago and nearby seamounts) justifies considering the potential role of internal tide activity. Although the dataset used in this study does not allow us to resolve these high-frequency processes explicitly, acknowledging their potential influence helps frame the observed vertical structure and its limits, especially when interpreting integrated diagnostics or assessing vertical coherence.

**Discussion (and in general):** the excessive use of short paragraphs is confusing and breaks the fluidity and coherence of the text. Try to group and link ideas.
**Response:** We thank the referee for this stylistic recommendation. Following this suggestion, we have revised the structure of the Discussion section to improve coherence and flow. Several short paragraphs have been merged where appropriate, and transitional sentences have been added to ensure smoother progression between ideas. We believe these changes enhance the clarity of the arguments and the overall readability of the discussion.

**l544:** Remove "true"; weakens argument.
**Response:** We removed the word "true" to avoid unnecessary subjectivity (line 709 of the revised manuscript).

**l565:** Avoid "freshwater" – ocean water is more or less salty.
**Response:** Following the referee's suggestion, we revised the text to refer more precisely to "equivalent freshwater fluxes" instead of simply "freshwater", thus clarifying that this flux is derived from the salt anomaly relative to the background salinity (line 742 of the revised manuscript).

**l568:** "as an ITE, Bentayga is distinct ": some of the eddies you mention in the previous sentence are also subsurface intensified. Rephrase.
**Response:** We removed the subjective statement that Bentayga is distinct and rephrased the text to focus on the role of its subsurface intensification in enabling the efficient transport of thermal and saline anomalies into the ocean interior, consistent with previously observed behavior in ITEs (lines 745–746 of the revised manuscript).

**l571–575:** "Interactions with upwelling filaments extending from the northwestern African coast (Fig. 15) likely imbued the Bentayga eddy with heat and salt signatures characteristic of these features ", what do you mean? See my comment before. Expand this paragraph, give evidence or remove.

**Response:** Following the referee's suggestion, we revised the paragraph to present a more cautious and scientifically grounded interpretation. We now describe the observed convergence of filaments towards the eddy center as suggestive but not conclusive, and present the hypothesis that the waters incorporated into the Bentayga eddy may have progressively deepened during offshore migration, following isopycnal adjustment and conservation of potential vorticity (lines 747–755 of the revised manuscript).

**l576–580:** Link with your results? Consider moving this paragraph to the introduction or remove.
**Response:** We thank the referee for this observation. In the revised manuscript, we now support the biogeochemical relevance of the Bentayga eddy by incorporating a new subsection in the Results section (Sect. 3.4), as suggested by Referee 1. There, we analyze the spatial distribution of dissolved oxygen and apparent oxygen utilization (AOU), showing a subsurface oxygen-deficient and low AOU signature that likely reflects the recent trapping of waters from Saharan upwelling filaments. Although we cannot fully confirm this hypothesis with the available data, the observed anomalies align qualitatively with satellite imagery from August 2022, which shows filaments converging toward the eddy's center. Therefore, we consider that the paragraph in lines 576–580 of the original version of this manuscript, is now effectively linked to our results and is better contextualized. For this reason, we have chosen to retain it in the Discussion section, where it helps underscore the potential ecological implications of the Bentayga eddy (lines 756-762).

**l595:** "It is plausible that the six identified AEs ..." provide evidence or remove.
**Response:** Following the referee's suggestion, we removed the speculative statement regarding the six identified AEs potentially undergoing similar interactions. The revised text now focuses on the persistent presence of multiple mesoscale vortices near the Bentayga eddy, supported by visual inspection of satellite altimetry imagery (with reference to a supplementary video; lines 775–783 of the revised manuscript).

**l603, l621, etc.:** "potentially biogeochemical properties " provide evidence or remove.
**Response:** We thank the referee for this valuable observation. In the revised manuscript, we have replaced the ambiguous phrase "potentially biogeochemical properties" with a more precise description of the observed hydrographic signatures. Specifically, we now refer to "oxygen-deficient waters with relatively low AOU values" within the eddy core (Section 3.4), which are consistent with the recent entrainment of upwelling filament waters originating from the Saharan coast. Although the available dataset does not include direct measurements of nutrients or carbon, the combined evidence from in situ oxygen and AOU anomalies and satellite-derived surface convergence patterns supports the interpretation that Bentayga likely transported biogeochemically distinct waters into the ocean interior. We have revised the text accordingly to more accurately reflect these observations while maintaining a cautious and evidence-based interpretation.

**l11–12:** "The intrathermocline nature of the eddy developed during the growth phase, and was shaped by surface convergence enhanced by upwelling filament interactions, followed by isopycnal deepening offshore.", where do you explain and demonstrate this?
**Response:** Following the referee's suggestion, we revised our analysis and updated the corresponding statement in the abstract to reflect a more cautious interpretation (see lines 668–676 and 747-755 of the revised manuscript). We now present the development of the intrathermocline nature and the associated processes (surface convergence and isopycnal deepening) as plausible hypotheses rather than demonstrated facts, based on the available observations (lines 12–16 of the revised manuscript).

We hope that our responses address all the reviewers' concerns and that the revised manuscript is now suitable for publication in *Ocean Science*.

**References**

J.T. Allen, D.A. Smeed, A.J.G. Nurser, J.W. Zhang, and M. Rixen. Diagnosis of vertical velocities with the qg omega equation: an examination of the errors due to sampling strategy. *Deep Sea Research Part I: Oceanographic Research Papers*, 48(2):315–346, 2001. ISSN 0967-0637. doi:

https://doi.org/10.1016/S0967-0637(00)00035-2. URL `https://www.sciencedirect.com/science/article/pii/S0967063700000352`.

Alexandre Barboni, Solange Coadou-Chaventon, Alexandre Stegner, Briac Le Vu, and Franck Dumas. How subsurface and double-core anticyclones intensify the winter mixed-layer deepening in the mediterranean sea. *Ocean Science*, 19:229–250, 3 2023. ISSN 18120792. doi: 10.5194/os-19-229-2023.

Bàrbara Barceló-Llull, Pablo Sangrà, Enric Pallàs-Sanz, Eric D. Barton, Sheila N. Estrada-Allis, Antonio Martínez-Marrero, Borja Aguiar-González, Diana Grisolía, Carmen Gordo, Ángel Rodríguez-Santana, Ángeles Marrero-Díaz, and Javier Arístegui. Anatomy of a subtropical intrathermocline eddy. *Deep-Sea Research Part I: Oceanographic Research Papers*, 124:126–139, 6 2017. ISSN 09670637. doi: 10.1016/j.dsr.2017.03.012.

Igor Belkin, Annie Foppert, Tom Rossby, Sandra Fontana, and Chris Kincaid. A double-thermostad warm-core ring of the gulf stream. *Journal of Physical Oceanography*, 50:489–507, 2 2020. doi: 10.1175/JPO-D-18-0275.1. URL `https://doi.org/10.1175/JPO-D-18-0275.s1`.

Anthony Bosse, Ilker Fer, Jonathan M. Lilly, and Henrik Søiland. Dynamical controls on the longevity of a non-linear vortex : The case of the lofoten basin eddy. *Scientific Reports*, 9, 12 2019. ISSN 20452322. doi: 10.1038/s41598-019-49599-8.

Dudley B. Chelton, Roland A. deSzoeke, Michael G. Schlax, Karim El Naggar, and Nicolas Siwertz. Geographical variability of the first baroclinic rossby radius of deformation. *Journal of Physical Oceanography*, 28:433 – 460, 1998. doi: 10.1175/1520-0485(1998)028⟨0433:GVOTFB⟩2.0.CO;2. URL `https://journals.ametsoc.org/view/journals/phoc/28/3/1520-0485_1998_028_0433_gvotfb_2.0.co_2.xml`.

Eric A. D'Asaro. Generation of submesoscale vortices: A new mechanism. *Journal of Geophysical Research: Oceans*, 93:6685–6693, 6 1988. ISSN 0148-0227. doi: 10.1029/jc093ic06p06685.

P. Garreau, F. Dumas, S. Louazel, A. Stegner, and B. Le Vu. High-resolution observations and tracking of a dual-core anticyclonic eddy in the algerian basin. *Journal of Geophysical Research: Oceans*, 123:9320–9339, 12 2018. ISSN 21699291. doi: 10.1029/2017JC013667.

A. Martínez-Marrero, B. Barceló-Llull, E. Pallàs-Sanz, B. Aguiar-González, S. N. Estrada-Allis, C. Gordo, D. Grisolía, A. Rodríguez-Santana, and J. Arístegui. Near-inertial wave trapping near the base of an anticyclonic mesoscale eddy under normal atmospheric conditions. *Journal of Geophysical Research: Oceans*, 124:8455–8467, 2019. ISSN 21699291. doi: 10.1029/2019JC015168.

James C. McWilliams. Vortex generation through balanced adjustment. *Journal of Physical Oceanography*, 18:1178–1192, 1988. doi: 10.1175/1520-0485(1988)018⟨1178:VGTBA⟩2.0.CO;2.

P. Sangrà, J. L. Pelegrí, A. Hernández-Guerra, Igor Arregui, J. M. Martín, A. Marrero-Díaz, A. Martínez, Andry W. Ratsimandresy, and A. Rodríguez-Santana. Life history of an anticyclonic eddy. *Journal of Geophysical Research: Oceans*, 110:1–19, 3 2005. ISSN 21699291. doi: 10.1029/2004JC002526.

P. Sangrà, M. Auladell, A. Marrero-Díaz, J. L. Pelegrí, E. Fraile-Nuez, A. Rodríguez-Santana, J. M. Martín, E. Mason, and A. Hernández-Guerra. On the nature of oceanic eddies shed by the island of gran canaria. *Deep-Sea Research Part I: Oceanographic Research Papers*, 54:687–709, 5 2007. ISSN 09670637. doi: 10.1016/j.dsr.2007.02.004.

Lu Sha Yu, Anthony Bosse, Ilker Fer, Kjell A. Orvik, Erik M. Bruvik, Idar Hessevik, and Karsten Kvalsund. The lofoten basin eddy: Three years of evolution as observed by seagliders. *Journal of Geophysical Research: Oceans*, 122:6814–6834, 8 2017. ISSN 21699291. doi: 10.1002/2017JC012982.

---

## Author Response (AR2)

**Response to Referee 1 - Final Comments**
**on the reviewed manuscript entitled:**
**"Mesoscale dynamics of an intrathermocline eddy in the Canary Eddy Corridor"**
**(doi: 10.5194/egusphere-2025-99)**

**July 2025**

We sincerely thank the handling editor, Prof. Ilker Fer, and Dr. Anthony Bosse (Referee #1) for their thoughtful and constructive final comments. We have carefully addressed the remaining issues raised during this last round of review. All comments were gratefully received and have been incorporated into the revised version of the manuscript. Each comment is presented as stated by the referee, followed by our response.

**Response**

**On dissolved oxygen data::** "There is no mention of dissolved oxygen calibration. Did you perform winkler oxygen titration during the cruise and compared the CTD values with it?"
**Response:** We thank Dr. Bosse for pointing this out. Winkler oxygen titration was indeed performed during the cruise, specifically for calibrating the dissolved oxygen measurements collected during the OceT phase. The resulting calibration yielded a linear relationship with a slope of 1.0957, an offset of $-1.0462$ $\mu$mol kg$^{-1}$, and a coefficient of determination $R^2 = 0.9908$. The precision of the Winkler titrations, expressed as the coefficient of variation, was 0.54% ($\pm 1.01$ $\mu$mol kg$^{-1}$). Although no discrete oxygen samples were collected during the SeaSoar phase due to its continuous sampling configuration, oxygen records from both phases were highly consistent, indicating stable and reliable sensor performance throughout the cruise. These details have been incorporated into the final version of the manuscript (lines 182–187 of the revised version). Furthermore, the description of Section 3.4 has been updated (lines 467-493 of the revised manuscript) to reflect the use of calibrated oxygen data. While the main qualitative features of the oxygen distribution remain unchanged, the corrected values resulted in minor adjustments to the quantitative interpretation. These include slight shifts in the magnitude of subsurface oxygen minima and the amplitude of lateral gradients, but they do not alter the overall conclusions of the study.

**Figures 5 and 7:** Figure 5 : It seems more logic to me to have T,S and then N in third panel. (same with fig 7 : T,S then sigma)
**Response:** We agree with Dr. Bosse. The panels in Figures 5 and 7 have been reordered accordingly: conservative temperature ($\Theta$) and absolute salinity ($S_A$) now appear in the first two panels, followed by brunt-väisälla frequency (N) in Figure 5 and potential density anomaly ($\sigma_\theta$) in Figure 7.

**l431/Figure 9:** I would like the authors to discuss the effect of cyclogeostrophy as a source of the observed ageostrophic velocities..
**Response:** We thank Dr. Bosse for this important remark. A dedicated evaluation of the cyclogeostrophic balance as a source of the observed ageostrophic velocities has been included in the revised manuscript (Section 3.3, lines 446–466). In particular, we calculated the cyclogeostrophic Rossby number and analyzed the relative contributions of centripetal and Coriolis forces within the eddy structure. Furthermore, a brief discussion of these results has been added to the Discussion section (lines 767–778) to better contextualize the dynamical regime of the eddy in relation to previous studies in the region. These additions reinforce our interpretation of the eddy as being partially in cyclogeostrophic balance, especially near its inner core.

We hope that our responses satisfactorily address all of the reviewer's comments, and that the revised manuscript is now suitable for publication in *Ocean Science*.